# WBM v.1.0.0: A scalable gridded global hydrologic model with water tracking functionality

Danielle S. Grogan[1], Shan Zuidema[1], Alex Prusevich[1], Wilfred M. Wollheim[1,2], Stanley Glidden[1], Richard B. Lammers[1]

[1]Earth Systems Research Center, Institute for the Study of Earth, Oceans, and Space, University of New Hampshire, Durham, NH, 03824, US
[2]Department of Natural Resources and the Environment, University of New Hampshire, NH, 03824, USA

*Correspondence to*: Danielle S. Grogan (danielle.grogan@unh.edu) and Shan Zuidema (shan.zuidema@unh.edu)

**Abstract.** This paper describes the University of New Hampshire Water Balance Model, WBM, a process-based gridded global hydrologic model that simulates the land surface components of the global water cycle and includes water extraction for use in agriculture and domestic sectors. WBM was first published in 1989; here we describe the first fully open source WBM version. Earlier descriptions of WBM methods provide the foundation of the most recent model version detailed here. We present an overview of the model functionality, utility, and evaluation of simulated global river discharge and irrigation water use. This new version adds a novel suite of water source tracking modules that enable analysis of flow-path histories on water supply. A key feature of WBM v.1.0.0 is the ability to identify the partitioning of sources for each stock or flux within the model. Three different categories of tracking are available: (1) primary inputs of water to the surface of the terrestrial hydrologic cycle (liquid precipitation, snowmelt, glacier melt, and unsustainable groundwater); (2) water that has been extracted for human use and returned to the terrestrial hydrologic system; and (3) runoff originating from user-defined spatial land units. Such component tracking provides a more fully transparent model in that users can identify the underlying mechanisms generating the simulated behavior. We find that WBM v.1.0.0 simulates global river discharge and irrigation water withdrawals well even with default parameter settings, and for the first time we are able to show how the simulation arrives at these fluxes by using the novel tracking functions.

## 1 Introduction

Global hydrologic models (GHMs) are one of the primary tools used in the study of macro-scale hydrology, and the past 30 years has seen the development of numerous GHMs. These include the Water Balance Model WBM (Vörösmarty et al. 1989), VIC (Liang et al. 1994), WaterGAP (Döll et al. 2003), H08 (N. Hanasaki et al., 2008a, 2008b), PCR-GLOBWB (Sutanudjaja et al., 2018), and others (Telteu et al., 2021). The terrestrial hydrology concepts and structures from these models have now been incorporated into several land surface models (LSMs), e.g., NASA LIS (Kumar et al., 2006) and the Community Land Model (CLM; Lawrence et al., 2019), and Earth system models (ESMs) such as and WRF-Hydro (Gochis et al., 2020) and the Community Earth System Model (CESM; Zeng et al., 2015), and others such as the U.S. National Water

Model (Cohen et al., 2018). GHMs represent the land surface component of the hydrologic cycle, converting time series of weather and landcover variables into estimates of water storage and flux values. These models have been applied to many questions of both basic and applied hydrology, such as climate change and other anthropogenic impacts on global river systems (Bosmans et al., 2017; Döll et al., 2012; Haddeland et al., 2014; Hanasaki et al., 2008b; Vörösmarty et al., 2000a,

2010; Wada et al., 2011), groundwater depletion (Petra Döll et al., 2014; Gleeson et al., 2012; Grogan et al., 2017; Wada et al., 2012), and the role of water extractions in sea level change (Gleeson et al., 2012; Konikow, 2011; Pokhrel et al., 2012). GHMs have also been used extensively in the study of food security and agricultural yields (Biemans and Siderius, 2019; Döll and Siebert, 2002; Elliott et al., 2014; Haqiqi et al., 2020; Liu et al., 2017; Schewe et al., 2014) as well as formed the foundation for water quality models (Mineau et al., 2015; Stewart et al., 2011a; Wit, 2001; Wollheim et al., 2008a,b;

Zuidema et al., 2018) and the inputs for flood inundation models (e.g., Yamazaki et al., 2011). Recently, GHMs have been employed in interdisciplinary studies to evaluate human-hydrologic systems and the food-energy-water nexus, such as human and economic impacts of flooding (Dottori et al., 2018), hydropower (Mishra et al., 2020; Turner et al., 2019), powerplant cooling capacity (van Beek et al., 2012; Stewart et al., 2013; Webster et al., 2022), water markets (Rimsaite, 2021), irrigation decision-making under climate change (Zaveri et al., 2016), and virtual water trade (Dalin et al., 2017;

Konar et al., 2013). Recent overviews of GHM literature are also provided in Sutanudjaja et al. (2018) and Telteu et al. (2021).

GHMs were developed to quantify land surface hydrologic fluxes at global and continental scales, and these models generally capture the macro-scale behavior of the water cycle in both natural and human systems (Telteu et al., 2021). Model limitations include poor simulation during low runoff periods and a tendency to overestimate mean annual runoff and

discharge (Zaherpour et al., 2018). While early GHMs only represented natural hydrologic fluxes, the recent addition of human impacts were shown to greatly improve river discharge estimates and in most cases lowered the overly-high estimates of average annual river flow (Veldkamp et al., 2018). Despite these improvements, there have been calls to better represent regional water management, co-evolution of the human-water system and improved human water management information in GHMs (Wada et al., 2017). A large challenge for macro-scale hydrological modelers is to better capture the human

decision-making around water movement, use, and consumption. One method for achieving this is via linking models from the social sciences to hydrological models (e.g., Mishra et al., 2020; Webster et al, 2022; Zaveri et al., 2016). The model described in this paper, WBM v.1.0.0, captures all the major land surface water stocks and fluxes with a focus on human alterations of the water cycle. A significant contribution of this model version is the ability to track water depending on its source or use through the entirety of the system, highlighting how movement of water for human use interacts with the

natural water system.

## 1.1 Tracking water sources

As pressures on water resources increase through both climate change and intensifying human water demand (e.g., Vörösmarty et al., 2000a), it is important to know the origin of regional water resources. While some basins may be supplied by steady precipitation or recharging aquifers, others rely on seasonal snowpack, fossil groundwater, irrigation returns, glacial melt, or monsoon rains. Each of these water sources comes with their own set of management challenges and opportunities, making knowledge of water sources a vital component of water resource planning. It may seem obvious where a basin or region's water comes from; however, human water use introduces complexities into the terrestrial water cycle that can obscure the often lengthy and circuitous pathway that waters take from source to use (Grogan et al., 2017; Zuidema et al., 2020). The discussion here is confined to the terrestrial water cycle as GHMs do not, by definition, simulate the atmosphere. Under natural conditions, most of the water that enters a river basin travels from the land surface through soils and groundwater through headwater basins (Alexander et al., 2007) and then through the full river system to the ocean or endorheic outlet. Humans withdraw large quantities of water from these natural pathways and because no human water use activity is completely consumptive water extracted from river and groundwater systems is returned either to its original source or diverted to an alternate pool. Irrigation accounts for ~70% of all freshwater withdrawals (Rosegrant and Cai, 2002), and globally is ~50% efficient (Döll and Siebert, 2002; Gleick et al., 1993), returning approximately half of all extracted irrigation water back to surface water and groundwater storages. The repetition of this activity causes iterative cycles of water extraction and return over annual to decadal time scales, creating complex, circuitous pathways. The pathways that water travels impact water quality (Huang et al., 2022; Mineau et al., 2015), food security (Kadiresan and Khanal, 2018), and governance of water resources through transboundary interactions (Zeitoun and Mirumachi, 2008). Furthermore, humans develop hydro-infrastructure to intentionally impound (Lehner et al., 2011; Zuidema and Morrison, 2020) and divert rivers (Ghassemi and White, 2007), and engage in artificial recharge of groundwater pools (Dillon et al., 2019). These activities divert water through natural and artificial stocks, masking the identity of the original source of the water.

Understanding the journey of certain sources of water illuminates their role in downstream water resource issues and how human-induced complex pathways make attribution of upstream changes to downstream effects increasingly difficult. In this paper we present three examples of tracking parcels of water through the hydrological cycle by preserving key attributes related to water sources, return flows from water extraction, and an identifier assigned to all runoff generated from a given land area. This novel modeling method maintains the identity of a parcel of water as it travels through natural and anthropogenic pathways, illuminating previously obscured connections between sources, uses and fates, as well as offering a potential useful tool for understanding water quality changes throughout watersheds.

In this paper, we provide a detailed description of WBM v.1.0.0, its performance compared to observations of global hydrologic fluxes when using default parameterizations, and examples of how the tracking functionality can be used to evaluate the role of human alterations to the global hydrologic cycle. We review previous studies that have used earlier

versions of WBM, and provide guidance for setting up and running WBM v.1.0.0.

## 2 WBM model description

### 2.1 General overview

WBM (Grogan, 2016; Wisser et al., 2010a) is a process-based, gridded hydrologic model that simulates spatially and temporally varying water volumes and quality (Fig. 1) operating at daily time steps. It was one of the first GHMs developed

(Vörösmarty et al., 1989), and is now joined by many other similar GHMs and LSMs in its representation of the terrestrial portion of the water cycle. WBM represents all major land surface components of the hydrologic cycle, and tracks fluxes and balances between the atmosphere, above-ground water storages (e.g. snowpack), soil, vegetation, groundwater, and runoff. A digitized river network connects each grid cell to the next, enabling simulation of flow through river systems. WBM includes domestic and industrial water requirements and use, agricultural water requirements and use (irrigation and

livestock), and hydro-infrastructure (dams and inter-basin transfers). While the model is considered global, it can be run for any region and any spatial resolution given available input data at the appropriate scale. For example, WBM has been operated at a local scale of ~120-meter grid cell resolution over a 400 km$^2$ watershed (Stewart et al., 2011a), and at global scales (Grogan et al., 2017; Wisser et al., 2010a) (Table A1).

WBM is modular and is able to accept climate, land use/land cover, water management, and water demand inputs from other models and data sources, such as glacier melt models (e.g., Huss and Hock, 2015; Rounce et al., 2020), reservoir operation data (Zuidema et al., 2020), or econometric land use models (Zaveri et al., 2016). The modular components can be turned on or off with binary flags in a model initialization file. This allows users to turn on/off the anthropogenic interaction with the water cycle, made up of the following functions: (1) water extraction for irrigation, (2) rainfed crop water evapotranspiration,

(3) water use for livestock, municipalities, and industrial production, (4) inter-basin transfers, and (5) reservoirs and dams. In contexts where water quality is simulated, fluxes of solutes from other models such as the terrestrial biogeochemical model PnET (Aber et al., 1997) provide the relevant boundary conditions to WBM (Samal et al., 2017). While WBM is modular, the core hydrologic framing requires the following inputs: a digital river network identifying flow direction at the resolution of the model grid (such as STN-30p (Vörösmarty et al., 2000b), MERIT (Eilander et al., 2021; Yamazaki et al.,

2019), HydroSHEDS (Lehner et al., 2008), or any other standard flow grids); soil available water capacity and root depth; daily average temperature, and total daily precipitation. The model requires a spinup step to allow water stocks to reach equilibrium prior to the model simulation period. At the beginning of a simulation, large reservoirs (described in Section 2.2.4) are initialized at 80% of their full capacity, the soil moisture storage pool (described in Section 2.2.1) is initialized at

50% of capacity, and all other stocks begin at 0% capacity. We recommend a minimum spinup time of at least 10 years, using a representative historical climatology of daily weather inputs to drive the spinup period. Model methods and results described here specifically refer to the open source release of WBM v.1.0.0 (Grogan and Zuidema, 2022).

Most features of WBM have been described in prior publications, and the documentation included in the supplemental material and WBM GitHub repository provides details and equations for all the WBM v.1.0.0 model methods. Here, we give a general overview, describe updates and additions to previously published methods, and point to the most recent and relevant citations that accurately describe the version of the model presented here.

## 2.2 Model description

WBM simulates terrestrial hydrologic fluxes at a daily time step using rasterized grids in either geographic or projected coordinates. Inputs to WBM can be in any GDAL-readable format, and some parameter sets and databases are input as delimited text files. Most geospatial data are input as (potentially multi-layer) raster grids commonly in GeoTIFF or NetCDF formats. Many parameters controlling WBM behavior can be input as single scalar values or as geospatial grid or vector fields. All inputs are coordinated through simple text *initialization* files. This structure makes it possible to build simple scripts that automatically generate WBM model inputs, which can be used for developing batches of simulations, or for performing sensitivity and uncertainty analyses using user preferred algorithms.

Previous work to parameterize WBM to match historical observation has used a combination of manual and automated fitting. Table 1 presents a cross-section of parameters that are typically varied to control WBM's response in individual watersheds. Default values are typically found to be reasonable for both forested temperate watersheds and global average conditions, but poor correspondence with observational data in some watersheds is expected when simulating large regions with default values. WBM users should calibrate the parameters listed in Table 1 (and possibly others) for regional modeling. A complete listing of parameters and inputs is provided with the model source code as a spreadsheet.

### 2.2.1 Land surface fluxes

Water enters the land surface – and therefore enters the WBM modelling framework – via precipitation. This precipitation can be intercepted by the vegetative canopy, collect as snow, enter soil storage, or become surface runoff.  Water that enters soils in excess of the soil's field capacity infiltrates into the shallow groundwater pool. Bare surfaces and vegetation collectively lose water to the atmosphere through evapotranspiration.

*Precipitation and snow:* Precipitation is partitioned into solid (snow) and liquid (rain) within WBM according to temperature thresholds.  Snow accumulation and melt, both expressed in terms of snow water equivalent (SWE), are also functions of temperature thresholds. These snow thresholds are fully described in Wada et al. (2012) and Grogan (2016). Accumulated

snow is represented as a single layer. For regions with high elevational gradients, a sub-grid cell binned distribution of elevations can be used to partition the grid into liquid/solid precipitation portions and snow accumulation/melt portions. If sub-grid elevation snow processes are not used, the same snow processes apply to the entire grid cell. The sub-grid cell elevation method described in Mishra et al. (2020) and Grogan et al. (2020) is elaborated on here. The elevation distribution of each model grid cell is calculated from a 30 or 500 meter or finer resolution digital elevation model (DEM), resulting in binned elevation categories of $\Delta H$ vertical bands. The size of the bins is user-defined and can be from 0 to 5000 meters, with a default bin $\Delta H$ size of 250 meters. A temperature lapse rate, $L$ [°C/km], is applied to the mean daily temperature, $T$ [°C] at the reference elevation, $H_{ref}$ [m] for each binned elevation category, resulting in an adjusted mean temperature, $T_e$ [°C], for the portion of each grid cell in elevation bin category $e$.

$$T_e = T + \frac{L}{1000}\left(H_e - H_{ref}\right) \tag{1}$$

The reference elevation, $H_{ref}$ [m], is the average elevation of the grid cell represented by the temperature dataset. Precipitation rates are assumed to be equal across all elevation bins $e$, such that $P^e = P$, where $P^e$ [mm/day] is the precipitation rate at elevation $e$, and $P$ [mm/day] is the input precipitation rate. Snow water equivalent (SWE) in elevation bin $e$, $S^e$ [mm], is updated through timesteps of length $dt$ :

$$\frac{dS^e}{dt} = P_s^e - M^e \tag{2}$$

Where the frozen precipitation rate, $P_s^e$ [mm/day] is a function of the temperature at elevation $e$, $T^e$ [°C], and a reference temperature $T_s$ [°C]:

$$P_s^e = \begin{cases} P \ if \ T^e < T_s \\ 0 \ if \ T_s \leq T^e \end{cases} \tag{3}$$

and calculation of snowmelt at elevation bin $e$, $M^e$, [mm/day] follows the methods from Willmott et al. (1985):

$$M^e = \begin{cases} 2.63 + 2.55\,T^e + 0.0912\,T^e P \ \ if \ T_m < T^e \\ 0 \qquad\qquad\qquad\qquad\quad if \ T^e \leq T_m \end{cases} \tag{4}$$

Total SWE, $S$, [mm/day] in the grid cell at each time-step is the sum of all SWE values at each elevation band $e$ multiplied by the corresponding fraction of grid cell area represented by elevation bin $e$, $f^e$:

$$S = \sum_{e=1}^{n} S^e f^e \tag{5}$$

Variables controlling SWE accumulation include the snowfall threshold $T_s$, with a default value of -1 °C; the snow melt threshold $T_m$, with a default value of 1 °C; and the lapse rate L, with a default value of -6.4 °C/km. Both $T^e$ and $L$ can be constants for the whole simulation domain, or they can be a spatially variable gridded input layer.

At high elevations and cold climates, it is a common case that annual snowfall exceeds annual snowmelt volume. In reality, this excess snowpack converts to ice and forms glaciers. WBM does not internally simulate glacier formation or dynamics;

this causes unrealistic, infinite snow accumulation. To address this problem, users can define a threshold (e.g., 5000 mm of snow water equivalent) above which snow water volumes are shifted down elevation bands on the date of annual snowpack minimum (assumed to be August 15 in the Northern hemisphere and February 15 in the Southern hemisphere). If there is no elevation bin in the grid cell in which snow is melting, snow water is further shifted downstream to the next grid cell, following the direction of flow as defined by the digital river network.

*Canopy interception:* Vegetation intercepts incoming precipitation, preventing some of the total precipitation from reaching soils below and adding to the total evapotranspiration flux. The canopy intercepts liquid precipitation only. WBM uses canopy rain interception formulations from Deardorff (1978) and Dickinson (1984),:

$$\frac{dW_i}{dt} = (P - P_t) - E_c, \quad where \ W_i \leq W_i^{max} \tag{6}$$

where $W_i$ is canopy water storage [mm], $W_i^{max}$ [mm] is the canopy water storage capacity, $P$ [mm/day] and $P_t$ [mm/day] are liquid precipitation and throughfall (see section *precipitation and snow* above), and $E_c$ [mm/day] is evaporation of the canopy water.

Canopy water storage is limited by its capacity $W_i^{max}$ which is be proportional to the Leaf Area Index (LAI) [m$^2$/m$^2$]

$$W_i^{max} = C_{LAI} \cdot LAI \tag{7}$$

where $C_{LAI}$ is canopy interception coefficient [mm], which typically ranges from 0.15 to 0.25 mm (Dingman, 2002); WBM uses a default value of 0.2 mm, as suggested in Dickinson (1984).

The canopy water evaporation rate $E_c$ [mm/day] is a function of canopy water storage, $W_i$ [mm], canopy water storage capacity, $W_i^{max}$ [mm], and the open water evaporation rate, $E_{ow}$ [mm/day] (Deardorff, 1978):

$$E_c = E_{ow} \left( \frac{W_i}{W_i^{max}} \right)^{2/3} \tag{8}$$

Throughfall is then calculated as the amount of water over-filling the canopy interception pool, while accounting for evaporation over the course of the time-step.

*Open water and Impervious surfaces:* WBM represents direct storm runoff over impervious surfaces (Zuidema et al., 2018), which prevents water from entering soil and increases storm runoff. If provided with a map of impervious surface fraction of grid cell area, WBM assumes no soil water holding capacity and does not calculate canopy interception on those areas. To define the fraction of precipitation that is routed directly to streams, WBM calculates an effective impervious area adapted from Alley and Veenhuis (1983):

$$f_{imp}^{eff} = f_{imp}^{0.4} \qquad (9)$$

Given an input dataset of the fraction of grid cell open water, $f_{ow}$, areas (e.g., lakes and ponds), WBM treats open water areas as direct contributors of storm runoff to river systems; open water grid cell have no soil infiltration, surface retention pool, or shallow groundwater pool. WBM limits the sum of impervious surface and open water areas to 97.5% of grid cell area for continuity, except for expansive lakes occupying entire grid cells which are masked from any terrestrial water balance calculations. Endorheic lake grid cells are also fully masked from terrestrial water balance calculations; they are treated as water outlets in the same way ocean grid cells adjacent to river mouths are outlets. Direct storm runoff, $R_{strm}$ [mm/day], is calculated as the sum of incoming precipitation, $P$ [mm/day], and snow melt water, M [mm/day], multiplied by the sum of the effective impervious area fraction, $f_{imp}^{eff}$, and open water fraction, $f_{ow}$:

$$R_{strm} = \left(f_{imp}^{eff} + f_{ow}\right)(P + M) \qquad (10)$$

Storm runoff, $R_{strm}$ [mm/day], is routed directly to streams. The remainder of precipitation and snowmelt water are routed to soil infiltration. If soil is already saturated, this remainder contributes to surface runoff and shallow groundwater recharge (see below for descriptions of these processes); Hortonian (infiltration excess) flow is not simulated.

*Soil moisture balance:* Soil moisture balance, $W_S$ [mm], is calculated by tracking a grid cell's water inputs, water outputs, and soil moisture pool holding capacity. WBM simulates a single soil layer, and does not explicitly represent verticle fluxes of water through the soil. The soil moisture pool's available water capacity, $W_{cap}$ [mm], is determined by the rooting depth, $R_d[mm]$, soil field capacity, $F_{cap}$ [-], and soil wilting point, $W_{pt}$ [-]:

$$W_{cap} = R_d(F_{cap} - W_{pt}) \qquad (11)$$

WBM can take these soil and vegetation parameters – rooting depth, field capacity, and wilting point – as inputs and calculate the soil available water capacity as described in Eq. 11, or it can take available water capacity as a model input.

Water inputs to the soil are from throughfall of liquid precipitation, $P_t$ [mm/day], and snow melt, $M$ [mm/day]. Output is via actual evapotranspiration, $AET$ [mm/day], modified by a soil drying function, $g(W_s)$, and gravity drainage $D$ [mm/day]. Soil moisture balance calculations for natural landcovers are fully described in (Wisser et al., 2010a) and crop landcovers in (Grogan, 2016). Change in soil moisture is calculated each time step [day] as:

$$\frac{dW_s}{dt} = P_t + M - AET - D \qquad (12)$$

where gravity drainage $D$ [mm/day] is a function of the soil available water capacity, $W_{cap}$ [mm], actual evapotranspiration, $AET$ [mm/day], throughfall of liquid precipitation, $P_t$ [mm/day], snow melt, $M$ [mm/day], and the water depth stored in the soil moisture pool on the previous time step:

$$D = \begin{cases} \left(W_s^{k-1} + P_t dt + M dt - AET dt - W_{cap}\right)/dt & \text{if } W_{cap} < (W_s^{k-1} + P_t dt + M dt - AET dt) \\ 0 & \text{if } W_{cap} > (W_s^{k-1} + P_t dt + M dt - AET dt) \end{cases} \tag{13}$$

This gravity drainage water becomes surface runoff and/or recharge to the shallow groundwater storage pool.

*Potential and actual evapotranspiration:* Evaluation of different potential evapotranspiration (PET) functions is provided in (Vörösmarty et al., 1998); the version of WBM described here has options to use the Hamon (1963), Penman-Monteith

(Penman, 1948; Monteith, 1965), and FAO Drainage Paper No. 56 modification to Penman-Monteith (Allen et al. 1998) PET functions. The Hamon method requires only two climate inputs (temperature and precipitation), while the other two functions require additional inputs of air humidity (relative, absolute, or dew/wet bulb temperature), wind speed vectors, and cloud cover. The Hamon and Penman-Monteith functions are both described in Vörösmarty et al. (1998), and the FAO Drainage Paper No. 56 (Allen et al., 1998) modification to Penman-Monteith PET is described in the WBM model

documentation provided in the supplemental materials.

Actual evapotranspiration (AET) from naturally vegetated land areas is a function of the PET, soil moisture, and soil properties; these soil properties are field capacity, wilting point, and rooting depth (see Eq. 11 above). In a given time step, if soil moisture is sufficient to meet PET, then AET = PET. Otherwise, PET is modified by a soil drying function, $g(W_s)$. The amount of

265 water that can be drawn out of the soil moisture pool depends on the current soil moisture and the available water capacity. These functions are described fully in Wisser et al. (2010a) and Grogan (2016). Default model inputs represent the land surface as a generic, reference vegetation type (Allen et al., 1996), with soil drying parameters from Federer et al. (2003) estimated to best match global average runoff. Evapotranspiration from land cover types other than a generic reference vegetation can be represented, given input data on the sub-grid cell fraction occupied by these land cover types and a set of associated parameters.

When using sub-grid cell land cover inputs, WBM simulates a full soil water balance for each portion of the grid cell identified as a unique land cover type. Model output provides a grid-averaged value for each stock and flux. For cropland land cover inputs, sub-grid cell crop-specific water balance values can be output for soil moisture, PET, irrigation water applied (for irrigated crops), and blue water and green water use by crop (for irrigated crops). For fine resolution simulations, inputs identifying the dominant land cover type can be used to parameterize the entire grid cell, or land-cover can be used to average

necessary parameters *a priori*. Crop ET calculation methods are from Allen et al. (1998), with default parameter values for crops from Siebert and Döll (2010). While AET from other land cover types (e.g., forest or grassland) can be parameterized and simulated, no published study has yet used this option of WBM. Actual evapotranspiration from other consumptive water uses are described below in Section 2.2.5.

Open water evaporation applies to the fraction of grid cells containing terrestrial free water surfaces, including river surface area, lake and reservoir area, and inter-basin transfer canal area (see 2.3.4 below for a description of inter-basin transfer canals).

Open water evaporation rates can be input to WBM, available from reanalysis models such as MERRA2 (Gelaro et al., 2017), estimated as a multiplier on PET in the absence of an input dataset; the default multiplier in WBM is 1.0. The river surface evaporation, $E_{riv}$, is calculated as a function of open water evaporation rates, $O$, and river geometry:

$$E_{riv} = \min \left( \sqrt{A} \cdot y_R E_{ow}, \ W_R \right), \tag{14}$$

where $A$ is the grid cell area [m$^2$], $y_r$ is the stream width [m], $E_{ow}$ is the open water evaporation rate [m/day], and $W_r$ is the storage of water in the river [m$^3$]. Hydraulic geometry relations used to estimate stream width are described below in Section 2.2.3.

*Surface runoff:* When water enters a grid cell in excess of the volume that can be stored in soils, the canopy, and lost through evapotranspiration, then gravity drainage occurs resulting in both surface runoff and recharge. The distribution of this excess water between surface runoff and shallow groundwater recharge is defined by a model parameter which sets the fraction of drainage water that recharges shallow groundwater; the complement of this value is treated as surface runoff. To capture the hydrodynamic response of runoff generation following precipitation and melt events, water passes through either a surface retention pool or a shallow groundwater pool, described below. Once the runoff water leaves either of these pools, it joins with storm runoff and forms total land runoff that is then routed downstream as river flow.

Surface runoff, $R_S$ [mm/day], is retained in the surface runoff retention pool, $W_{SRP}$ [mm], prior to draining to the stream network. This temporary storage of surface runoff in the surface retention pool represents flow over the land-surface and temporary storage in ephemeral pools and wetlands. The drainage rate, $R_{SRP}$, [mm/day] from the surface runoff retention pool, $W_{SRP}$ [mm], follows a tank drainage formulation:

$$R_{SRP} = C_{SRP} \sqrt{2 \, G \, W_{SRP}} \tag{15}$$

Where $C_{SRP}$ is a unitless discharge coefficient of the surface runoff retention pool and includes unit conversions, and $G$ is gravitational acceleration.

There is an upper limit, $T_{SRP}$ [mm], imposed on the storage volume in the surface runoff retention pool. This limit captures the response of over-filled surface topographic depressions. When the volume of the surface runoff retention pool exceeds this limit, then the over-flow water, $R_{EXC}$ [mm/day], is moved to the river. This helps to capture flashy hydrodynamic responses more accurately during extreme events (Zuidema et al., 2020). Change to the storage value of the surface runoff retention pool $W_{SRP}$ is:

$$\frac{dW_{SRP}}{dt} = R_S - R_{RSP} - \delta(t - t_E)R_{Exc} \tag{16}$$

where $R_S$ [mm/day] is surface runoff, $R_{SRP}$ [mm/day] is the drainage rate out of the surface runoff retention pool, where $t_E$ are times when the surface runoff pool exceeds the limit, $\delta$ represents the Dirac delta, the integral of which over one timestep equals unity, and $R_{EXC}$ [mm/day] is the over-flow water.

The balance of the surface runoff retention pool is calculated as a split operator in three stages:

1. $W_{SRP}^1 = W_{SRP}^k + R_s dt$      (17)

2. $W_{SRP}^2 = W_{SRP}^1 - R_{SRP} dt$      (18)

where $R_{SRP} = C_{SRP}\sqrt{2\,G\,W_{SRP}^1}$      (19)

3. $W_{SRP}^{k+1} = W_{SRP}^2 - R_{Exc} dt$      (20)

where $R_{Exc} = \begin{cases} (T_{SRP} - W_{SRP}^2)/dt & if \ W_{SRP}^2 > T_{SRP} \\ 0 & if \ W_{SRP}^2 \leq T_{SRP} \end{cases}$      (21)

Where $W_{SRP}^k$ and $W_{SRP}^{k+1}$ are the storage in the surface retention storage pool at the previous and present time-step, respectively. The threshold for storage in the surface runoff retention pool ($T_{SRP}$) is set to 1000 mm by default, which effectively turns off this functionality unless an alternate value is defined. Decreasing $T_{SRP}$ to values in the range of 15 to 50 mm increases the flashy response of the model in temperate climates, enabling users to calibrate this parameter to capture regional variations in storm responses.

*Glacier runoff:* Another flux of water from the land surface to the rivers is glacier runoff. While WBM does not simulate glacier formation and dynamics beyond routing accumulated snowpack downstream as described above, it can take inputs of glacier area and runoff generated on that area. The glacier area within each grid cell is removed from the land area simulated by WBM; all water accumulation and runoff from that land area is taken from the glacier input dataset. WBM assumes that this land area, which is typically a fraction of a grid cell, sits at the highest elevation within the grid. To avoid double-counting precipitation inputs onto this land area (which is accounted for by the glacier input dataset), WBM reduces grid cell precipitation linearly by the fraction of the grid cell covered by glacier area. Each glacier has a single designated outlet location even in the case that the full glacier covers multiple grid cells, and it is also assumed that runoff from the glacier area all flows directly into the outlet grid cell's river system. These methods are first described in Mishra et al. (2020), and were developed to make use of rasterized output from the Python Glacier Evolution Model (PyGEM; Rounce et al., 2020), which provides glacier runoff at a monthly timestep. PyGEM's standard output format is not gridded; rather, post-processed PyGEM output is required as input for WBM (Prusevich, et al., 2021).

**2.2.2 Groundwater**

*Shallow groundwater storage pool:* As noted above, when water enters a grid cell in excess of the volume that can be stored in soils, the canopy, and lost via evapotranspiration, then runoff and recharge both occur. The portion of that excess water that becomes recharge is defined by the recharge fraction parameter, with a default value of 0.5. Alternative non-default input values can be a constant applied to the whole simulation domain, or a gridded layer to reflect its spatial variability. The recharge water enters a below-soil storage pool called the shallow groundwater storage pool. This shallow groundwater pool generates baseflow (i.e., subsurface runoff) by leaking water to the river system stream reaches in the same grid cell where recharge occurred. The leakage rate, $R_{SGW}$ [mm/day], is a function of a hydrodynamic groundwater constant, $\beta$ [d$^{-1}$], applied to the depth of water stored in the shallow groundwater pool, $W_{SGW}$ [mm]:

$$R_{SGW} = \beta \cdot W_{SGW} \tag{22}$$

The default value for $\beta$ is 0.025.

*Unsustainable groundwater:* Following the GHM methods of Hanasaki et al. (2008a) and Wada et al. (2012), WBM additionally represents an unsustainable groundwater source. WBM's implementation of unsustainable groundwater was first described in Wisser et al. (2010a), and again in Grogan et al. (2015; 2017), Liu et al. (2017), and Zaveri et al. (2016). Here, as in previous WBM and other GHM publications, unsustainable groundwater is defined as groundwater used in excess of the recharge stored in the shallow groundwater pool. We acknowledge that this definition does not capture the complex nature of surface water-groundwater interactions; however, this definition has been adopted by the GHM community as sufficient for macro-scale representations of the large volumes of water required to meet agricultural water uses that are clearly in excess of surface water and short-term (yearly to decadal) groundwater recharge supplies (Hanasaki et al., 2008b; Wada et al., 2012; Grogan et al., 2017; Hanasaki et al., 2018) The unsustainable groundwater source is not defined as a stock or storage pool, and so no state variable is associated with it. When the demand for water extractions (see Section 2.2.5 below) exceeds the water supply available from surface water and shallow groundwater, WBM has the option to allow the residual, or a parameter-defined fraction of the residual, to be supplied from an unlimited unsustainable groundwater source. This effectively defines unsustainable groundwater use – known alternatively as groundwater mining or the use of fossil groundwater when recharge is known to have occurred pre-historicaly (Jasechko et al. 2017)  – as any groundwater extraction in excess of the long-term recharge rates applied to the shallow groundwater pool and represents an additional source of water entering the simulated hydrologic system. Prior work (e.g., Gleeson et al., 2012; Grogan et al., 2015; Grogan et al., 2017; Wada et al., 2012; Zaveri et al., 2016) has shown that assuming that this unsustainable water source is available is reasonable at a macro-scale, and allows GHMs to evaluate aquifer mining at large scales and compare to groundwater-based mass change observations from the GRACE satellite (Sutanudjaja et al., 2018).

### 2.2.3 River discharge

WBM has a horizontal water transport model that represents the flow of rivers in one dimension. The foundation of this model is the digital river network, which defines exactly one flow direction for each grid cell.  As grid cells connect into

networks, these form the representation of river systems. Note that every grid cell has a flow direction, regardless of whether enough water accumulates to actually flow through the grid cell or not (e.g., an arid region with no or low precipitation would have no flow, but would have a defined network of flow directions, as described by STN-30p network (Vörösmarty et al., 2000b)). The model offers two options for how to calculate river flow velocity: 1) a Muskingum-Cunge solution of the Saint-Venant flow equations (Maidment, 1993), and 2) a linear reservoir routing solution. We find that the Saint-Venant flow equations are appropriate only for simulations of relatively coarse grid cell resolution – half-degree by half-degree or larger – where much of the river's volume remains within the grid cell over a 24-hour time period. For the finer resolution simulations – 5-minute grid cell size and smaller – that are now common amongst many GHMs, the linear reservoir routing method is more appropriate. The Muskingum-Cunge solution is fully documented in Wisser et al. (2010a) and Grogan (2016), and the linear reservoir routing method follows common formulations (Dingman, 2002, p. 429). Linear reservoir routing calculates reach outflow as a function of the water volume within each grid cell, with a release coefficient that is a function of celerity (rapidity of downstream motion) and reach length. Both methods are described again in the model documentation in the supplementary materials.

*Hydraulic geometry:* WBM incorporates both downstream and at-a-station stream geometry relationship assumptions to calculate river width, depth, and velocity from discharge. WBM assumes that each grid cell has a single representative stream reach and calculates a rolling average of annual mean discharge for each reach in a simulation over the previous five-years of a simulation. The long-term mean discharge, $\bar{Q}$, [m³/s] is then used to estimate the long-term mean depth, $\bar{z}$ [m], width, $\bar{y}$ [m], and velocity, $\bar{u}$ [m/s], using down-stream hydraulic geometry relations and scaling factors from Park (1977):

$$\bar{z} = \eta \bar{Q}^{\nu} \tag{23}$$

$$\bar{y} = \tau \bar{Q}^{\phi} \tag{24}$$

$$\bar{u} = \delta \bar{Q}^{\epsilon} \tag{25}$$

where $\eta$ $\nu, \tau, \phi, \delta$ and $\epsilon$ are user defined variables, with optional default values listed in Table 1.

Instantaneous estimates of the three variables ($z$ [m], $y$ [m], and $u$ [m/s] for depth, width and velocity, respectively) are given as functions of instantaneous $Q$ [m³/s] and mean discharge $\bar{Q}$ [m³/s], scaled by appropriate at-a-station hydraulic geometry exponents (Dingman, 2009):

$$z = \bar{z}\left(\frac{Q}{\bar{Q}}\right)^{f} \tag{26}$$

$$y = \bar{y}\left(\frac{Q}{\bar{Q}}\right)^{b} \tag{27}$$

$$u = \bar{u}\left(\frac{Q}{\bar{Q}}\right)^{m} \tag{28}$$

In the above equations, parameters $f, b$ and $m$ are all user defined variables, with optional default values from Leopold and Maddock (1953), listed in Table 1.

### 2.2.4 Hydro-infrastructure

*Dams and reservoirs:* Large dams and reservoirs alter river flows and provide water supplies to surrounding areas. Provided a database containing the required information, WBM simulates the impact of reservoir operations on river flow, and it uses the water stored in reservoirs as supply for water extractions and consumptive uses (see Section 2.2.5 below). The input dam database must have the following information to be of use to WBM: the year of dam construction, the reservoir area and capacity, the upstream catchment area, the main purpose, and the location. The database may optionally include information

on the year a dam was removed, if applicable. Dam databases with this information include the Global Reservoir and Dam Database (GRanD; Lehner et al., 2011), and the Hydrologically Consistent Dams Database (HydroConDams; Zuidema and Morrison, 2020).

WBM employs a general reservoir water release rule, with parameter modifications for dams of different purposes such as

irrigation supply or flood control. A general water release rule is designed to maintain outflows approximately equal to average annual inflows, but to release less water when reservoir levels are low and more water when reservoir levels are high. Water levels considered "high" or "low" are based on the purpose of the dam and can be parameterized for specific dams or set of dams. Dams on irrigation reservoirs are additionally parameterized with a time series of downstream irrigation water requirements, ensuring that water is released downstream from the dam during the time of greatest water extraction

demand. In reality, many irrigation reservoirs are connected to downstream irrigated areas by canal systems that flow directly from the reservoir and do not rely on dam operations. WBM does not represent these canal systems, and so uses dam water releases to account for this canal-enabled downstream flow of water. Full reservoir release methods, along with parameter values assigned to different dam types, are documented in Rougé et al. (2021). Alternatively, discharge out of individual dams can be input directly to WBM, thereby making calculated reservoir storage a function of observed reservoir

output (Zuidema et al. 2020); this ensures releases match historical records in cases where WBM's default functions vary too far from observed reservoir operations.

Reservoirs with a storage capacity below a given threshold (default is 1 km$^3$) are treated as unmanaged spillway dams with the spill gate geometry determined from the stream geometry for the average annual flow. Water release from these structures are calculated from the hydraulics formulation for those dam structures given in the US Army Corps Engineers

hand book (US Bureau of Reclamation, 1987; US Army Corps of Engineers, 1987). Natural lakes are treated in a similar way as the spillway dams, but the gate geometry is determined as those from instantaneous riverbed geometry described above.

*Small irrigation reservoirs:* Rainwater harvesting for irrigation water supply is represented in WBM's small irrigation

reservoir module. These small reservoirs do not dam rivers as larger reservoirs do, and so do not alter river flow. Rather, they collect rainwater and surface runoff, storing it on the land surface and preventing it from reaching the rivers system. Note, these are not run-of-river reservoirs, but structures on the land surface. We do not know of any global or even regional

dataset that describes the location and capacity of these small irrigation reservoirs. WBM's small irrigation reservoir methods were developed and first described in Wisser et al. (2010b), where a range of capacities were simulated to provide a sensitivity analysis and quantify the potential importance of these highly localized water supply systems.

*Inter-basin transfers:* Inter-basins transfers are large canals, tunnels, or pipelines that move water across river basin boundaries. These large projects alter flows in both the sending and receiving river systems and can be used to supply water for consumptive uses. WBM simulates how inter-basin transfers alter the flows in both the sending and receiving rivers, though it does not explicitly represent the routing of water discharge through the canal system. WBM's inter-basin transfer methods were first developed and described in Zaveri et al. (2016) and described again in Liu et al. (2017). Five parameters are used to simulate the water transfer: the water sending point latitude and longitude, the water recipient latitude and longitude, a minimum allowed sending river flow, a maximum allowed canal intake flow, and a water release rule for flow volumes between the minimum and maximum. A database of India's inter-basin transfers was used by WBM in Zaveri et al. (2016) and is included as supplemental information to that publication.

### 2.2.5 Water extraction and consumptive water use

Water extractions from rivers, reservoirs, and groundwater are an important part of simulating water supply and changes in human-hydrologic interactions. WBM first implemented water extractions for irrigated agriculture (Wisser et al., 2008; Wisser et al., 2010a), which globally is known to account for ~70% of all freshwater extractions (Rosegrant and Cai, 2002). Modules for water supply for livestock, domestic, and industrial use, which are less consumptive than irrigation water and account for a smaller proportion of total global extractions, were added to WBM in Liu et al. (2017). When water is removed from a storage (e.g., reservoirs or the shallow groundwater pool), the storage value of that stock is updated within the daily time step.

Water withdrawals are taken from different water stocks and fluxes based on a given priority order of both water users and water sources; this rule set has a number of user input options and parameters making it highly flexible and customizable. The default priority order for withdrawal by water users within a grid cell is: (1) domestic, (2) industrial, (3) livestock, and (4) irrigation. In turn, the withdrawals from each user group come from water storage and flux pools in the following order until the requested withdrawal water volume is met:

1) Small irrigation reservoirs (available to livestock and irrigation water use only);
2) Shallow groundwater. When shallow groundwater is extracted for domestic, industrial, and livestock use, all the water in the shallow groundwater pool can be extracted, up to the volume requested by the sector. When this source is extracted for irrigation, an optional parameter, $r_{sg}$ [-], defines the target ratio of groundwater-to-total withdrawals for irrigation water extractions. This parameter can be a constant or a spatially variable grid. If $r_{sg}$ is not defined, then all available shallow groundwater is extracted for use (up to the water demand) in this step. In the case that $r_{sg}$ is defined,

then this first groundwater withdrawal step takes water from shallow groundwater, $SGW$, up to the volume defined by the product of $r_{sgw}$ and irrigation water demand, even if there is more shallow groundwater available and the irrigation water demand, $D$ [mm], is greater than the defined water amount, such that shallow groundwater withdrawal, $W_{sgw}$, is at this step is:

$$W_{sgw} = \min{(r_{sgw}D, SGW)} \tag{29}$$

3) Surface water in a river or reservoir within the same grid cell. Stream water available for extraction, $S_e$, is the sum of water retained in river and reservoir storage at the end of the previous time step, $W^{k-1}$, and the volume of water flowing through the reach during the previous time-step, $Q^{k-1}$, limited by a scaling factor that is set to the default value of 0.8:

$$S_e = 0.8\,(W^{k-1} + Q^{k-1}) \tag{30}$$

The scaling factor 0.8 prevents river reaches from being completely dried out by water extractions.

4) Shallow groundwater – second extraction for irrigation only. If the parameter $r_{sgw}$ is defined in a way that limited shallow groundwater extraction for irrigation to less than the available shallow groundwater volume in Step 2, and there is still residual water demand, then at Step 4 water volumes up to the remainder of the shallow groundwater storage volume can be extracted. By combining Steps 2 and 4, the target irrigation groundwater-to-total withdrawal ratio is achieved only in the case that the sum of surface and shallow groundwater volumes is sufficient to meet this ratio; Step 4 ensures that fulfilling water withdrawal demands using sustainable resources within the grid cell takes priority over achieving the target ratio. This step does not apply to livestock, domestic, or industrial water extractions, as no ratio parameter is applied to those water uses.

5) Surface water in a river or reservoir outside the given grid cell that has the largest storage + discharge volume within a set of parameter-defined radii. A different parameter can be set for irrigation water use than for other uses, representing the differences in irrigation and municipal water supply infrastructure. The default radius value is 100 km for all water uses; the user can define a set of alternative constant scalars or gridded layers of values.

6) Unsustainable groundwater (UGW). Water available for extraction from this pool may be limited by the UGW allowance ratio, if defined. Because this source of water has no stock value, the allowance ratio applies a scaling factor of $\leq 1$ to the water withdrawal demand. This scaling factor is independent of $r_{sgw}$, and if not defined, this pool is unlimited.

*Irrigation:* Given inputs of irrigated land area and associated crop-specific parameters, WBM calculates the agronomic water requirements for optimal crop growth over its three growing seasons: (1) planting and development, (2) growth, and (3) harvesting. In WBM, crops extract water from the soil moisture pool each day of the crop's growing season. Given sufficient water in the soil moisture pool, the amount of water used by each crop is the crop's potential evapotranspiration. When soil moisture levels drop below a crop-specific threshold, the difference between the soil moisture level and field capacity is defined as the irrigation water requirement. This method of crop irrigation water requirements follows FAO

guidance (Allen et al., 1998), as is typical of GHMs. WBM's crop irrigation water requirement methods have been described in: Grogan et al. (2015, 2017), Liu et al. (2017), Wisser et al. (2010a), Zaveri et al. (2016) and Zuidema et al. (2020).

    Alternatively, WBM has the option to calculate a daily crop gross irrigation water requirement instead of using the crop-specific soil moisture threshold to trigger water extractions. This option is useful for simulations with large grid cell sizes,
where the calculation of average soil moisture over large irrigated areas leads to unrealistically high irrigation water demands in a single day. When using this option, WBM estimates gross crop irrigation water requirements each day, equal to the difference between soil moisture content and field capacity and modified by either the classical irrigation efficiency parameter or the irrigation technology-derived classical efficiency for the day (described below). Irrigation water is then extracted from water sources each day, and stored in an irrigation water storage pool that doesn't interact with other fluxes
within the model until the day when the crop-specific soil moisture threshold is reached. When this threshold is reached, water is moved from the irrigation water storage pool to soil moisture. This option extracts relatively small amounts of water from water stocks each day, instead of larger amounts of water on the day that the soil moisture threshold is reached. These smaller, daily extractions may better simulate the temporal distribution of irrigation activity over large grid cell areas.

The amount of water required by a crop to achieve AET = PET is less than the amount of water that must be extracted from a water source due to inefficiencies in irrigation water extraction, transportation, and application. WBM has two options for calculating the gross irrigation water extraction required as a function of net irrigation water required by the crop: (1) the irrigation efficiency method, and (2) the irrigation technology method. In both cases, water extracted in excess of net irrigation water requirements are returned to surface and groundwater systems on the same day as extraction. Returns to the
surface water system are treated as surface runoff (see above description of surface runoff), and are added to the surface runoff storage pool. Returns to the shallow groundwater system are treated as shallow groundwater recharge (see above).

    The irrigation efficiency method is standard for GHMs and described in Grogan et al. (2015, 2017), Liu et al. (2017), Wisser et al. (2010a), and Zaveri et al. (2016). In this method, classical irrigation efficiency is an input to WBM and directly
modifies the net irrigation water requirement by a spatially varying constant. Classical irrigation efficiency is defined as the ratio between net irrigation water required and gross water extractions. Net irrigation water requirements include water transpired by the crops, and associated soil evaporation that is unavoidable. As described in in Grogan (2016) and Wisser et al. (2010a), net irrigation water requirements for rice paddies also include an additional water volume, representing the water needed to enable flooding at the start of the growing season and maintenance of the flood paddy water level throughout the
season to compensate for percolation. The volume of water added to initially flood the rice paddies is an input parameter with a default value of 50 mm of depth applied over all irrigated paddy rice areas. The daily additional water application rate used to maintain the paddy depth is based on the rate of water percolation through the underlying soils. This is also an input dataset, with methods for calculating percolation rates from soil property data described in Wisser et al. (2010a). Both the

initial paddy flood water and the daily maintenance water are included in irrigated rice's net irrigation water volume, and the irrigation efficiency parameter is applied to these volumes in the same way it is applied to other net irrigation water requirements.

The irrigation technology method in WBM is first described in Zuidema et al. (2020), and represents non-consumptive irrigation water losses as a function of irrigation technology-specific parameters and open water evaporation rates (which can be input or calculated as a function of weather inputs). In this second method, inputs on the spatial distribution of different irrigation water conveyance and application technologies (Jägermeyr et al., 2015, 2016) is required, and the inefficient water losses that occur over space and time are calculated within WBM as a function of irrigation technology type and weather variables. Classical irrigation efficiency is therefore calculated and provided as a time- and space-varying model output.

*Blue water and green water use for irrigation:* Falkenmark and Rockström (2006) introduced the concept of "blue water" and "green water" into the GHM literature to distinguish between direct precipitation and irrigation water sources in crop AET. Blue water is defined as liquid water that can be extracted from aquifers, surface water reservoirs (lakes and dams), and river systems, and green water is defined as soil moisture water originating from direct precipitation (including snowmelt) (Falkenmark and Rockström, 2006). WBM can estimate the flux of blue and green water via evapotranspiration by irrigated crops. Note that all evapotranspiration from rainfed crops is by definition green water. All water that becomes irrigated crop evapotranspiration must first enter the soil moisture pool. Water enters the soil moisture pool by either (1) direct precipitation or snow melt, which is green water, or (2) irrigation from surface or groundwater, which is blue water. We assume that water in the soil moisture pool is well mixed on a daily time step. Therefore the evapotranspiration out of that pool has the same proportions of blue and green water as the soil moisture pool itself. Optional model output variables include the grid cell average soil moisture that is made up of blue and green water [mm], grid cell total evapotranspiration of blue and green water from the soil storage pool [mm/day], crop-area specific soil moisture values of blue and green water [mm] (e.g., blue water stored in soils under a specified input crop type), and crop-specific evapotranspiration of blue and green water [mm/day].

*Livestock:* Livestock require water for drinking and for service water, which includes washing and cooling. WBM uses the methods and default parameter values (Table 2) provided by Steinfeld et al. (2006) to calculate livestock water use by animal type. Daily livestock water, $L_w$ [m$^3$/d], for each livestock type is calculated each day as:

$$L_w = (I_l + s_l T + B_l)D_l \tag{31}$$

where $I_l$ [m$^3$/head/day] is the minimum water demand for livestock type $l$, $s_l$ [m$^3$/head/°C/ day] is the temperature induced consumption requirement for livestock type $l$ [-], $T$ is the daily mean air temperature, with a minimum value of 0 [°C]; $B_l$ [m$^3$/head/ day] is the daily service water volume required per animal, and $D_l$ is the density of livestock type $l$ in the grid cell [animal head/grid cell]. Additionally, an animal population growth rate can be applied to each livestock head density

category to represent increases in population over a given single-year value of animal head density data (the year of $D_l$, input reference livestock density). This is useful as limited global livestock density data is available. Livestock is assumed to consume 5% of its water extractions, with the remaining 95% returning to the system via runoff; the ratio of consumption to return flows can be modified by user-defined input parameters.

*Domestic and industrial:* Households and industry extract water for a range of purposes, and at rates that have great spatial variability. WBM represents these extractions based entirely on an input per capita water extraction rate and a population density map, such that domestic water use, $U_d$ [m³/grid cell/day], is:

$$U_d = u_{dom} A D_{pop} \qquad (32)$$

And industrial water use, $U_i$ [m³/grid cell/day] is:

$$U_i = u_{ind} A D_{pop} \qquad (33)$$

Where $A$ [km²] is the area of the grid cell, $u_{dom}$ [m³/person/day] is the per capita domestic water withdrawal, $u_{ind}$ [m³/person/day] is the per capita industrial water use, and $D_{pop}$ [persons/km²] is the population density. Domestic and industrial water use each have unique return fraction coefficients, which default to uniform values of 84% and 89% respectively.

### 2.2.6 In-stream nitrogen and water temperature

*Nitrate-nitrogen concentration:* WBM estimates in-stream and in-reservoir nitrate-nitrogen (N-NO₃), dissolved organic nitrogen (DIN), and/or total nitrogen (TN) concentration. In-stream N-NO₃ concentrations are a function of point source nitrate inputs from wastewater treatment plants, non-point source nitrate inputs from the land surface, and in-stream denitrification. Wastewater treatment plant contributions to in-stream nitrate are calculated using data on served population and waste treatment type, as described in Samal et al. (2017). Nitrate inputs from land are estimated as a function of simulated grid cell runoff and the estimated nitrate concentration in runoff from different land use types. Estimation of land use-specific runoff nitrate concentrations are described in Wollheim et al. (2008a). The suite of parameters describing nitrate concentration in runoff from different land use types may require region-specific calibration, based on high spatial resolution nitrate sampling from headwater catchments along a gradient of human land use and flow conditions (Wollheim et al. 2008a). The model default values are found to be adequate for moderately developed landscapes with modest agricultural cover in the northeastern United States (Samal et al., 2017; Simon, 2018; Stewart et al., 2011a). In-stream (Stewart et al., 2011a) and in-reservoir (Simon, 2018) denitrification are calculated using temperature-corrected denitrification along the benthic surface assuming efficiency loss kinetics, following Mulholland et al. (2008) and Wollheim et al. (2014).

*Water temperature:* River temperature is calculated following Stewart et al. (2013) with an addition to account for air humidity and canopy shading (see documentation in the Supplemental Materials for details). Temperature is first calculated on the landscape, mixing air temperatures depending on the timing of shallow groundwater recharge. River temperature

requilibration is then calculated through a combined empirical and deterministic re-equilibration procedure given by Dingman (1972). The reequilibration is a function of channel hydraulics, air temperature, solar radiation, humidity, and wind speed.

### 2.2.7 Water source tracking

WBM tracks water from each source (water inputs to each individual grid cell) through all flows and stocks within the model. Stocks within each grid cell include soil moisture, small reservoir storage, shallow groundwater storage, surface retention and irrigation storage pools, rice paddy flood waters, river storage, and large reservoirs. Flows are infiltration into soils, surface runoff, recharge to shallow groundwater, baseflow, river discharge, water discharge from reservoirs, evaporation, evapotranspiration, inter-basin transfers, water extracted for human water use, and return flows from human water use. These stocks and flows are depicted in Figure 2. WBM's tracking functionality retains information about the generative mechanism (i.e., the water source) as water flows across the landscape through the river network. This includes through processes such as extraction for human use, and subsequent redistribution according to hydrologic flow-paths.

The same tracking algorithm applies to all water source components. For any water component $c$ in water storage stock $S$ at timestep $k$ in a given grid cell:

$$S_c^k = \frac{(S_c^{k-1} \cdot S^{k-1}) + \Sigma_i (I_{c,i} \cdot I_i) - \Sigma_i (S_c^k O_j)}{S^k} \tag{34}$$

where $S_c^k$ is the fraction of stock $S$ composed of component $c$ at time $k$. $S^k$ is the total volume of stock $S$ at time $k$. $I_i$ are inflows to and $O_j$ are outflows from stock $S$, with $I_{c,i}$ the fractions of the $i$th flow composed of component $c$ all at timestep $k$. Component stocks ($S_c^k$) are updated throughout the timestep such that the solution is split into multiple operators as the various fluxes impact each stock.

WBM performs three types of component tracking: primary source component tracking (Fig. 2) representing the initial input of water into the water balance equations, return flow component tracking representing water that has been reintroduced to the hydrologic cycle following human extraction, and runoff from labeled land attributes (Table 3). The model user can choose any combination of sources to track simultaneously, as the tracking modules are independent and each can be turned on or off in a given model simulation. A user interested in understanding the role of snowmelt as a component of streamflow downstream of a mountainous region would use primary source component tracking, whereas a user interested in understanding the potential for anthropogenic contaminants to be present in streamflow would use return flow component tracking. If a user was interested in runoff generated within any political boundary, land attribute tracking could be used. The intersection of different tracking components is not calculated; by turning on both primary source and return flow component tracking, for example, WBM will not calculate the fraction of irrigation return flow composed of snowmelt. Primary source components were first described in Grogan et al. (2017), where only the unsustainable groundwater component was

analyzed. Return flow components were first described in Zuidema et al. (2020); land cover mask components are described here.

All stocks and flows are considered well-mixed, so that the flows out of a stock have the same fractional water source components as the stock itself. All stocks are initialized with $S_c = 1$ for one of the set of components that are tracked. For
example, in primary source component tracking, all stocks are initialized as 100% rain water; as the model goes through a spin up stage, water from the other components are added to these stocks. At the beginning of a simulation, large reservoirs are initialized at 80% of their full capacity, the soil moisture storage pool is initialized at 50% of capacity, and all other stocks begin at 0% capacity. We recommend a minimum spinup time of 10 years to allow all stocks to reach equilibrium storage, and importantly for many stocks to accumulate the different tracked water components. WBM operates at a daily
time-step, and for some stocks (e.g. river discharge) our well-mixed assumption is appropriate; however, other stocks are typically not well-mixed at the daily time-scale; for example reservoirs (Håkanson, 2005) and groundwater (Hrachowitz et al., 2013) are known to mix at longer time-scales. Therefore, we consider these fluxes with caution at short time-scales (days to years), but find them informative when averaged over long-periods (years to decades).

Return flow tracking has an additional option for re-setting the stock component values after spinup has completed. At the end of spinup (prior to the simulation period), stocks can be reset to 100% relict water. Relict water is defined as any water stored in simulated water stocks at the end of spinup, and makes no assumptions about the source, age, or use condition of the water. This option allows the user to interpret changes to stock components that occur only within the simulation period, removing assumptions about starting compositions. New water entering the system during the
simulation period as precipitation or glacier runoff is tagged as "pristine" water. This option is one way to explicitly track the fate of components that enter the simulation at the onset of the representative simulation period (Zuidema et al., 2020).

Note that the land surface label tracking can track multiple land labels at once that can include sets of political
boundaries, land-cover types, soil types, biogeographic or climate zones, or other identifiers such as the grid cell Strahler stream order or distance of a grid cell from the river mouth. These land labels can occupy entire grid cells, or be provided as a set of grid cell fractional coverage (i.e., a percentage of each grid cell is covered by each label type). WBM will track each identified land label with a unique numerical ID input via a raster-based mask of unique values.

**3 Model evaluation**

River discharge is the observational data against which most GHMs validate, in part due to the abundance of high quality global river discharge data and in part due to the fact that river flow is an integrative result of all the land surface fluxes

simulated by GHMs. Here we first summarize published validation of WBM output in recent relevant papers (section 3.1).

We make note of where these evaluations make use of prior code branches (e.g., the C++ version of WBM, or FrAMES) or are regionally-specific. Then we present an evaluation of global river discharge simulated by the open source WBM v.1.0.0 version described here (section 3.2). We also evaluate the model's estimation of water extraction for irrigation against the only global dataset avilailabe for this metric.

**3.1 Published WBM validation and evaluation**

This section reviews the literature of WBM publications that include validation and/or evaluation of model components that are included in the WBM open source model. These papers report a variety of different evaluation metrics, which we summarize here.

*Global river discharge:* The Perl/PDL version of WBM (which is described here) was most recently evaluated against global discharge from the Global Runoff Data Centre reference dataset (GRDC, 2015) in Grogan (2016). Grogan (2016) reports that a linear regression of modeled versus observed average annual river discharge for the years 1980 – 2009 typically shows strong agreement ($r^2$ values between 0.74 and 0.87), but that this agreement varies with the choice of input climate data set. In comparing four different climate input datasets, UDEL (Willmott and Matsuura, 2001) and NCEP (Saha et al., 2014)

climate inputs were found to provide the best global discharge simulations, with over 40% of all GRDC stations achieving a Nash-Sutcliffe Efficiency (NSE; Nash and Sutcliffe, 1970), a typical hydrologic evaluation metric, of $> 0$, meaning that the model predicted observations better than the mean of historical observations. There is also spatial variation in model performance; as can be seen in Grogan (2016), WBM river discharge matches observations best in temperate and tropical regions, but performs poorly in arid climates. Spatial variation in validation metrics is also in part due to the choice of

climate inputs. Overall WBM simulations from Grogan (2016) are biased low compared to observations. These results are consistent with global river discharge evaluation of the C++ version of WBM (also called WBMplus) in Wisser et al. (2010a), who report an average Model Bias Error (MBE) of runoff of -1.2 mm month$^{-1}$ from 1901 – 2002. Fekete et al. (2002) also compared WBM (C++ version) global river discharge to GRDC data, and reports a positive mean bias in runoff of 7.9 mm yr$^{-1}$. All three published global river discharge evaluations show that simulated discharge performs better in larger

catchments than in smaller ones. All three simulations used a 0.5 degree grid cell resolution; we refer readers to the publications themselves for descriptions of parameter value choices, as the level of calibration and the setting of default parameters varies depending on the study.

*Regional river discharge:* WBM can be used for sub-continental scale, or regional, studies; in this case, a finer spatial

resolution must be used, and model parameters can be calibrated to better fit local conditions, and regional river discharge data is used for evaluation. Grogan (2016) and Zaveri et al. (2016) evaluated WBM against river discharge data in India, using discharge and runoff data from the India Water Resources Information System (India-WRIS) and FAO AQUASTAT

(Frenken, 2012), respectively. They report that the Nash-Sutcliffe Efficiency (NSE; Nash and Sutcliffe, 1970) is > 0 for 15 of the 20 IWRIS sites, and average annual runoff from WBM compares well with AQUASTAT reports for the 8 largest river basins in India. These continental-scale simulations of India used the same 0.5 degree spatial resolution as the global simulations, along with a regional climate driver (APHRODITE; Yatagai et al., 2012). A finer, ~100 km$^2$ (6 arc-minute) grid cell resolution simulation of northeastern North America had NSE values > 0 at 82% of the 791 USGS gage stations used for comparison in Grogan et al. (2020). A very fine resolution, 1 km grid cell scale simulation of the Trishuli Basin in Nepal is evaluated in Mishra et al. (2020), where overall agreement with reported monthly mean river discharge is shown (NSE > 0.7), though seasonal variation shows that WBM underestimates summer high flows in some years, and in other years over-estimates high flows over a period of 11 years. A similarly fine-resolution simulation (~ 1km) of the Upper Snake River Basin of Idaho, U.S., is evaluated in Zuidema et al. (2020); seasonal discharge in headwaters compares well (NSE = 0.9) with USGS gage data, though WBM demonstrates a positive bias (discharge values are too high) and large variation in seasonal discharge in the basin's small tributaries. All fine resolution WBM simulations described here used non-default parameter sets that were calibrated to regional data, unlike the global runs described above. Even with regional calibrations the simulations result in similar outcomes as the global analyses: WBM river discharge typically compares well to observations, though better in larger than smaller river basins, and better when aggregated to a monthly time step rather than daily. Default parameters provide good performance at large (continental to global) scales, but calibration is required for local to regional studies to account for local deviations of parameters from the global means. Additionally, simulated river discharge disagrees with observations immediately downstream of dams that either aren't represented in the input dam database, or that are operated with decision rules not captured by WBM's reservoir operation algorithms, as described in Rougé et al. (2021).

*Irrigation water extractions:* WBM is often used for agricultural applications, and so has been well validated against FAO country-level reported irrigation water extraction data globally in Grogan (2016), Grogan et al. (2017), Wisser et al. (2008), Wisser et al. (2010a) (Fig. 3), and regionally in Zaveri et al. (2016) and Zuidema et al. (2020). Notably, Wisser et al. (2008) quantifies the high uncertainty in irrigation water withdrawals as a function of input climate and crop map data. Globally, WBM-simulated average total irrigation water extraction for the years 1963 – 2002 varies from 2,200 km$^3$year$^{-1}$ to 3,800 km$^3$year$^{-1}$ in Wisser et al. (2008), with the large difference in values due entirely to the choice of climate input and crop map. While the evaluation data used in all the WBM publications is fully independent of model input data, it should be noted that most irrigation water extraction data are reported statistics, not direct observations.

*Additional validation and evaluation metrics:* In addition to river discharge and irrigation water extractions, regional studies have evaluated against metrics that are relevant to their application. For example, Zaveri et al. (2016) qualitatively evaluated WBM's change in groundwater levels in the Indian state of Punjab using well level data; Grogan et al. (2020) evaluated simulated snow water equivalent across northeastern North America, and Zuidema et al. (2020) evaluated WBM's snowmelt

onset timing. The swow water equivalent evaluation found that for most study sites in the northeastern United States, goodness-of-fit ($r^2$) values were > 0.5, but model performance was poor where winter precipitation is dominated by lake-effect snow, and where climate is moderated by the coastal warming effect (Grogan et al., 2020). Zuidema et al. (2020) found that the timing of peak runoff generation due to snowmelt was well-captured by WBM, but that the onset of snowmelt had an early bias in most years.

*Validation of FrAMES:* The FrAMES model (Wollheim et al., 2008a,b; Stewart et al., 2013) functions for river temperature and in-stream nitrogen concentrations have been incorporated into the open source version of WBM described here. Despite having a different name, FrAMES is part of the WBM model family, as it added modules for in-stream processes to the WBMplus model code base (see section Model Code below). While there has yet to be a published evaluation of the open source WBM implementation of these functions, the FrAMES model nitrogen functionality is evaluated globally in Wollheim et al. (2008a) and regionally in Samal et al. (2017) and Stewart et al. (2011b). River temperature simulations are evaluated across northeastern North America in Stewart et al. (2013). FrAMES also has an in-stream chloride module; while WBM does not yet have this module implemented, chloride is an informative metric for evaluating river discharge as this solute is a conservative tracer. We report FrAMES chloride validation findings here to show how well discharge matches observations, as the river discharge functions in WBM and FrAMES are the same. In Zuidema et al. (2018), simulations of river discharge, temperature, and chloride in the Merrimack and Piscataqua River watersheds of New England, U.S., were assessed using approximate Bayesian computation (Sadegh and Vrugt, 2013), which provides information on the best regional parameterization for the model. The best parameter estimates resulted in simulated flow-duration curves with an NSE of 0.93 compared to USGS gage data. Further, Zuidema et al. (2018) found that default WBM parameters for the hydrodynamic groundwater constant and $C_{SRP}$, while slightly different from the best performing parameters, still resulted in good agreement with observations.

**3.2 Open Source WBM Model evaluation**

Above, we reviewed previously-published WBM validations. As none of the prior versions of WBM code have been released open source, it is important to validate the exact model structure in this first open source release. Previous versions of WBM and related model code (Table 7) all used the same underlying structure as WBM v.1.0.0 with regards to all the basic terrestrial water balance variables: evapotranspiration, soil moisture balance, surface runoff generation, subsurface runoff (aka baseflow), shallow groundwater recharge, and river routing. The most recent WBM publications (since 2016) have included the same agricultural water use module as WBM v.1.0.0; tracking as described here was first implemented in Grogan et al. (2017). Differences between prior publications and WBM v.1.0.0 as described here are mainly in parameter values (here we use all default values for a general model demonstration) and in some cases recent publications have implemented additional region-specific modules not included in WBM v.1.0.0 (e.g., Zuidema et al., 2020).

In this section, we evaluate results from a global open source WBM simulation that uses publically available data inputs, and provides a comprehensive selection of tracking outputs. The simulation ran for 270 hours on a Dell PowerEdge R510 with Intel Xeon processors (2.93 Gbps) and simulated 2.3M grid cells for 10 years following 10 years of spinup.

### 3.2.1 Model setup

Here we use a global 5-minute spatial resolution WBM simulation for evaluation. WBM is first initiated with a 10 year spinup to bring stocks to an equilibrium state. Results shown below are from simulated years 2000 – 2009. All model input datasets are listed in Table 4. All parameters are set to default values. The model initialization file used for this simulation is available from Grogan et al. (2022b).

### 3.2.2 Evaluation data and methods

*River discharge:* We evaluate WBM using default parameter values (Table 1) against daily and monthly river discharge records from the Global Runoff Data Centre (GRDC, 2020) which we downloaded in February of 2020. The GRDC user terms of agreement prohibits sharing of this data, but the same data can be requested directly from the GRDC. We also compare WBM global annual river discharge results to other published global estimates (Table 5).

GRDC stations were filtered based on three criteria. The first criteria is that the station must have data within the simulation time frame of years 2000 – 2009. The second criteria is that within the time frame, the station must have at least 12 total observations for monthly evaluation, or at least 365 total observations for daily evaluation. The third criteria compares the GRDC-reported catchment area of a station to the catchment area of the best-matching MERIT river network grid cell within a 3-by-3 grid centered on the latitude/longitude point defined by the GRDC station. Only GRDC stations with catchment area differences of less than 10%, once the best area match within the 3x3 grid is identified, are included. Applying these criteria leaves 322 stations for daily and 344 stations for monthly evaluation.

We evaluate simulated daily and monthly average discharge with the Index of Agreement, $d$, (Willmott, 1981):

$$d = 1 - \frac{\sum_{i=1}^{n}(O_i - P_i)^2}{\sum_{i=1}^{n}(|P_i - \bar{O}| + |O_i - \bar{O}|)^2} \tag{35}$$

Where $P_i$ are predicted (i.e., simulated) discharge values, $O_i$ are observed discharge values, and $n$ is the number of observations. The value of $d$ can range from 0 for a model that is not a better predictor than the mean observed value, to 1.0 for a perfect match of predictions to observations.

In order to measure systemic bias, we also calculate the Mean Bias Error (MBE):

$$MBE = \frac{1}{n}\sum_{i=1}^{n} P_i - O_i \tag{36}$$

We additionally calculate the Nash-Sutcliffe Efficiency (Nash and Sutcliffe, 1970) and the Kling-Gupta Efficiency (Gupta et
al., 2009) metrics, which are both classic skill scores that indicate if model skill is better than predicting the mean of the
observations (value > 0 indicates better skill).

*Irrigation water withdrawals:* We compare WBM-simulated irrigation water withdrawals by source to reported country-
level water withdrawal statistics from AQUASTAT (FAO, 2016), as well as other model-based estimates of withdrawal in
the literature.

### 3.2.3 Results

*Daily river discharge:* Overall, global daily average discharge is simulated with moderate agreement to observations; the
average index of agreement over all stations is 0.56, and average Mean Bias Error (MBE) is -0.07 mm day$^{-1}$ (Fig. 4a,e). For
the Nash-Sutcliffe Efficiency (Fig. 4c) and Kling-Gupta Efficiency (Fig. 4d) metrics, we find that 43% and 54% of basins
have values greater than 0, respectively. However, there is substantial spatial variation in these metrics, with the mean highly
influenced by the relatively large number of GRDC stations in the Americas compared to other continents. The lowest single
river discharge MBE value is -5.5 mm day$^{-1}$, which occurs in Southeast Asia (Fig. 4b).

*Monthly river discharge:* Overall, global monthly average discharge is simulated with good agreement to observations; the
average index of agreement over all stations is 0.69, and average Mean Bias Error (MBE) is -0.14 mm month$^{-1}$ (Fig. 5a). For
the Nash-Sutcliffe Efficiency (Fig. 5c) and Kling-Gupta Efficiency (Fig. 5d) metrics, we find that 48% and 58% of basins
have values greater than 0, respectively. These results are consistent with Wisser et al. (2010a), even though different climate
inputs and simulation time series were used.

Despite the global average good agreement, there is significant spatial variability, with lower MBE values across much of
South America and East Asia (Figs. 4e and 5e). There are also notable large regions without any evaluation data that meet
the criteria for inclusion in this analysis, including South Asia, Northern Africa, and the Middle East. Further, MBE values in
arid regions will always appear small due to the very low values in river discharge; relative bias metrics are better evaluation
tools for these regions.

*Annual discharge comparison:* Many prior studies have estimated global river discharge (Table 5), providing a range of
values from as low as 29,485 km$^3$ year$^{-1}$ (Oki et al., 2001) to as high as 47,884 km$^3$ year$^{-1}$ (Liang and Green, 2020). Variation
between estimates is caused by several factors, including but not limited to: model structure, model calibration, input data,
years simulated, simulation domain (e.g., some studies exclude Greenland and/or Antarctica), and inclusion of anthropogenic
impacts. See Schmied et al. (2014) for a review and analysis of GHM global discharge sensitivity and calibration. Global
annual river discharge as simulated by WBM v.1.0.0 for the years 2000-2009 is 42,957 km$^3$ year$^{-1}$, which is within the range
of prior studies (Table 5). Fewer studies report the contribution of exorheic (basins that discharge to the ocean) and

endorheic (basins that discharge to internal seas) discharge. WBM v.1.0.0 estimates exorheic discharge is 40,248 km$^3$ year$^{-1}$, and endorheic discharge is 2,709 km$^3$ year$^{-1}$. The exorheic estimate falls within published values, which range from 38,314

850   km$^3$ year$^{-1}$ - 46,221 km$^3$ year$^{-1}$. WBM v.1.0.0 estimates higher endorheic basin river discharge than previous studies, which report estimates of 993 km$^3$ year$^{-1}$ – 1,603 km$^3$ year$^{-1}$. It is possible that this higher estimate reflects WBM v.1.0.0's inclusion of Greenland, along with inclusion of runoff from glaciers, which is not present in most previous studies.

*Irrigation water withdrawals:* Simulated irrigation water withdrawals fall on the high end of previously-reported GHM-simulated global irrigation water use (Table 6). Note that Wisser et al. (2008) demonstrated a large uncertainty in GHM-simulated global irrigation water withdrawals as a function of input climate and crop map data. WBM simulations match well to AQUASTAT (FAO, 2016) country-level statistics on agricultural water use (Fig. 6) for most countries, with an R$^2$ value of 0.84 on a linear regression of country-year combinations included in both the AQUASTAT database and WBM

simulations. However, WBM simulates 2 to 3 times higher irrigation water use in China and Pakistan than reported by AQUASTAT, accounting for most (up to 90%) of the difference between WBM and the mean of other GHM-simulated global agricultural water withdrawals (Table 6). As can be seen in Figures 4 and 5, river discharge across much of Asia is under-estimated by this WBM simulation; which may reflect a low bias in precipitation inputs thereby contributing to the over-estimation of irrigation water withdrawals in China and across much of Asia (Fig. 6). Data from Grogan (2016) shows

that irrigation water requirements can be highly sensitive to climate inputs, especially in Asia; comparing six different climate inputs to WBM, results for irrigation water requirements in China vary from 615 km$^3$ year$^{-1}$ (driven by the NCEP (Saha et al., 2014) climate data) to 1,276 km$^3$ year$^{-1}$ (driven by the UDEL (Willmott and Matsuura, 2001) climate data).

**4 Water source tracking module demonstration**

WBM's unique water source tracking functions distinguish it from other GHMs. Here, we demonstrate the suite of tracking options available to model users: primary source tracking (4.1), return flow tracking (4.2), and land surface label tracking (4.3). Tracking output explains how the model arrives at simulated water stocks and flows. For example, river discharge is a collection of water flowing from different sources. These tracking functions make explicit what the sources are within the

model that form the simulated discharge. We caution that any model can arrive at a well-validated result through erroneous assumptions and aggregate errors. We find the component tracking increases the transparency of model assumptions; however, evaluation data for these tracking functions is not available at this time, and we rely on evaluation of the stocks and fluxes themselves (not the component composition) for model evaluation. Future regional scale work could make use of emerging datasets on DNA or geochemistry such as chloride (Zuidema et al., 2018) to evaluate return flows from human and

agricultural uses (Plummer et al., 2000), and stable water isotopic methods may be able to distinguish rain, snow melt, and glacier water sources ( Fekete et al., 2006; Fan et al., 2016; St Amour et al., 2005).

Here we use the same global, 5-minute spatial resolution WBM simulation as used for model evaluation to demonstrate the first two tracking examples: primary source tracking and the return flow tracking, as multiple tracking functions can be implemented within a single model run.

## 4.1 Primary source tracking

The primary source tracking function identifies all water entering the model system as originating from one of four categories: rain, snow, glacier runoff, or unsustainable groundwater. Note that the shallow groundwater pool is filled with water from one of these categories, and so shallow groundwater and baseflow are not primary source categories. Glacier runoff, as taken from a glacier melt model such as GloGEM (Huss and Hock, 2015) or the more recent PyGEM (Rounce et al., 2020) includes all the water fluxes that occur on the glaciated area. This means that glacier runoff includes the rain, snowmelt, and glacier ice melt from the glacier area. Figures 8 and 9 show the fraction of average annual discharge (Fig. 7) and shallow groundwater (Fig. 8) composed of each of the primary sources, for each grid cell. Global discharge is dominated by rain over most of the globe, with snowmelt an important contributor at high latitudes and high altitudes, and both glacier runoff and unsustainable groundwater important regionally. The composition of shallow groundwater mirrors that of discharge. Due to human redistribution, water inputs to the land-surface can support streamflow and agriculture far from where they occurred, as can be seen in Fig. 9, which shows the source, distribution, and use of glacier runoff. As can be seen in Fig. 10, water sources like glacier runoff and unsustainable groundwater contribute to river flows, and therefore water resources, far downstream of where glacier runoff or pumped unsustainable groundwater is input to the river network. Fig. 10 also shows how tracking can identify different contributions of source water to river flows through the year, as well as how glacier runoff is an important component of water supply far downstream in the basin late in the year.

## 4.2 Return flow tracking

The return flow tracking function labels water that flows back to the system after being extracted for irrigation, livestock watering, domestic, or industrial use. Irrigation return flows are identified separately from water returned by other human uses, but returns from domestic, industrial, and livestock uses are not tracked individually (but rather are lumped into one return category) for parsimony. These return flows have water quality implications, and through this tracking function WBM can identify when a body of water is increasingly composed of water returned from anthropogenic activities. At the beginning of a simulation, all water is considered "relict", which assumes no knowledge of the source of the water. New water entering the system during the simulation period as precipitation or glacier runoff is tagged as "pristine" water. This functionality was first published in Zuidema et al. (2020).

Fig. 11 shows the fraction of average annual discharge composed of irrigation return flows, and Fig. 12 shows the fraction of irrigation water withdrawals composed of irrigation return flows (water reuse). These fractional values cannot exceed 1, even as return flows are reused multiple times; when return flow water is extracted again for reuse, it simply retains its identity as

return flow, and does not contain any new information on the number of times it has been extracted. Return flows from all human water uses contribute to water quality issues, including excess nutrients from irrigation returns and pathogens from domestic and livestock returns, some proportion of which may be attenuated by the river network depending on flow

conditions (Huang et al. 2022) before being used again. Further, reuse of return water is an important consideration in studies evaluating the 'efficiency' of irrigation or other abstractions. Management actions that decrease returns in one region may reduce water availability downstream, which may promote extraction of alternative and potentially less sustainable sources of water (Grafton et al., 2018; Grogan et al., 2017).

**4.3 Land surface attribute tracking**

**4.3.1 Model setup**

Here we use the same set of model inputs and parameters as the global 5-minute spatial resolution WBM simulation described above, but reduce the spatial domain to only simulate grid cells downstream of headwaters in the U.S. state of Wyoming. One additional model input is required for the land surface attribute tracking: identification of which grid cells are

within each of the U.S. states that intersect the spatial domain. This input (which includes a gridded file and an accompanying attribute text file) allows WBM to track water that originates as runoff within each U.S. state as it travels downstream through four major river basins (the Mississippi, Columbia, Colorado, and Great Basin) which span both sides of the continental divide. Applications of this technique would be useful for research involving transboundary conditions in river basins or using land cover masks to understand urban/rural or forest/non-forest effects in regional hydrology.


**4.3.2 Results**

Tracking runoff generated by different U.S. states demonstrates how the land surface attribute tracking can be used to identify contributions of water from any user-identified spatial attributes. The basins simulated here all contain cities and both extensive and intensive irrigated areas; the land surface attribute tracking maintains the U.S. state identification of all

surface and shallow groundwater withdrawals and return flows as water travels through the system from headwaters to river outlets. Fig. 13a illustrates how this tracking is useful for identifying multiple land attribute contributors to river discharge at a point. Fig. 13b demonstrates the spatial distribution of water from one land attribute through many downstream systems. Particularly in Fig. 13b we can see how human extractions of water – which can occur across grid cells – spreads the tracked land attribute's contributed water across the landscape.


**5 Model code**

*Brief history of model code development:* WBM was originally written in FORTRAN, and first published in Vörösmarty et al. (1989). The first publication described WBM as a continental-scale model of water balance and fluvial transport, and presented an application to South America. The first global applications were published in Vörösmarty et al. (1998) and

Vörösmarty et al. (2000a). Over its 30+ year history of development, WBM has been re-written in several programming

languages, and branches have been developed for specific applications (Table 7). Table 7 describes each branch of WBM, with its acronym (e.g., WBM vs. PWBM), the application for which the branch was developed, and key publications. Many of the branches are still in use by a variety of research groups, including researchers at The University of New Hampshire (WBM v.1.0.0), City College of New York (WBM), University of Alabama (WBMsed), and University of Massachusetts

(PWBM).

*WBM Open Source code:* The WBM version described here is written in Perl/PDL. The coding language was changed from C++ to Perl/PDL by the University of New Hampshire research group in 2010 to make use of PDL functionality that was

unique to that language at the time, namely:

1) Efficient parallel processing of matrix operations on large spatial matrices allowing increased computational performance similar to than C or Fortran through the use of binary PDL operators/functions and multithreading,

2) Adding pre-compiled custom functions written in inline C (PDL PP modules), and

3) Fully integrating the river transport module with the land surface component of WBM to simulate the full

downstream effects of water withdrawals from the rivers. Prior versions of WBM resolved the time component prior to the spatial component of the model; this prevented implementation of water extractions and inter-basin transfers.

The open source Perl/PDL version of WBM described here includes all the functionality of the original FORTRAN WBM model, the WBMplus model, and some aspects of the FrAMES model. All model branches run on Linux operating systems.

The open source WBM code described here is composed of three main files: wbm.pl, which is the main model script; WBM.pm, a module providing WBM specific functionality; and RIMS.pm, a module providing geospatial and temporal transformation utilities. The entire modeling framework is dependent upon other software: perl, PDL, gdal, ogr, and NetCDF. The model input data repository (https://wbm.unh.edu/, Grogan et al., 2022) also includes a Singularity container which has pre-installed the required operating system and software dependencies for ease of model use by the research

community.

WBM can run high density grids in a simulation domain up to about 3 million active grid cells on an average rack system server and utilize CPU parallelization (multi-threading) for a performance boost. Smaller spatial domains can be run on a personal desktop or laptop computer.


*Code implementation:* WBM is rasterized and generally used with uniformly spaced gridded data (typically in geographic coordinates, but accepting any GDAL-readable format), keeping values of gridcell-specific area in memory for flux calculations. The model is modular, with many options to turn on or off irrigation and other human water extractions. Options are controlled by the user through a selection of direct inputs, on/off flags, and output variables requested of the

model. Stocks and fluxes including irrigation demand, evapotranspiration, and runoff generation are calculated in the first

portion of the time-step loop utilizing vectorized and efficient array utilities of the perl Data Language. Water entering stream reaches throughout the network is then submitted to a routing function that traverses a directed, noncylical graph of all grid cells that ensures an upstream-to-downstream calculation order, written in an inline, pre-compiled format to maximize computational efficiency. In a number of areas, the model makes use of split-operator solutions to facilitate both

tracking functionality and the complex interactions between human water withdrawals and natural systems. This simple method allows WBM to re-calculate water stocks and fluxes after water extraction occurs and again after return flows, such that the final stock and flux values at the end of a time step are modified from the first instance of calculation at the beginning of the time step. As noted by others, leveraging of split-operator solutions for hydrologic models provides a tradeoff between efficiency and accuracy in numerical solutions, which is warranted in some cases (Clark et al., 2015).


*How to use WBM:* The WBM workflow involves 5 basic steps:
1) Prepare input data, metadata, and parameter files
2) Write a model setup file with the extension *"\*.init"*
3) Test setup file by running WBM with flags "-test", "-noRun", and "-err".

4) Execute the model code (wbm.pl)
5) Perform post-processing, if needed, with automatically generated utilities for temporal aggregation of select or all output variables.

A detailed instruction manual is included in the model's github repository, along with perl utilities commonly used in steps 1 and 5.


In Step 1, the model user must collect all input data required for the given model simulation. Each spatial data set and database must be described in a metadata file with the extension *".init"*. All data and model input ".init" files are simple text files with formatting that conforms to a perl hash. Input file unit conversions (e.g., converting temperature data from ºC to ºF) do not need to be performed prior to running WBM. Rather, the user can define a conversion slope and intercept for

linear transformations within the metadata ".init" files, and WBM will automatically calculate the new units through the RIMS.pm module.

In Step 2, the model user writes a model setup file with the extension *".init"* that lists all model inputs as well as other key parameters such as the start year, end year, list of output variables to save, and output directory location. This setup file

points directly to the input data .init metadata files, and includes options to directly define parameter values and set binary on/off flags for particular modules. Most important is the identification of the digital river network. The input river network file determines the model simulation grid spatial resolution, spatial extent, projection, and defines non-land grid cells (which are set to a no data value). Other input datasets will automatically be clipped (extent reduced) and re-gridded (either through

resampling or aggregation) to match the extent and grid cell resolution of the input river network file. This means that the
model user does not need to do these spatial transformations prior to starting the model.

In Step 3, the user tests the model setup and produces an optional input data pre-processing script. Test mode and "noRun"
mode call the input data reading functions from RIMS.pm and set up the model run's output directory. This step is used to
identify any errors in the model setup, which are commonly issues such as incorrect file paths, syntax errors in the ".*init*"
files, or formatting errors in the raw data files. Executing wbm.pl in test and noRun mode also automatically generates a
custom build_spool.pl script (written to the model run's output directory) that can optionally be executed prior to Step 4 to
pre-process all input data files that require requisite spatial clipping, re-gridding, or unit conversions. If build_spool.pl is
executed, the results of input data pre-processing are saved as binary files that are read directly by WBM; these files can also
optionally be saved as netCDF files for ease of analysis, so the user can evaluate the results of the processing step. If the
custom build_spool.pl script is not executed prior to starting the model in Step 4, pre-processing will automatically be
executed in the model's run time. Note, this automatic option only produces binary files, and does not output any netCDF
files. The build_spool.pl utility can leverage multiple CPUs to efficiently build binary input files; the automatic option
processes all binary files in a single process with a steep reduction in model simulation time.

In Step 4, the model user executes wbm.pl via direct command line entry. The code wbm.pl has several flag options,
including -h for help, -v for verbose mode, and others described in the instruction manual. The model setup file is the only
required argument to wbm.pl. Under verbose mode, detailed statistics of model run time, domain aggregate water balances,
and water supply metrics are reported to the user during each time-step, with more complete accounting of water balances
reported at the end of each year of the simulation.  Model run state files are written at the end of spin-up, and at the end of
each year, and (optionally) more frequently. This frequent saving of state files enables users to re-start simulations in the
event of an interruption (e.g., from power loss) without losing significant wall-time.  Model output files are written in the
same spatial resolution and domain as the input digital river network.

Step 5 is the most application- and user-specific. The raw daily model output is rarely the final product of analysis; temporal
and spatial aggregation or point-location time series extraction are most commonly required to evaluate output and produce
research results. The model automatically generates daily-to-monthly and daily-to-yearly temporal aggregations, and the
setup file has a binary on/off option that enables automatic temporal aggregation to climatology (daily, monthly and yearly)
averages using either the entire simulation period or specified year groups for averaging; there is also an input field for
automatic spatial aggregation. Perl utilities for these operations are included in the model github repository.
**6 Discussion**

WBM's simulation of global hydrologic fluxes are similar to many other GHMs (Tables 5 and 6). Global estimates of discharge are in line with other GHM estimates, and correspondence with observations is globally reasonable, and error-prone in specific locations. Model performance (Figures 4 and 5) is best in North America, where observational data density is high, climate reanalysis data used as input data to WBM has greater observational density to draw from (Gelaro et al. 2017), and the majority of historic regional calibration activities were focused (Stewart et al. 2011, Samal et al. 2017, Zuidema et al. 2018, 2020, Grogan et al. 2020). Low biases in discharge throughout Asia may reflect biases in input precipitation fields (Grogan 2016), or that globally assumed parameters are unrepresentative of these landscapes. Higher estimates of discharge predicted by WBM than several prior estimates (Table 5) likely reflect a combination of an increased rate of precipitation during recent decades (Blunden and Arndt, 2020), and more advanced estimates of global precipitation (Gelaro et al. 2017). The distribution of model performance metrics presented here (Figures 5 and 6) are comparable to other recent global syntheses (Lin et al. 2019, Harrigan et al. 2020). While the default parameterization can miss key distinguishing features of local hydrologic responses, the inclusion of anthropogenic processes is a critical feature for providing sufficient hydrologic simulations (Zaherpour et al. 2018). Irrigation withdrawals predicted by WBM v.1.0.0 reflect country-specific estimates from AQUASTAT over most of the globe (Figure 6). High biases for irrigation water withdrawals in Asia, particularly China and Pakistan, correspond spatially with low biases in discharge and may reflect both data and parameterization issues that should be resolved in work focusing on these specific regions. Considering the high degree of flexibility in WBM parameters and the broad range of modern meteorological input data, reasonable hydrologic simulations should be attainable within any region of the globe. However, it is important for model users to use a spatial resolution appropriate for regional study domains, and to use local data to parameterize and evaluate the model for each new study domain.

WBM's tracking functionality opens unique options for model-based experimentation with potentially important management inplications within a GHM. Oftentimes, hydrologic modeling studies provide insight to the relative importance or effect of a particular hydrologic process by switching processes on and off, thereby creating slightly different systems. These studies identify the role of a specific process in a system by comparing two or more structural or parametric model configurations with and without representation of a particular process. Such analogies are most powerful when used to understand the effect of hydrologic fluxes which are expected to fundamentally change such as glacial melt (Rounce et al., 2020), or have been historically absent in previous hydrologic modeling such as surface depressions (Rajib et al., 2020). Similar approaches may test the effectiveness of different management strategies, such as the effect of managed aquifer recharge on aquifer head and river flow (Niswonger et al., 2017; Tran et al., 2019; Van Kirk et al., 2020; Zuidema et al., 2020). In other cases, this approach has been used to assess the difference between a hypothetical natural system (with no human impacts) and a human-impacted system (Wada et al., 2016).

By using the tracking methods described here, it is possible to attribute a portion of water flows to a specific process, location, water source, or flow-path, without altering the represented system from an existing or experimental configuration. This is fundamentally different from the on/off method of evaluating process or source importance that has been more commonly used in the literature. WBM's tracking module achieves this by attributing the water stored and moving between each pool within each grid cell a composition of processes or water sources that brought that water to a point in space and

time. For example, this tracking function facilitates calculation of irrigation returns in future withdrawals that make estimation of effective irrigation efficiency (Haie and Keller, 2008) possible under suites of hypothetical management configurations (Zuidema et al. 2020). As there are no equivalent empirical analogues, evaluating the tracking component compositions of any flux is not presently possible; however, tracking functionality creates a more transparent representation of the assumptions that drive model results.


    As described in Weiler et al. (2018), several different water tracking methods have been employed by regional hydrologic models, though as of this wriring, no global hydrologic model other than WBM employs these types of tracking methods. Regional hydrologic model tracking methods include synthetic scalar transport, solute transport, particle tracking, and the "effective tracking" used in HBV-Light (Stahl et al. 2017, Weiler et al. 2018). WBM's tracking fits into the class of

"effective tracking" methods described by Weiler et al. (2018), and is a simplified version of the synthetic scalar transport method, analogous to solute transport where mixing within compartments of the model is substituted for a full calculation of the advection-dispersion equation. Insights provided by effective tracking into the sources of discharge and water provisioning are most relevant for evaluating human water resources (Weiler et al. 2018).

**7 Conclusions and future work**

    The open source global hydrologic model WBM v.1.0.0 represents not only the natural terrestrial hydrologic system, but also human interactions with water resources. These interactions include hydro-infrastructure and water extractions for use by irrigation, livestock, domestic, and industrial sectors. WBM v.1.0.0 provides a novel water component tracking functionality that enables GHMs for the first time to attribute the influence of different water sources and flow paths on stocks and fluxes

such as river discharge or irrigation water supply. Tracking illustrates the importance of teleconnections between input sources and human uses, such as the withdrawal of glacier water far downstream, or the extraction of agricultural returns for subsequent reuse. It does this by calculating the impact of water introduced by a flux without the need to estimate the effects by altering the system through their absence, which is critical for understanding how we interpret how the system is, rather than how a similar system might be. Evaluation of the global model shows good agreement with observed river discharge

and water extractions, though the evaluation metrics have large spatial variability that highlights the need for parameter calibration when using WBM v.1.0.0 for regional analyses. On-going development of WBM focuses on modules that improve the representation of human interactions with the water cycle, increased temporal resolution options, and data assimilation functionality for use in operational forecasts.

### Code and data availability

WBM v.1.0.0 is open source and distributed under the terms of the GNU Public License version 3, as published by the Free Software Foundation. Model code is provided in a GitHub repository: https://github.com/wsag/WBM, and release v.1.0.0 is archived on Zenodo (Grogan and Zuidema, 2022, https://zenodo.org/record/6263097#.Yhhvk5PMKRs). Input data required to reproduce the simulations presented here that cannot be directly downloaded from other sources due to either lack of availability or substantial pre-processing requirements for use in WBM v.1.0.0 (see Table 4) are provided for download here: https://wbm.unh.edu/ (Grogan et al., 2022; https://dx.doi.org/10.34051/d/2022.2). The GitHub repository will be updated as bug-fixes, new modules, and further development occurs. Development and maintenance of the main branch of WBM continues at the University of New Hampshire, and we welcome contributions from other parties.

### Author contributions

DG contributed to conceptualization, methodology, formal analysis, validation/evaluation, visualization, and original draft writing. SZ contributed to software development, data curation, and draft writing. AP contributed to software development (lead developer), investigation, data curation, and draft writing. SG contributed to software development, investigation, and data curation. RL contributed to conceptualization, funding acquisition, project administration, supervision, writing – review and editing. WW contributed to funding acquisition, project administration, supervision, and writing – review and editing.

### Competing interests

The authors declare no competing interests.

### Acknowledgements

This work was supported by: The U.S. Department of Energy, Office of Science, Biological and Environmental Research Program, Earth and Environmental Systems Modeling, MultiSector Dynamics under Grant DE-SC005171 and Cooperative Agreements DE-SC0016162 and DE-SC0022141. The National Science Foundation Division of Earth Sciences grant no. 10388018; Division of Social and Economic Sciences grant no. 1639524; Division of Chemical, Bioengineering, Environmental, and Transport Systems grant no. 1855937; Division of Behavioral and Cognitive Sciences grant no. 1114851; NH EPSCoR New England Sustainability Consortium grant no. EPS-1330641; NH EPSCoR Ecosystems and Society grant no. EPS-1101245; the Plum Island Long Term Ecological Research site grant no. OCE-1637630; and the Graduate Research Fellowship Program grant no. DGE-0913620. The National Aeronautical and Space Administration, Earth Science Division's High Mountain Asia program grant no. NNX17AB28G and grant no. 80NSSC20K1595 and the Earth Science Division's Sea Level Change program grant no. 80NSSC20K1296. The United States Environmental Protection Agency, Science to Achieve Results (R836169), Assessing the Contribution of Small Streams to Use and Non-use

WaterQuality Values Using Modeling, Stakeholder Participation, and Decision Theory. The Swedish funding agency Formas under grant #2017-00,608 via Stockholm University.

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

 **Appendix 1**

**Table A1.** Examples of WBM applications over different regions across the globe.

| Region | Citations |
|---|---|
| Global | Fekete et al., 2006; Grogan, 2016; Grogan et al., 2017; Liu et al., 2017; Schewe et al., 2014; Vörösmarty et al., 2000a, 2010; Wisser et al., 2008; Wisser et al., 2010a,b |
| Arctic | Bring et al., 2017; Rawlins et al., 2003, 2005, 2019; Rawlins et al., 2006a,b; Shiklomanov et al., 2013 |
| Asia | Douglas et al., 2006; Grogan et al., 2015; Groisman et al., 2020; Mishra et al., 2020; Zaveri et al., 2016 |
| Africa | Vörösmarty et al., 2005 |
| South America | D'Almeida et al., 2006; Vörösmarty et al., 1989 |
| North America | Grogan et al., 2020; Rougé et al., 2021; Samal et al., 2017; Stewart et al., 2011b, 2013; Vörösmarty et al., 1998; Webster et al., 2022; Zuidema et al., 2018, 2020 |
| Tropics | Douglas et al., 2005; Douglas et al., 2006 |

**Table 1.** Default parameter values,with minimum (Min) and maximum (Max) suggested parameter ranges, and the WBM default value (Default) when the parameter value is not user-defined.

| Parameter | Description | Units | Min | Max | Default |
|---|---|---|---|---|---|
| $\alpha$ | Evapotranspiration response to soil moisture drying | — | 2 | 20 | 5 |
| $C_{LAI}$ | Maximum canopy interception storage | mm | 0 | 1 | 0.2 |
| $\gamma$ | Percolation fraction below root zone | — | 0.1 | 0.9 | 0.5 |
| $\beta$ | Baseflow release time-constant | $d^{-1}$ | 1e$^{-3}$ | 0.1 | 0.025 |
| $C_{SRP}$ | Quickflow release coefficient | — | 0.2 | 1.0 | 0.75 |
| $T_{SRP}$ | Threshold storage allowed in quickflow pool | — | 10 | 50 | 1000[a] |
| $L$ | Temperature lapse rate | °C/km | -8 | -6 | -6.49 |
| $T_s$ | Snowfall temperature threshold | °C | -2 | 0 | -1 |
| $T_m$ | Snow melt temperature threshold | °C | 0 | 2 | 1 |
| $R_{perc}$ | Fraction of irrigation returns to groundwater | — | 0 | 1 | 0.5 |
| $R_{ind}$ | Fraction of returns from industrial use | — | 0 | 1 | 0.86 |
| $R_{dom}$ | Fraction of returns from industrial use | — | 0 | 1 | 0.89 |
| $U_w$ | Speed of wave propagation (celerity) | m s$^{-1}$ | 1 | 3 | 2.18 |
| | Hydraulic Geometry | | | | |
| $\eta$ | Coefficient relating mean discharge and depth | — | 0.1 | 0.33 | 0.25 |
| $\nu$ | Exponent relating mean discharge and depth | — | 0.1 | 0.5 | 0.40 |
| $\tau$ | Coefficient relating mean discharge and width | — | 3.7 | 10 | 8.00 |
| $\phi$ | Exponent relating mean discharge and width | — | 0.2 | 0.7 | 0.58 |
| f | Exponent relating instantaneous discharge to mean depth | — | 0.35 | 0.75[b] | 0.40 |
| b | Exponent relating instantaneous discharge to mean width | — | 0 | 0.25[b] | 0.10 |
| m | Exponent relating instantaneous discharge to mean velocity | — | 0.25 | 0.65[b] | 0.50 |

[a] – Default value effectively defines no upper bound to storage within the surface runoff pool.

[b] – The sum of f, b, and m (and $\nu$, $\phi$, and $\epsilon$) must equal 1.

**Table 2.** Default livestock parameters for the livestock water use module.

| Livestock | Slope, $s_j$ | Intercept, $I_l$ | ServiceWater, $B_l$ | Population Growth Rate |
|---|---|---|---|---|
| buffalo | 0.345 | 16.542 | 5 | 0.001863 |
| cattle | 0.345 | 16.542 | 5 | 0.001863 |
| goats | 0.215 | 4.352 | 5 | 0.003731 |
| pigs | 1.4575 | -6.14 | 25 | 0.000309 |
| poultry | 0.019 | 0.1823 | 0.09 | 0.13397 |
| sheep | 0.57 | -0.35 | 5 | 0.003 |

**Table 3**: Tracking component categories, and the identification of the water source components tracked

| Tracking group | Water components tracked |
|---|---|
| Primary Source Components | Rain* <br> Snow <br> Glacier runoff <br> Unsustainable Groundwater |
| Return Flow | Pristine (no return) <br> Domestic/Livestock/Industrial returns <br> Irrigation returns <br> Relict*,[1] |
| Land surface labels | ID_1, ID_2, …, ID_N |

*This component comprises 100% of reservoir and soil moisture stocks prior to spinup
[1]Relict water is defined as water stored in all water storage pools (aka stocks) at the beginning or end of spinup.

**Table 4.** Model input datasets for WBM simulations presented here.

| Input data type | Input data | Download link or website | Citation |
|---|---|---|---|
| River network | MERIT 5-minute river network | http://hydro.iis.u-tokyo.ac.jp/~yamadai/MERIT_Hydro/ | Yamazaki et al. (2019) |
| Precipitation (daily) | MERRA 2 (prectotcorr variable) | https://gmao.gsfc.nasa.gov/reanalysis/MERRA-2/ | Gelaro et al. (2017) |
| Temperature (daily average) | MERRA 2 | https://gmao.gsfc.nasa.gov/reanalysis/MERRA-2/ | Gelaro et al. (2017) |
| Dams and reservoirs | HydroConDams v2.0 for the continental US, and GrAND v1.3 for outside the continental US. | https://dataverse.harvard.edu/dataset.xhtml?persistentId=doi:10.7910/DVN/5YBWWI and https://globaldamwatch.org/grand/ | Lehner et al. (2011); Zuidema and Morrison (2020) |
| Soil available water capacity | Harmonized world soil database v1.2 | https://www.fao.org/soils-portal/data-hub/soil-maps-and-databases/harmonized-world-soil-database-v12/en/ | Fischer et al. (2008) |
| Root depth* | Effective rooting depth from Yang et al. (2016), gap-filled with the FAO/UNESCO digital soil map of the world v3.6 | https://doi.org/10.4225/08/5837b3aa9cb90 and https://www.worldcat.org/title/digital-soil-map-of-the-world-and-derived-soil-properties/oclc/52200846 | FAO/UNESCO (2003); Yang et al. (2016) |
| Glacier runoff, volume and area* | GloGEM glacier model | | Huss and Hock (2015) |
| Crop maps and calendars* | MIRCA2000 v1.1 | https://www.uni-frankfurt.de/45218023/MIRCA | Portmann et al. (2010) |
| SW:GW ratio* | FAO AQUASTAT | https://www.fao.org/aquastat/statistics/query/index.html;jsessionid=71F6F6340C470CFBE92D71489546AA39 | FAO (2015) |
| Irrigation Efficiency | Rasterized data from Table 1 of Döll and Siebert (2002) | https://agupubs.onlinelibrary.wiley.com/doi/full/10.1029/2001WR000355 | Döll and Siebert (2002) |
| Rice paddy percolation rate* | Derived from the FAO/UNESCO soil map of the world | https://www.fao.org/soils-portal/soil-survey/soil-maps-and-databases/faounesco-soil-map-of-the-world/en/ | FAO/UNESCO (2003), with derived data described by Wisser et al. (2008) |

*Primary data was processed for formatting, gap-filling, or to generate a calculated product; the resulting formatted files are provided for download at https://wbm.unh.edu/ (Grogan et al., 2022) for simulation reproducibility.

**Table 5:** Model estimates of global river discharge, ordered from lowest (based on the low end of a range, for models that report ranges) to highest. Parentheses show the exorheic + endorheic discharge for studies that provide a separation of external and internal basin discharge.

| Source | Model name | Year | Value (km³ year⁻¹) |
|---|---|---|---|
| Oki et al. (2001) | 11 LSMs + TRIP | 1987 – 1988 | 29,485 |
| Rost et al. (2008) | LPJmL | 1971 – 2000 | 35,355 - 37,119 |
| Van Beek et al. (2011) | PCR-GLOBWB | 1958 – 2001 | 35,387 - 36,812 |
| Nijssen et al. (2001) | VIC | 1980 – 1993 | 36,006 |
| Döll et al. (2003) | WaterGAP 2 | 1961 – 1990 | 36,687 |
| Wisser et al. (2008) | WBMplus | 1901 – 2002 | 37,401 |
| Dai and Trenberth (2002) | WBM + RTM | Mean from gauge record | 37,288 |
| Döll et al. (2009) | WaterGAP | 1961-1990 | 38,164 - 39,564 |
| Widen-Nilsson et al. (2007) | WASMOD-M | 1961 – 1990 | 38,605 |
| Vorosmarty et al. (2000) | WBM | 1961 – 1990 | 39,294 |
| Fekete et al. (2002) | WBM | 1970-1980 | 39,307 (38,314 + 993) |
| Fekete et al. (2000) | WBM | - | 39,476 (38,401 + 1,075) |
| Schmied et al. (2014) | WaterGAP | 1971 – 2000 | 40,002 – 46,822 |
| Gerten et al. (2004) | LPJ | 1961 – 1990 | 40,143 |
| Sutanudjaja et al. (2018) | | 2000 – 2015 | 42,393 – 43,978 |
| *This study* | WBM v.1.0.0 | 2000 – 2009 | 42,957 (40,248 + 2,709) |
| Liang and Green (2020) | Empirical | 1905 – 2016 | 47,884 (46,221 + 1,663) |


**Table 6:** Global estimates of total irrigation water withdrawal, including this study's simulation.

| | Source | Model | Year | Value (km$^3$ year$^{-1}$) |
|---|---|---|---|---|
| Total global irrigation water withdrawal | Döll and Siebert (2002) | WaterGAP | 1961 – 1990 | 2,452 |
| | Wisser et al. (2008) | WBMplus | 2000 | 2,000 – 4,100 |
| | Rost et al. (2008) | LPJmL | 1971 – 2000 | 2,555 |
| | Sulser et al. (2010) | IMPACT | 2000 | 3,128 |
| | Wada et al. (2011) | PCR-GLOBWB | 1958 – 2001 | 2,057 |
| | Pokhrel et al. (2012) | MATSIRO | 2000 | 2,462 (± 130) |
| | Döll et al. (2014) | WaterGAP | 2003 – 2009 | 2,400 |
| | Wada et al. (2014) | PCR-GLOBWB | 1979 – 2010 | 2,217 – 2,885 |
| | Hanasaki et al. (2018) | H08 | 2000 | 2,544 (± 75) |
| | Grogan et al. (2017) | WBM | 2000 | 3,244 (± 240) |
| | Sutanudjaja et al. (2018) | PCR-GLOBWB | 2000 – 2015 | 2,309 – 2,735 |
| | AQUASTAT (FAO, 2016) | *Reported statistics* | 2000 | 2,434 |
| | This study* | WBM v.1.0.0 | 2000 – 2009 | 3,889 (± 126) |

*Uncertainty estimate is the standard deviation of annual values from 2000 – 2009.

**Table 7.** Major WBM code branches, along with their history of new functionality and if the code branch is still in use.

| Model full name | Acronym | Language | New functions | Key publications | In use |
|---|---|---|---|---|---|
| Water Balance Model | WBM | FORTRAN | Original: continental to global scale water balance | D'Almeida et al. (2006); Vörösmarty et al. (1989) | No |
| Water Balance Model | WBM | C/C++ | Original: continental to global scale water balance | Fekete et al. (2002); Vörösmarty et al. (1998 2000, 2005, 2010) | Yes |
| Pan-Arctic Water Balance Model | P/WBM and PWBM | FORTRAN | Added permafrost functions for pan-arctic applications | Rawlins et al. (2003, 2005); Rawlins et al. (2006a,b) Rawlins et al. (2021a,b) | Yes |
| Framework for Aquatic Modeling of the Earth System | FrAMES | C++ | Constituent fluxes into river systems, accounting for transport and fate of nitrogen, chloride, and E. coli<br><br>Water temperature | Miara et al. (2017, 2019); Miara and Vörösmarty (2013); Mineau et al. (2015); Samal et al. (2017); Stewart et al. (2011b; 2013); Wollheim et al. (2008a,b); Wollheim et al. (2015); Zuidema et al. (2018) Huang et al. (2022) | Yes |
| Water Balance Model plus | WBMplus | C++ | Irrigated agriculture and reservoirs | Wisser et al. (2008); Wisser et al. (2010a,b) | No |
| WBM sediment | WBMsed | C++ | Sediment transport | Cohen et al. (2013, 2014); Dunn et al. (2019) | Yes |
| Water Balance Model | WBM | Perl/PDL | Added rainfed agriculture, other land cover types, inter-basin transfers, domestic and livestock water demand, tracking.<br><br>Includes FrAMES functionality, and water temperature | Grogan (2016); Grogan et al. (2015, 2017, 2020); Haqiqi et al. (2021) Liu et al. (2017); Mishra et al. (2020); Webster et al. (2022) Zaveri et al. (2016); Zuidema et al. (2020) Grogan et al. (2022a) | Yes |
| Water Balance Model v.1.0.0* | WBM v.1.0.0 | Perl/PDL | Code released open source | This publication | Yes |

*This version of WBM is the open source model described in this paper.

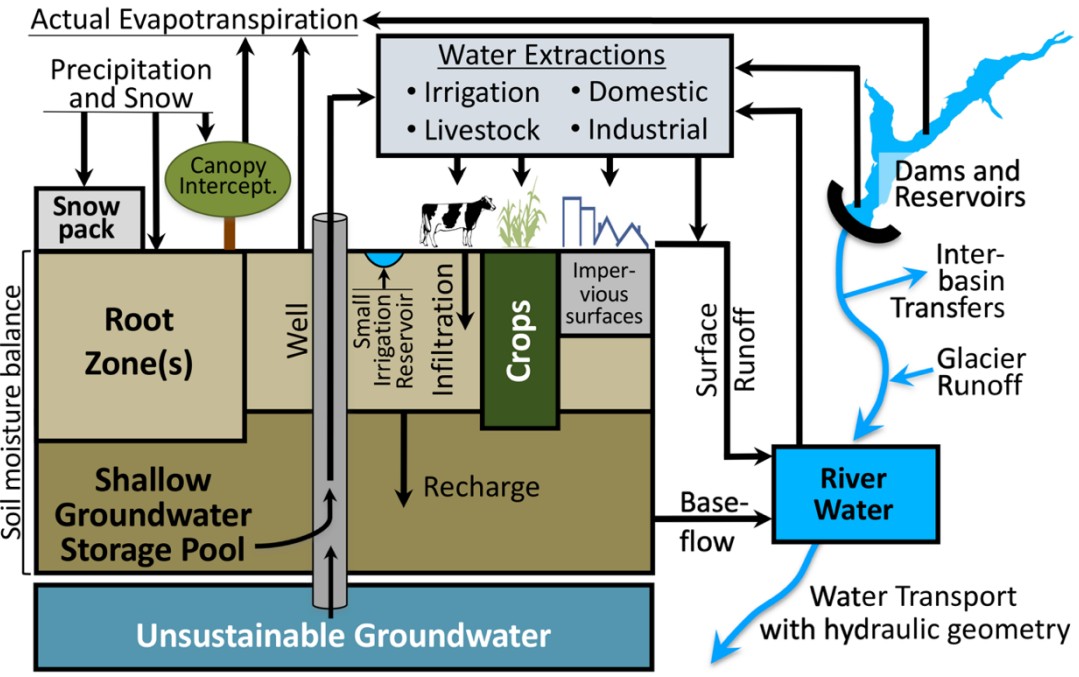

**Figure 1.** Water Balance Bodel schematic showing major fluxes and storages, which are described below in sections 2.2.1 –
2.2.6. The land surface fluxes are described in section 2.2.1, and are: precipitation and snow, canopy interception, open
water, runoff from impervious surfaces, soil moisture balance, actual evapotranspiration, surface runoff, and glacier runoff.
Both the shallow groundwater storage pool and the unsustainable groundwater are described in section 2.2.2. River water,
including baseflow and hydraulic geometry, is described in section 2.2.3. Section 2.2.4 describes dams and reservoirs, inter-

basin transfers, and small irrigation reservoirs. All water extractions are described in section 2.2.5. The model operates on
daily time steps and over grid cells defined by the digital river network. Grid cell resolutions have been used in the range
from 30 arc minutes to 120 m.

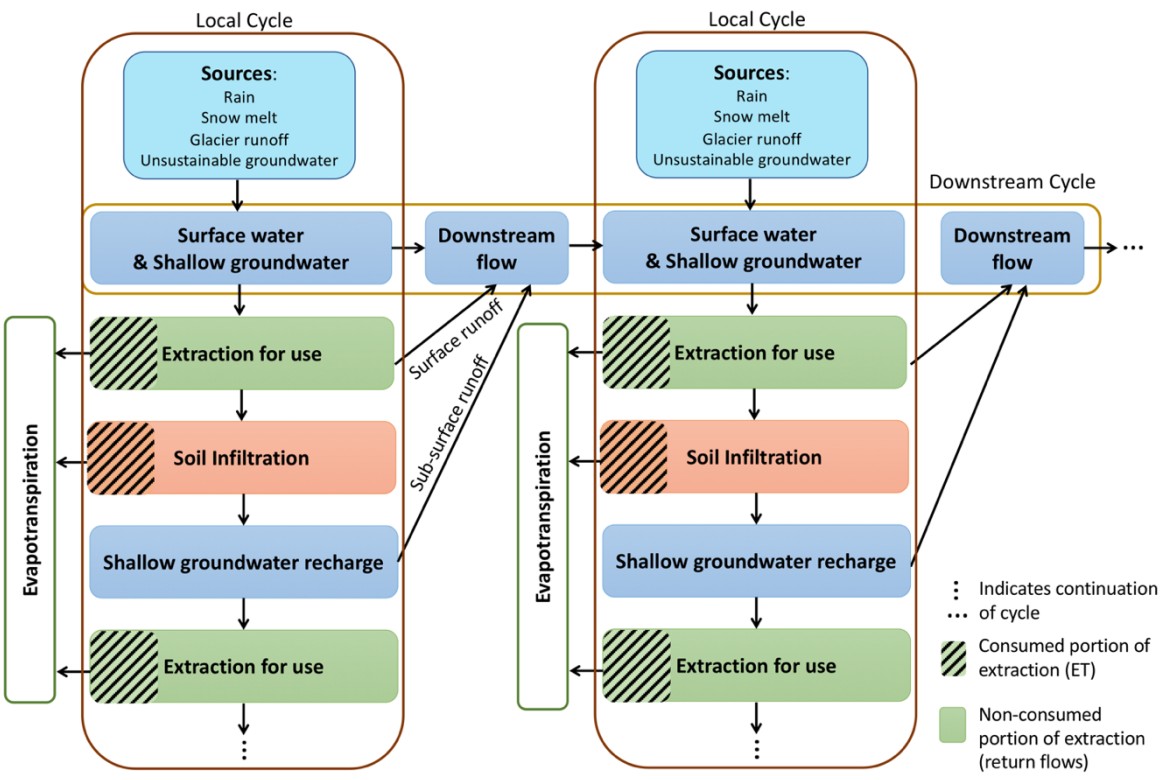

**Figure 2:** Primary source component tracking schematic. All surface and shallow groundwater is composed of the four primary sources: rain, snow melt, glacier runoff, and unsustainable groundwater. When surface and/or shallow groundwater is extracted for use, this initiates both a local cycle and a downstream cycle of water use and re-use. In the example shown here, water is extracted and applied to soils (irrigation). A portion of the extracted water and a portion of the soil water becomes evapotranspiration (the consumed portion, shown with hashes). Some of the water applied to soils percolates to the shallow groundwater pool. Water from the shallow groundwater pool can be extracted again, continuing the local water re-use cycle. Water extracted for use, and water from the shallow groundwater pool, generate runoff that moves downstream. This initiates a downstream cycle in which this water can be re-extracted for use from the surface water sysem. Downstream cycles intersect with local cycles, as water from the four primary sources are input in every locality. Figure modified from Grogan et al. (2017).

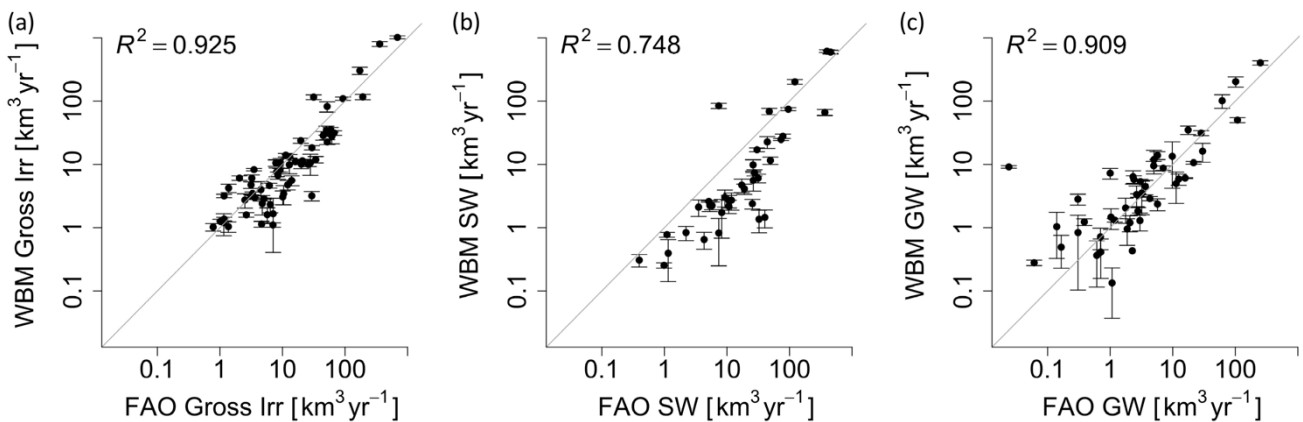

**Figure 3.** WBM modeled annual irrigation water withdrawals compare well to FAO AQUASTAT (FAO, 2016) country-level reported (a) total irrigation, (b) surface water use, and (c) groundwater use. Note that both the x and y axes are on a log scale. Total values from updated simulations (WBM v.1.0.0) are reported in Table 6. Figure modified from Grogan et al. (2017).

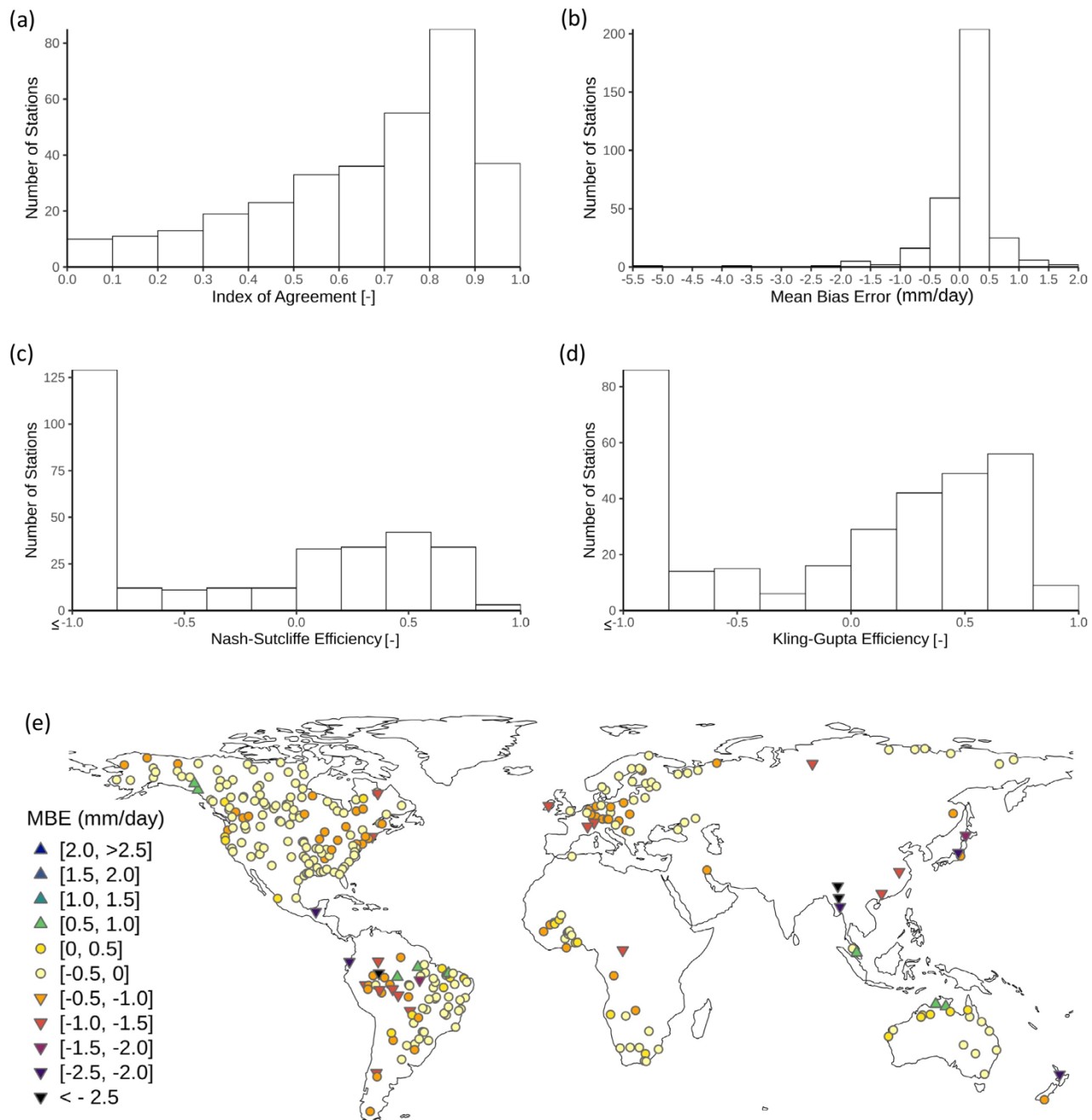

**Figure 4.** Frequency distribution of the Index of Agreement (a), the Mean Bias Error (b), the Nash-Sutcliffe Efficiency (c), and the Kling-Gupta Efficiency (d) for daily average discharge. Panel (e) shows a map of Mean Bias Error in daily discharge [mm/day], illustrating the spatial variation in bias. Average Index of Agreement is 0.56, and average MBE is -0.07 mm day⁻¹.

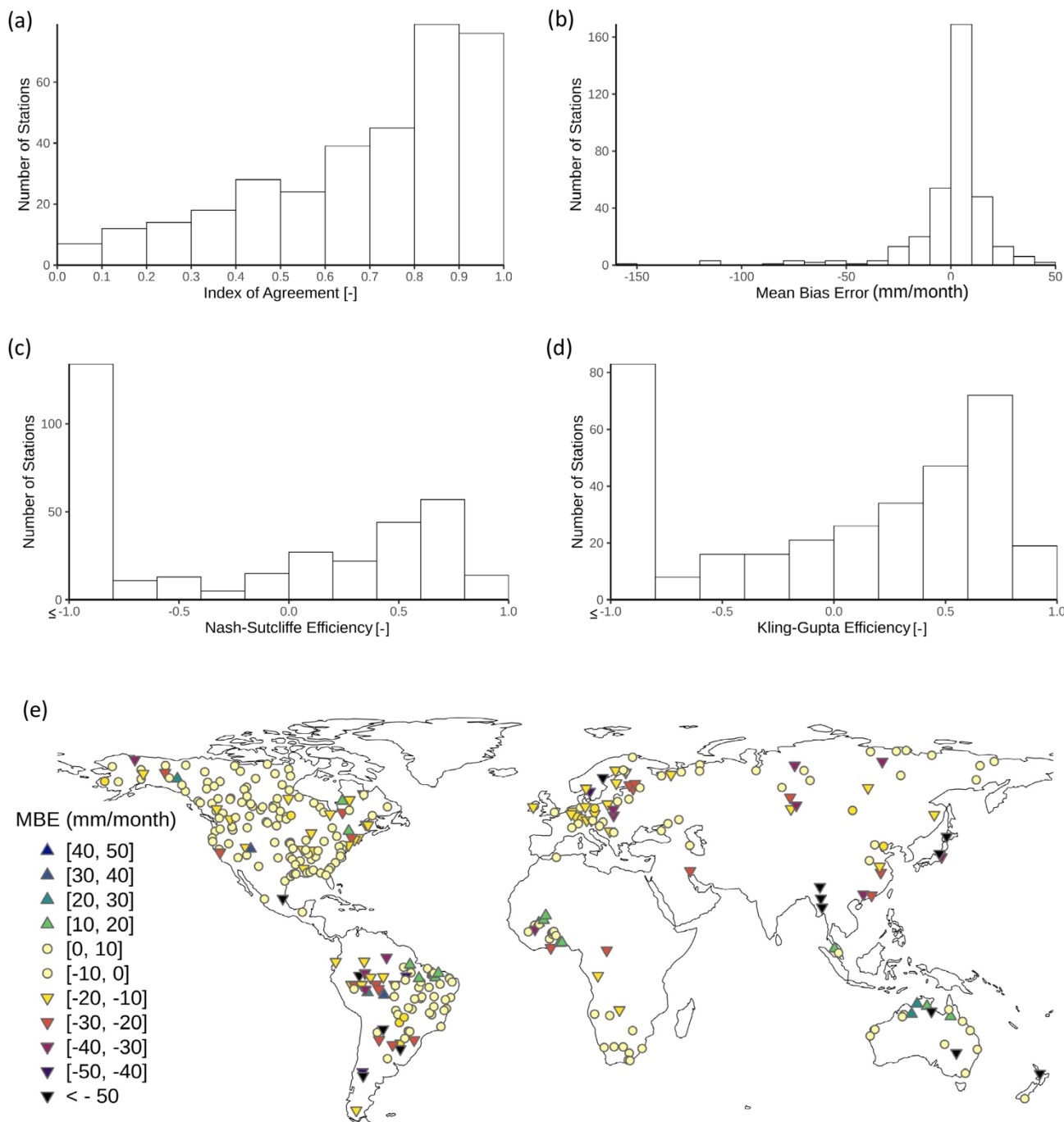

**Figure 5.** Frequency distribution of the Index of Agreement (a) and the Mean Bias Error (b), the Nash-Sutcliffe Efficiency (c), and the Kling-Gupta Efficiency (d) for monthly average discharge. Panel (e) shows a map of Mean Bias Error in daily

discharge [mm/month], illustrating the spatial variation in bias.. Average Index of Agreement is 0.69, and average MBE is -0.14 mm month$^{-1}$.


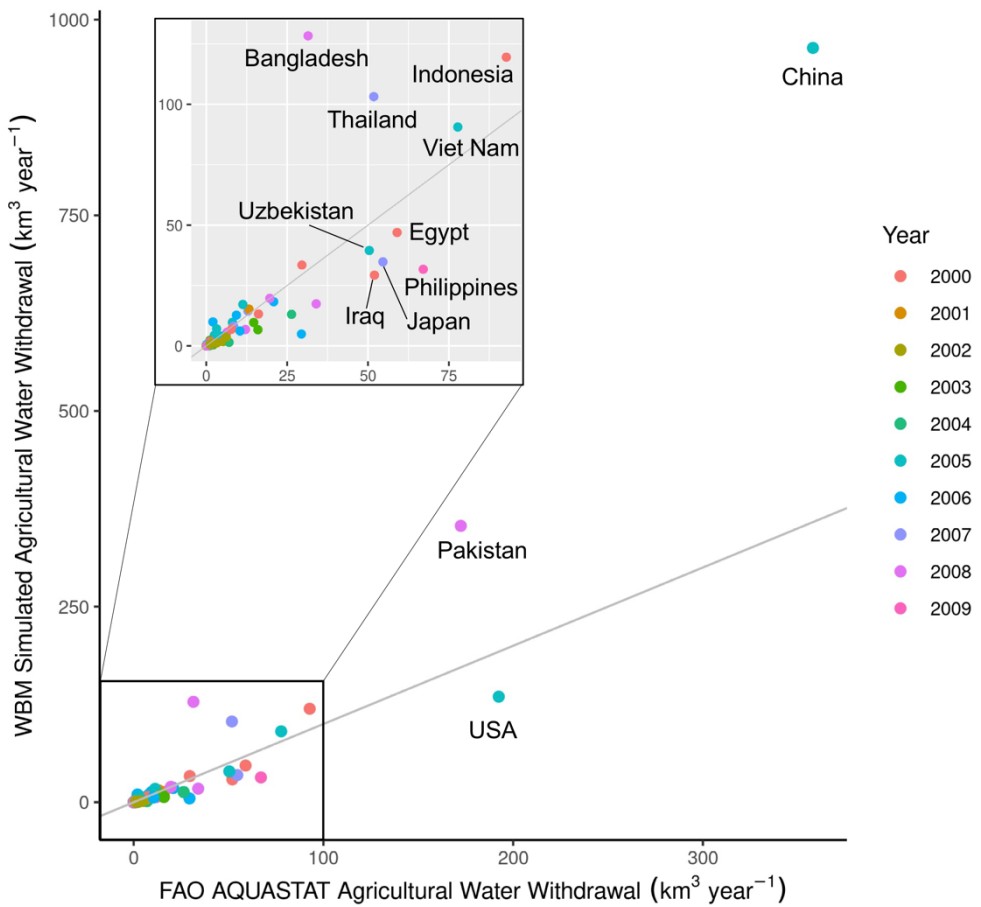

**Figure 6.** WBM-simulated irrigation water withdrawals compared to FAO AQUASTAT-reported values, by country. The 1:1 line is shown in grey. Countries with FAO-reported agricultural water withdrawals < 100 km³ year⁻¹ are shown in the inset. Countries with FAO-reported agricultural water withdrawals > 50 km³ year⁻¹ are labeled.

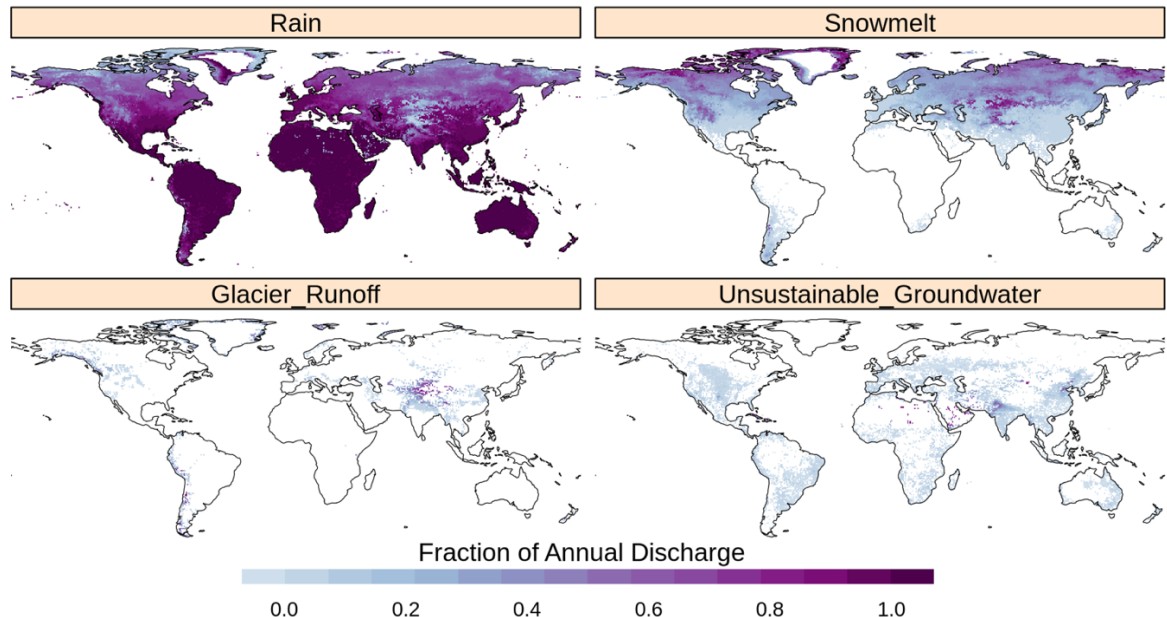

**Figure 7.** The fraction of annual average discharge composed of the four different primary source water components used in the primary source tracking method.

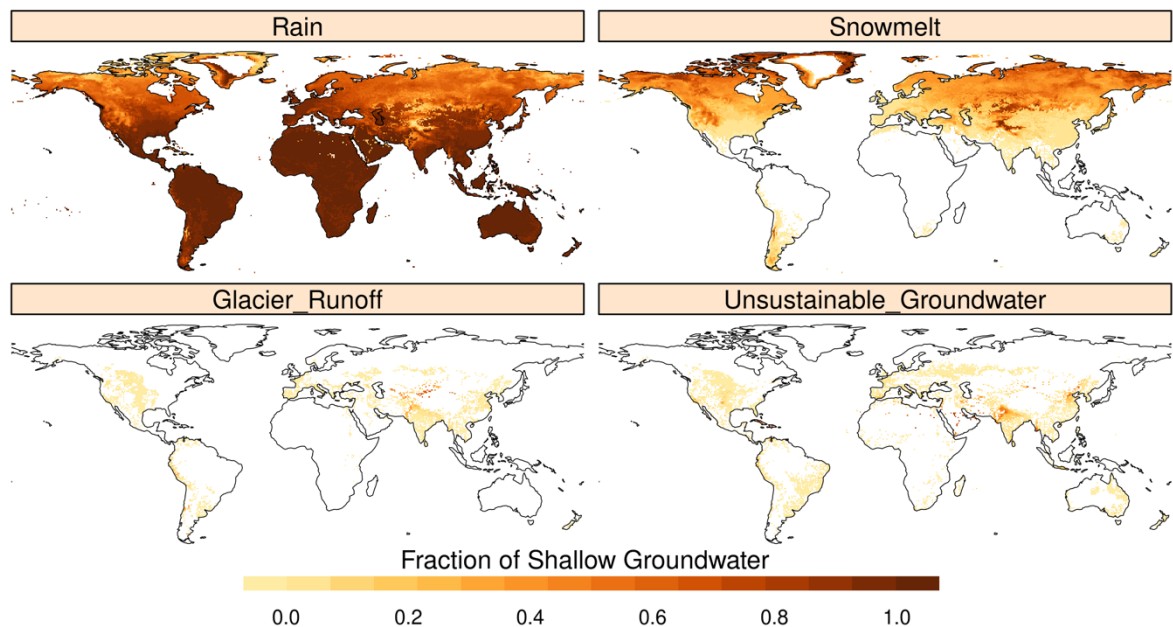

**Figure 8.** The fraction of annual average shallow groundwater storage composed of the four different primary source water components used in the primary source tracking method.

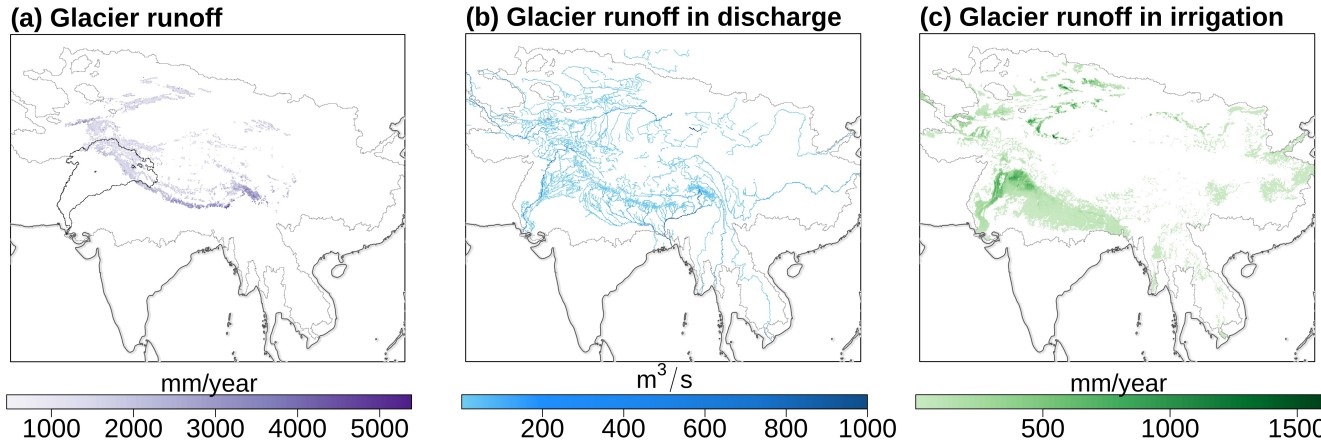

**Figure 9.** Tracking glacier runoff (a) downstream through rivers (b) and irrigation water use (c) in High Mountain Asia, a region where glacier meltwater is an important resource. While glacier water originates in the mountains, its' use in agriculture is extensive due to reuse through the river network and shallow groundwater stores, and retention in and distribution from large reservoirs. The boundary of all High Mountain Asia basins with glaciers at their headwaters is shown in grey. Panel (a) also shows the Indus basin boundaries for reference in comparing with Figure 10.

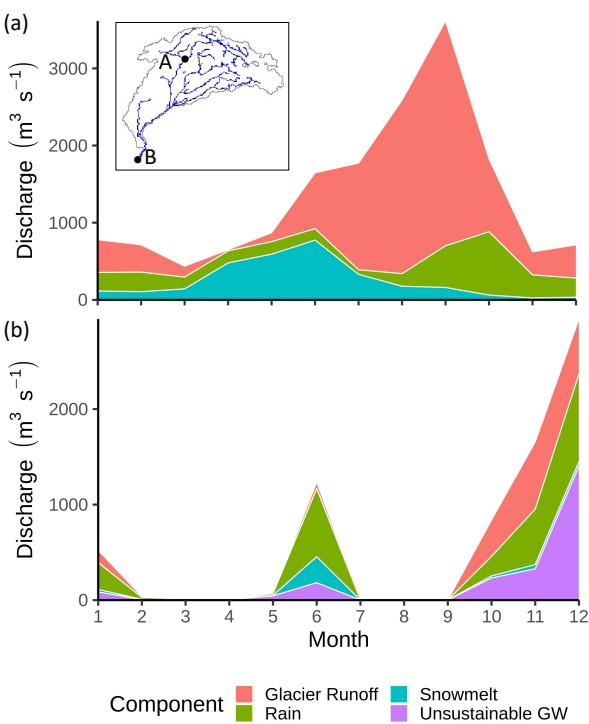

**Figure 10.** Monthly discharge by primary source component at two points (A shown in panel (a), B shown in panel (b)) along the Indus River (basin shown in inset). Point A is in the headwaters, with substantial contributions from snowmelt and glacier runoff. Point B is the river mouth. The mouth of the Indus River runs dry in this simulation due both the seasonality of precipitation in the monsoonal region, and the large amounts of water extracted from the river for use. This use and re-use of water can be seen in the distribution of primary water sources remaining at the mouth of the river, which include

unsustainable groundwater (purple) that was pumped from upstream sources, as well as snowmelt (blue) and glacier runoff (red), both of which are generated far upstream of the river mouth.

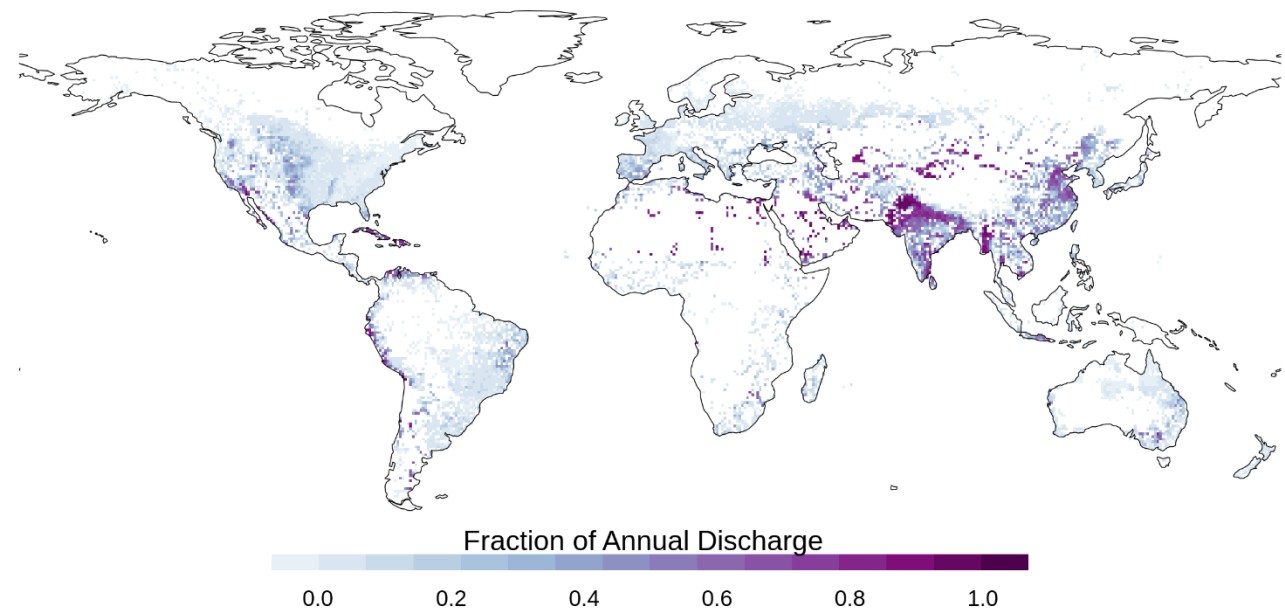

**Figure 11.** Fraction of average annual discharge composed of irrigation return flow water, simulated by the return flow tracking function. The large fraction of irrigation return flows in northern India and Pakistan results from this region being one of the most intensively irrigated regions in the world, combined with very low (~30%) classical irrigation efficiency (Zaveri et al., 2016; Grogan et al., 2017). This combination means that large volumes of water are extracted for irrigation, and nearly two-thirds of that water returns to the system.


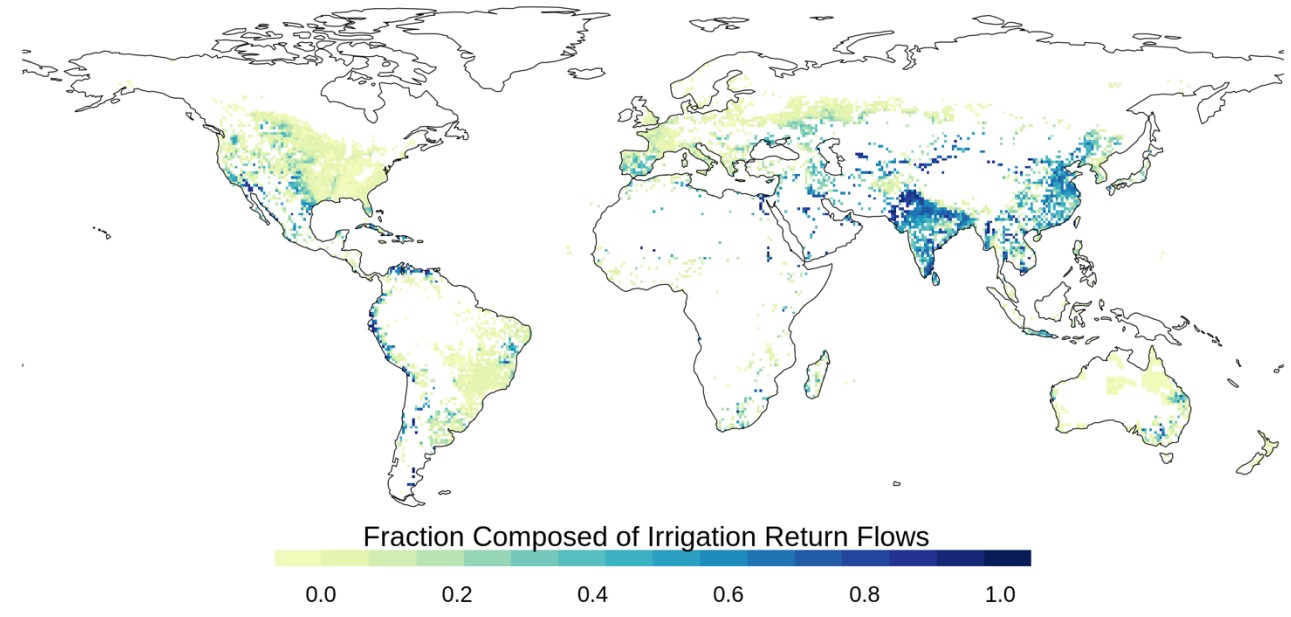


**Figure 12.** Fraction of average annual irrigation withdrawals composed of prior irrigation returns simulated by return flow tracking function.

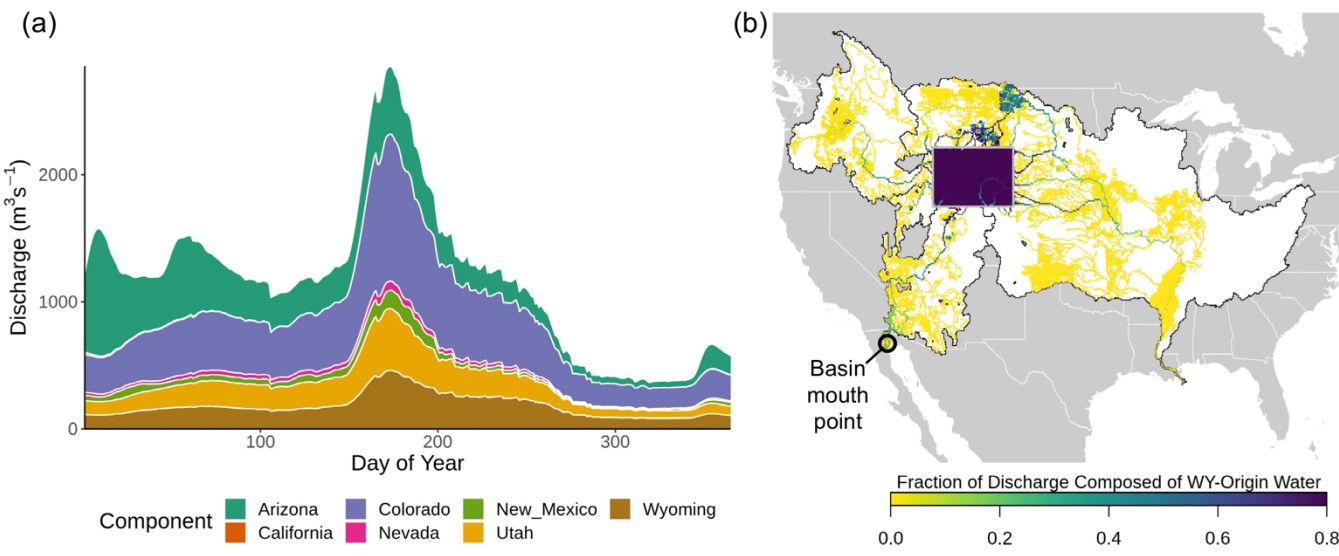

**Figure 13.** Demonstration of WBM's land surface attribute tracking function: (a) Discharge at the mouth of the Colorado River Basin (basin mouth point shown in panel (b). Colors show contribution of water from seven different U.S. states, accounting for the movement, storage, and cycles of extraction occurring across the basin's many upstream water uses. (b) Fraction of Wyoming state waters in river discharge on July 1, 2009. Basins in the study domain are outlined in black; grey regions are not simulated; the state of Wyoming is outlined in light grey. The state of WY appears dark in (b) because nearly all river discharge within the state of WY is >80% WY-origin water. Where rivers enter WY, such as the North Platte River, the color lightens as a larger portion of non-WY water enters the state boundaries; as more WY-sourced water is added to those rivers, the color gets progressively darker.