# Peer review of "WBM v.1.0.0: A scalable gridded global hydrologic model with water tracking functionality"

_Geoscientific Model Development, 2022_

## Author Response (AR1)

Author responses in bold, and original referee comments reproduced here for context.

**Reviewer #1**

**We would like to thank Reviewer #1 for their very thorough commentary, and this has significantly improved the paper.**

The model description paper of Grogan et al. describes an Open Source version of the model WBM. Overall, thy provide a well-structured summary of the model. I like to also highlight the availability of data and model source code. However, I also think the manuscript requires some clarifications to be a helpful addition to the scientific community.

Foremost, the abstract and introduction provide no indication of why the model is relevant and how its result already has or will contribute to our scientific knowledge. It is also unclear how this model differs from the vast collection of other global hydrological models. What are the features that make it unique? Why should I be interested as a potential user and scientist to have a closer look? What are the current challenges?

**We've modified the abstract, highlighting the novel aspects of WBM.**

Additional notes:

Previous GMD guidelines stated that the model version needs to be noted in the manuscript title. Please check if that is still the case.

**You are correct, we will add the version number to the title.**

11: what does long mean? Maybe instead, refer to the first published version in year X

**We will make this change to clarify that WBM has a publication history since 1989.**

12: So, the previous versions have not included it, and this is a new feature?

**While the tracking features have been used in prior publications such as Grogan et al. 2017 and Zuidema et al. 2020, the tracking module has not been described generally until this publication.**

14: I do not think it is necessary to refer to the GitHub link in the abstract. Please instead describe what makes WBM unique and why it is useful. I am halfway into the abstract and still have no idea why I should care about the model

15: Remove unnecessary technical detail in the abstract.

16-17: Ok, so what have you learned? What is the model able to do? Why should I care as a scientist and possible user?

**We will update the abstract to remove the GitHub link, reduce technical detail and expand upon what we learned, what the model can do, and why it is useful.**

17 - 22: Ok, so this is really interesting, but the sentences are long. If this is a unique feature of this model, it should be stated. In what new ways can we perform experiments with that model that are not possible with other models? After reading only the abstract, it is still unclear why I should care about this model and how it has maybe already contributed to science and will continue to be of interest. What are the scientific questions that it is designed to answer or will enable us to answer in the future? How does it differ from other models? How well or bad does it perform overall / compared to other models? What is the spatial resolution?

**We have revised the abstract for clarity and to highlight what is new.**

Introduction: I think you provide an excellent summary of what has been developed. However, I wonder if that should be condensed to a table instead. Half of the test is just references. Also, it would be nice to focus more on why we build these global models and what kind of questions they are supposed to answer, and what they can't do. There are obvious limitations, and people have been criticizing them a lot (sometimes fairly, sometimes not); because of that I think it is essential to highlight the ongoing discussion of what they are and what scientific insights we gained. And specifically, what the remaining challenges are - possibly hinting on your model? How is it different from all the literature that you are outlining?

**We appreciate the reviewer's constructive feedback regarding the framing we present in the introduction. While we agree with many of these comments, we do not feel the text should be converted to a table as this does not represent a comprehensive and detailed review of GHMs. Putting it into a table may give that impression.**

**We added a paragraph to the introduction discussing some recent model intercomparisons and evaluations (GHMs with a focus on human processes). We also added text on how WBM fits into the milieu of other GHMs.**

These are all questions that can be touched upon in the abstract.

Fig.1: This is very helpful. Could you add the timescales on which these fluxes and storages are simulated?

**We have expanded the caption to provide additional details including time scales and connecting the fluxes to the paper sections.**

Table 1: I think this can be moved to the supplement.

**We moved the table to Appendix 1.**

199: This documentation should be appended as supplemental material or uploaded somewhere to provide a doi. If the GitHub repository is lost, this link is not really helpful. This is also the case in various other places in the manuscript.

**Yes, we will include the GitHub documentation as part of the supplemental material.**

Fig3: The y-axis is different on the plots and thus confusing. Also, the quality does not seem to be high. Not much to see when zooming in.

**We adjusted the figures to a common y-axis range and improved the resolution.**

Please also add a comparison to other global models. If it performs worse, state why the model's unique features are still useful.

**We added a new table (Table 5) that compares WBM discharge estimates to those of other GHMs, and added a paragraph comparing WBM to other discharge estimates.**

Further, I was expecting to see something like Fig 6 and 5 here. Maybe move Fig. 3 to the supplement and refer to the result section.

**We've removed figure 3, since it can just be referenced. This section is about previously-published results, so Figs 5 and 6 must go later where we evaluate the new model.**

Fig.4: Please refer to Table 6.

**We now refer to the relevant table in the Fig 4 (which is now Figure 3) caption.**

Table6: Please add the model name. Is that the absolute difference to the simulation that you are showing? Or the absolute value? The description text is confusing on this matter.

**We added model names, and changed to table title to clarify that the values are absolute values (not differences).**

**Reviewer #2**

This manuscript presents the UNH Water Balance Model, a hydrology model that has been developed over several decades, but released publicly (open source code) for the first time.

The authors provide a literature review of the model history and evaluate the current model performance against river discharge observations and irrigation water supply requirements. The WBM performs better across North America and Europe in terms of discharge and irrigation water supply, but relatively poorly across Asia and portions of South America.

The manuscript then provides examples of regional simulations including the Indus River watershed as well as the Wyoming headwaters contributing to river flow to the Colorado, Colombia and Mississippi River. This provides an opportunity to demonstrate the novel water component tracking functionality that enables identification of source regions and source stocks for river discharge and irrigation water supply.

The tracking feature appears to be a significant advance in river flow diagnostics and is capable of determining source spatial regions, and source water components (precip, agriculture, groundwater). This should be a valuable tool for land management and government policy makers. Finally, the authors provide an overview of WBM run-time instructions describing the necessary input data and setup scripts to perform a simulation.

This reviewer was impressed with the breadth of the manuscript included 1) a literature performance review, 2) multi-domain simulations with emphasis on the new diagnostic tracking functionality and 3) an overview of a model setup.

**We would like to thank Reviewer #2 for their kind words and thorough commentary on the paper. It will significantly improve the paper.**

This reviewer would have liked much more discussion devoted to WBM performance.

**In response to this comment and others, we added two new evaluation metrics to our discharge evaluation. We also added a new table (Table 5) that compares WBM discharge to other models and published estimates.**

The WBM had a strong high bias in simulating global irrigation water supply as compared to other studies (Table 6). This apparently was caused by a systematic underestimation of discharge for the China/Asia region, but very little discussion (only a mention regarding better parameters are needed) was devoted to this topic. Whereas the authors provided a comparison against similar hydrology models in terms of simulated irrigation supply, no comparison was provided for discharge rates against other models for better context and perhaps lead to a discussion of what model components or parameters are most in need of improvements. This reviewer would have liked more justification or explanation to describe the skill of WBM such that a new user could avoid certain regions or pay special attention to parameters which are poorly constrained. Also an inclusion/reference of a model tutorial would be helpful for new users to begin interacting with the WBM.

**In response to this comment and others, we added two new evaluation metrics to our discharge evaluation. We also added a new table (Table 5) that compares WBM discharge to other models and published estimates.**

**The reviewer also suggested the development of a full tutorial. While we realize that this may indeed be helpful for some users, we would like to see if the instruction manual we prepared and released through our GitHub repository is found to be lacking by the community before we endeavor to invest in the development of a full tutorial, which we understand to be quite time intensive.**

Line 66: Very nice explanation of the value of this source component tracking feature in this paragraph.

**We thank the reviewer for this kind assessment.**

Line 85: Global models come an out-of-the-box setup of preferred sub-model structures and parameterizations. There is a table devoted to the key default parameters (Table 2), but no discussion of the optimization process that is required for regional simulations, or representation of the range of parameters to make these regional simulations perform well. The authors do present some discussion and results based upon the contribution of uncertainty due to the climate forcing (Figure 3), but missed an opportunity to discuss the contribution of parameter uncertainty in the manuscript.

**Several reviewers expressed an interest in seeing greater detail presented regarding calibration and parameterization strategies, and we would like to provide this general response. Any revision to our manuscript will provide additional detail needed to form a baseline understanding of the parameterization strategy for the model; however, we acknowledge at the outset of this response that there is to date untapped potential for more rigorous evaluation of uncertainty quantification using WBM.**

**First, following comments from Reviewers 2 and 5, we expanded Table 2 (which is now Table 1) to include a greater cross-section of parameters commonly adjusted in regional studies is appropriate. We note that this table will be redundant to the WBM_Usage_and_Input_Reference.xlsx spreadsheet on the WBM GitHub page; however, we agree that providing a subset of this reference within the manuscript will improve the readability. We note that to the extent possible, we have relied on structuring the model consistent with empirically meaningful parameters. As such, values presented in Table 1 will often reflect syntheses of field observations and uncertainty as characterized therein. Other model parameters are more synthetic and have less direct connection to field observation. Many of these parameters have been evaluated through calibration exercises over the years in studies summarized in Section (3.1). The reasonable ranges included in Table 1 are based on what the authors consider as appropriate starting points for parametric uncertainty analyses based on a combination of prior experience and physical meaning.**

**Previous work to calibrate WBM has generally leveraged manual calibration, with several instances of more rigorous calibration attempts. Parameterization of core WBM components evolved through iterative attempts to capture response in both global and regional contexts. Generally, it has been found that parameterization schemes as represented by the default parameter values in WBM_Usage_and_Input_Reference.xlsx reflect reasonable compromises that adequately represent discharge time-series in global simulations, as well as regional contexts focused on temperate, humid, and modestly developed watersheds. We now point that uniform parameterization can be applied to unique watersheds to capture non-default response (Samal et al. 2017, Zuidema et al. 2018), or that spatially varying parameterizations can capture more finely resolved nuance in watershed properties (Zuidema et al. 2020).**

How modular is WBM in terms of testing particular hypotheses about hydrology and competing methods, etc. ? There is some brief description at the very end of turning flags on/off, but no specifics in how this influences representation of hydrology. Given the source tracking capability it would be interesting to test the impact of certain model assumption/hypotheses.

**We agree with the reviewer that combined with WBM's source tracking capability, varying the processes represented by the model provide for powerful tools to address hypotheses in regional and global hydrology. Generally, this strategy underlies much of the development that has gone into recent work that leverages WBM, including Grogan et al. (2017), Zuidema et al. (2020), Rougé et al. (2021). We have added new optional functionality throughout the history of development in part because it captured important processes that improved representations. While we acknowledge that such experimentation makes for compelling analyses that show the importance of capturing anthropogenic processes in GHMs (Veldkamp et al. 2018), we consider repeating such experimentation here to be beyond the scope of this manuscript.**

Model Description:

Line 140: I am assuming the representation of snow is considered single-layer, and does not include multiple layers. Things like snow properties and albedo are not explicitly taken into account. Also insolation and aspect are not considered within the melting term? Care to comment how this might influence your snow source? Has this ever been validated against gridded snow data sets?

**The snow model is a simple single-layer approach with elevation bands providing within grid cell variability. We have added this detail to the snow model description. Grogan et al. (2020) corroborated daily model predictions of snow water equivalent to 1034 observation points in the US Northeast, which is described in the Evaluation section.**

Line 200: "Actual evapotranspiration (AET) from naturally vegetated land areas is a function of the PET, soil moisture, and soil properties."

So rooting depth is not taken into account? What function or purpose is setting the soil moisture pool depth to that of the rooting depth? I assume it's a single layer soil subsurface then?

**We added details of the soil water balance calculation to the manuscript.**

**AWC is a difference between soil field capacity (porosity that can hold water) and wilting point (minimal porosity that plants cannot extract water from). Because it is a relative metric (m/m), it needs to be multiplied by the rooting depth to have it in mm.**

Line 210: "While AET from other land cover types (e.g., forest or grassland) can be parameterized and simulated, no published study has yet used this option of WBM. Actual evapotranspiration from other consumptive water uses are described below in Section 2.2.5."

Are the default parameters provided for forest/grassland types to calculate AET, or does the user have to provide them? If this option has never been used, how does the model treat forest/grassland if run globally – which includes forest and grassland cover?

**WBM parameters that control all hydrological cycle processes have default values that a user can set and overwrite. This public version of WBM has over 250 of them (see WBM_Usage_and_Input_Reference.xlsx in the GitHub repository). Regarding this particular comment, default parameters that control AET were inherited from Federer et al. (2003). Specifically it includes a flag from the Hamon PET model, and optimized soil drying parameters for the best estimate of global river runoff and discharge. The default AET settings do not use specific types of land cover and treat land as a generic land cover type, but the user can optionally set AET parameters for each land cover type. Using default settings in WBM is equivalent to treating the land-surface as covered by a reference vegetation (Allen et al. 1996).**

Line 275: The term "Shallow groundwater pool" is not used in Figure 1. I assume this is the same thing as "groundwater recharge pool"? If so, make sure the terminology is consistent.

**We have changed this to consistently refer to the shallow groundwater pool.**

Do the grid cells communicate for surface runoff and subsurface discharge or does this get routed directly to river transport model?

**WBM assumes that surface runoff and "subsurface discharge", which we call baseflow, do not interact on the landscape, but both contribute to river flow as independent flow-paths. We made this distinction clearer in the runoff description.**

The term "unlimited unsustainable groundwater source" seems confusing. Is there a better way to describe this fossil ground water?

**From the model perspective the water users are tapping into a pool of water that is not explicitly recharged and therefore it is unsustainable, even though this pool is not finite. The concept and the term "unsustainable" has been used in our previous published work (Grogan et al., 2017; Liu et al., 2017).**

**Several terms for this concept are used in the literature depending on the application, and we make particular reference to the useful table of definitions in Bierkens and Wada (2019). These different terms have been acknowledged in the text (starting on line 289). We selected our terminology as an adaptation to the "physically non-sustainable groundwater use or groundwater depletion" defined by Bierkens and Wada (2019), and our conceptualization is consistent with that use. Because we define a flux of water that satisfies demand unmet by local resources (i.e. non-sustainable groundwater use), this implies a pool of water from which this water is drawn, which we do not represent explicitly, that is best identified in our use as a tracking descriptor as "non-sustainable groundwater". We adopt "unsustainable" as equivalent to "non-sustainable" because the word "unsustainable" is defined in both the Merriam-Webster and Cambridge dictionaries and "non-sustainable" is not. While we do agree with the reviewer that it is the flux or the use of groundwater that is unsustainable, it is illogical to describe subsequent fates or flow paths abstracted unsustainably as "use". For instance, in describing the fractions of primary water components that drain to the ocean, it is awkward to describe a fraction of this discharge as "unsustainable groundwater use", when we are clearly treating it as representing a fraction of flow within the river system.**

**We are also unsatisfied with the alternative terminology posed (fossil groundwater) by the reviewer. Again, following definitions from Bierkens and Wada (2019), some terms imply knowledge of the age of recharge to the aquifer that we do not characterize in typical WBM simulations (e.g. recharging 12,000 years before present (Jasechko et al. 2017) in the case of fossil groundwater). Therefore, we prefer terminology that implies no assumptions of the era of recharge.**

**We acknowledge that the terminology that we have been using in this and our prior papers is not ideal, though it has been deliberated extensively. However, we view it as satisfactory for the time-being and for the purposes we are using it. We added text to clarify our definition of unsustainable groundwater in the manuscript.**

Line 550-555: Here you mention spin up time in the context of water source tracking, but I feel this discussion could come much earlier when describing the model dynamics and features themselves.

**We added spinup time details to the General Overview (Section 2.1).**

Table 4: Please define 'relict' here.

**We added the definition of this term to Table 3, and to Section 2.2.7.**

Because the model is becoming open source, the presumption is to allow for a wider user community. Do you provide a tutorial to familiarize the user with WBM? The authors provide a four-step description towards the end of the manuscript, plus a reference to an instruction manual, but a tutorial would be a great advance.

**There is an instruction manual with a quickstart guide in the GitHub repository. We'd like to write a tutorial in the future, and will do so if it fits with any of our upcoming funded projects.**

Model Validation:

Line 620:    I recognize this section is devoted to a summary of WBM literature, but it is difficult to evaluate the skill of the model without the context of comparing against other similar hydrology models.  There must be some model hydrology intercomparison studies to show here for global river discharge.  Certainly Figure 3 might benefit for a comparison against other models.

**We added a new table (Table 5) that compares WBM discharge estimates to other GHMs, along with a paragraph describing the findings. We also added 2 new validation metrics to the evaluation section.**

Line 624:  What sort of parameter calibration is performed here? Hand tuning, or more formal DA approaches? Are these parameters available for the user?

**Throughout the development history of WBM, default values for most parameters have been established through a combination of manual calibration, pre-calibration approaches such as Generalized Likelihood Uncertainty Estimation, and from literature review of applicable properties.  The parameter set used here reflects those default values, and no separate calibration was performed for this manuscript.**

**We revised Table 2 (which is now Table 1) to include more parameters, better descriptions of those parameters, and reasonable ranges for parameter values.**

**For a complete listing of parameters, their units, complete descriptions of their behavior and interaction with the model, the reviewer is referred to the WBM_Usage_and_Input_Reference.xlsx spreadsheet on the WBM GitHub page.**

Section 3.2.1:    Table 2 is good, but a physical definition for the parameters should be stated in the table and not just in text. Perhaps a summary table of parameter values for the literature review manuscripts performed at different regions/resolutions, in addition to the default values.

**We revised Table 2 (which is now Table 1) to include more parameters, better descriptions of those parameters, and reasonable ranges for parameter values.**

Line 690: "Above, we reviewed previously-published WBM validations. As none of the prior versions of WBM code have been released open source, it is important to validate the exact model structure in this first open-source release. "

Was there significant mechanistic changes to these pre-release versions? Was adding the tracking capability the only significant difference from this official release version? From Table 7 it seems like you add some new functionality from previous versions:  "Added rainfed agriculture,

other land cover types, inter-basin transfers, domestic and livestock water demand",  but you don't mention that here, and it seems to the reader that the only change is making it open source, when there are some structural changes.

**Yes, there were 'significant mechanistic changes' compared to many of the prior studies reviewed in Section 3.1.  Many of these studies targeted specific research questions requiring new, unique, and narrowly applicable process representation.  The open-source release presented here, represents a core trunk of process representation that we consider is widely applicable by itself, and is a useful starting point or baseline from which deviations in the code needed to accommodate future unique process representation can be described.  Therefore, the most significant mechanistic changes are the removal of processes that we consider one-off or too narrowly corroborated by empirical data to distribute as having validity over global scope.**

**We added WBM v.1.0.0 as a separate row to Table 7, with an entry that reflects the relationship between the released version and prior work under the name "WBM".**

Table 6:    Appreciated how the WBM irrigation withdrawal estimate was put in the context of other studies.   Would it be possible to construct something similar for global discharge?   Especially since the author attributes the high irrigation withdrawal estimate (China) to relatively high discharge rates in the Asia domain, it seems like it would be worthwhile to hone in on discharge biases, and diagnose where and what location these are occurring.

**We added a table comparing global WBM discharge to other estimates. The MBE maps in Figs 4 and 5 show the discharge issues in Asia. We also added a brief description of irrigation water withdrawal sensitivity to climate inputs in Asia and China specifically.**

Line 800: "Global discharge is dominated by rain over most of the globe, with snowmelt an important contributor at the poles, and both glacier runoff and unsustainable groundwater important regionally."

"Snowmelt is an important contributor at the poles":  That seems like an oversimplification.   Seems that Figure 8 shows there is significant snowmelt contributions well down into the northern mid-latitudes especially for mountainous terrain such as the Rocky Mountains and the Himalayas where a larger population rely on snow runoff for irrigation etc.   This should be mentioned.

**Thank you for this comment.  We modified the text in this section to read as follows:**

**Global discharge is dominated by rain over most of the globe, with snowmelt an important contributor at high latitudes and high altitudes, with both glacier and unsustainable groundwater important regionally.**

Figure 8:  Glacier run-off seems to be unrealistic in the SouthWest and MidWest US where none should be occurring.   Glaciers are apparently determined solely by snow input and melting algorithm and not prescribed like land cover types?

The Glacier Runoff panel in Figure 8 shows the fraction of glacier water found in river discharge.  There are glaciers found in the Colorado and Mississippi Rivers and the meltwater travels downstream.  That water is used for irrigation and there will be non-zero fractions of glacier water found throughout these regions.

Figure 11:   Perhaps outline the watershed domain of the Indus river in this figure within the larger figure 10 for better reference and perspective.

**The Indus river basin outline has been added to Figure 9 (which used to be Figure 10).**

Figure 11b:   There are no units in this figure.  Why is there such a large discontinuity between months 12 and month 1?    Why does glacier runoff have such a strong contribution upstream at point A, yet such a small contribution downstream at point B?

**We added units.**

**The plot reflects the large differences in flow between the months of December and January.  We note that there are similarly large differences in flow between the months of November and December.  Most of the water at the river mouth during December reflects negligible water withdrawals upstream for irrigation as there is no or little irrigation in the month of December.**

**Point A is located in the foothills of the Himalayas and a large portion of the basin upstream is covered by glacier. Downstream of Point A is the Indo-Gangetic Plain, which contains extensive irrigated agriculture, and large amounts of the glacier water are lost to evapotranspiration. The basin outlet (Point B) includes drainage from several eastern tributaries originating in India.  These tributaries are supported predominately by rain water, and from extraction of unsustainable groundwater.**

4.2 Return flow tracking

Line 867:   I feel like this "relict" and "pristine" distinction should be made earlier, because it is used in an earlier table.

**We added the definition to Table 3, and give the definition earlier now, in Section 2.2.7.**

Return water diagnostics and source water diagnostics and tracking seem to be some of the most useful components of the system.   Would it be easy to port these 'diagnostics' modules to other hydrology models or is it baked into the system?

**The code is deeply integrated into the model, however, the concepts and techniques could be adopted by other modeling groups and are very similar to the techniques used in HBV (Stahl et al. 2017, Weiler et al. 2018).**

Figure 12 and 13:   Care to comment on why the irrigation return flow waters are so high in the Northern India region?  It might help the reader contextualize this diagnostic,  and make for interesting discussion of what makes that region so unique.

**We added the following text to the Figure 11 (which used to be Figure 12) caption:**

**The large fraction of irrigation return flows in northern India and Pakistan results from this region being one of the most intensively irrigated regions in the world, combined with very low (~30%) classical irrigation efficiency (Zaveri et al., 2016; Grogan et al., 2017). This combination means that large volumes are water are extracted for irrigation, and nearly two-thirds of that water returns to the system.**

Figure 14:  These plots are interesting and potentially very valuable. Couple questions, in Figure 14b could the color scale be changed to emphasize the 0 to 0.2 discharge fraction?  A large part of the watershed domain seems to be the same color of green that is hard to distinguish at all.

Could you not color Wyoming such that the river network can be seen easily? It doesn't seem to be a need to color Wyoming dark blue since it is the headwater region.   Also in Figure 14a is this distribution pattern mostly driven by snowmelt?  It doesn't look like the distribution is picking up the summer monsoon season where a larger fraction of the water should derive from Arizona and New Mexico. I suppose this depends if the climate forcing captures the SouthWest monsoon.

**We did not color Wyoming separately from the rest of the grid cells.  All small rivers in Wyoming have 100% discharge originating in the state and therefore they are colored in the darkest blue.  In a few cases where rivers enter into Wyoming, such as the North Platte River, they reduce the fraction of Wyoming-sourced water and lighten the color.  As more Wyoming-sourced water is added to those rivers, the colors will get progressively darker.**

**We changed the color palette to better show the low values, outlined WY in gray, and added more detailed description to the figure caption.**

Regarding Figure 14a, yes, the large peak in Fig. 14a is a result of snowmelt.

Model Code development section:   Although I do find this section interesting it may be better suited for the appendix, such that the reader can immediately go from the results to the discussion.  You have provided steps, and there is access to an instruction manual, but do you also provide a simple user tutorial for a cut down domain and provide the input files so the user can familiarize themselves with the steps?

**Another reviewer expressed appreciation that this section was included.  Therefore, we will leave it where it is.  The tutorial request was discussed above.**

Discussion:

The tracking capabilities of this model which can attribute discharge source regions, or impact from agriculture on discharge are a great feature and worthy of discussion.  The authors refer to the modular nature of the model in order to toggle on/off different features, but they don't

provide a test case of this, where for a single experiment, different model structures/assumptions/parameters are switched up to identify the model sensitivity. Just a suggestion.

**We do provide a variety of simulation runs showing the breadth of the model's capabilities, and more detailed analyses can be found in the manuscripts cited that make specific use of this functionality (Grogan et al. 2017, Zuidema et al. 2020). We feel that any additional sensitivity analyses using tracking functionality would detract from the focus of documenting the WBM v1.0.0 model, and would be best suited to separate future work allowing us to keep the manuscript to a manageable length.**

I was certainly expecting at least more discussion concerning model performance related to river discharge and irrigation water withdrawal as covered in the first part of the manuscript. It was a bit concerning that the WBM was a bit of an outlier when estimating global irrigation water withdrawal, and this was partially attributed by the authors to low discharge rates across China and Asia. Very little to no discussion or explanation was provided for this. The discharge rates seemed to perform relatively well in other regions includes North America, but it was difficult to contextualize given no comparison in performance was provided from other hydrology models.

**In response to this comment and others, we added two new evaluation metrics to our discharge evaluation. We also added a new table (Table 5) that compares WBM discharge to other models and published estimates, and a paragraph which describes these new results.**

**References:**

**Allen, R.G., Pereira, L.S., Raes, D., Smith, M., others, 1998. Crop evapotranspiration-Guidelines for computing crop water requirements-FAO Irrigation and drainage paper 56. FAO, Rome 300.**

**Bierkens, M. F., and Wada, Y. (2019). Non-renewable groundwater use and groundwater depletion: a review. Environmental Research Letters, 14(6), 063002.**

**Federer, C. A., Vörösmarty, C., & Fekete, B. (2003). Sensitivity of annual evaporation to soil and root properties in two models of contrasting complexity. Journal of Hydrometeorology, 4(6), 1276-1290.**

**Grogan, D. S., Wisser, D., Prusevich, A., Lammers, R. B., and Frolking, S.: The use and re-use of unsustainable groundwater for irrigation: a global budget, Environ. Res. Lett., 12, 034017, https://doi.org/10.1088/1748-9326/aa5fb2, 2017.**

**Grogan, D. S., Burakowski, E. A., and Contosta, A. R.: Snowmelt control on spring hydrology declines as the vernal window lengthens, Environ. Res. Lett., 15, 114040, https://doi.org/10.1088/1748-9326/abbd00, 2020.**

Jasechko, S., Perrone, D., Befus, K. et al. Global aquifers dominated by fossil groundwaters but wells vulnerable to modern contamination. Nature Geosci 10, 425–429 (2017). https://doi.org/10.1038/ngeo2943

Liu, J., Hertel, T. W., Lammers, R. B., Prusevich, A., Baldos, U. L. C., Grogan, D. S., and Frolking, S.: Achieving sustainable irrigation water withdrawals: global impacts on food security and land use, Environ. Res. Lett., 12, 104009, https://doi.org/10.1088/1748-9326/aa88db, 2017.

Rougé, C., Reed, P., Grogan, D., Zuidema, S., Prusevich, A., Glidden, S., Lamontagne, J., and Lammers, R.: Coordination and Control: Limits in Standard Representations of Multi-Reservoir Operations in Hydrological Modeling, https://doi.org/10.5194/hess-2019-589, In review.

Samal, N. R., Wollheim, W., Zuidema, S., Stewart, R., Zhou, Z., Mineau, M., Borsuk, M., Gardner, K., Glidden, S., Huang, T., Lutz, D., Mavrommati, G., Thorn, A., Wake, C., and Huber, M.: A coupled terrestrial and aquatic biogeophysical model of the Upper Merrimack River watershed, New Hampshire, to inform ecosystem services evaluation and management under climate and land-cover change, 22, 18, https://doi.org/10.5751/ES-09662-220418, 2017.

Stahl, K., Weiler, M., Freudiger, D., Kohn, I., Seibert, J., Vis, M., Gerlinger, K., and Böhm, M.: Final report to the International Commission for the Hydrology of the Rhine basin (CHR), 146, 2017.

Sutanudjaja, E. H., van Beek, R., Wanders, N., Wada, Y., Bosmans, J. H. C., Drost, N., van der Ent, R. J., de Graaf, I. E. M., Hoch, J. M., de Jong, K., Karssenberg, D., López López, P., Peßenteiner, S., Schmitz, O., Straatsma, M. W., Vannametee, E., Wisser, D., and Bierkens, M. F. P.: PCR-GLOBWB 2: a 5-arcmin global hydrological and water resources model, 11, 2429–2453, https://doi.org/10.5194/gmd-11-2429-2018, 2018.

Veldkamp, T. I. E., Zhao, F., Ward, P. J., de Moel, H., Aerts, J. C. J. H., Müller Schmied, H., Portmann, F. T., Masaki, Y., Pokhrel, Y., Liu, X., Satoh, Y., Gerten, D., Gosling, S. N., Zaherpour, J., and Wada, Y.: Human impact parameterizations in global hydrological models improve estimates of monthly discharges and hydrological extremes: a multi-model validation study, Environ. Res. Lett., 13, 055008, https://doi.org/10.1088/1748-9326/aab96f, 2018.

Weiler, M., Seibert, J., and Stahl, K.: Magic components—why quantifying rain, snowmelt, and icemelt in river discharge is not easy, 32, 160–166, https://doi.org/10.1002/hyp.11361, 2018.

Zaveri, E., Grogan, D. S., Fisher-Vanden, K., Frolking, S., Lammers, R. B., Wrenn, D. H., Prusevich, A., and Nicholas, R. E.: Invisible water, visible impact: groundwater use and Indian agriculture under climate change, Environ. Res. Lett., 11, 084005, https://doi.org/10.1088/1748-9326/11/8/084005, 2016.

Zuidema, S., Wollheim, W., Mineau, M. M., Green, M. B., and Stewart, R. J.: Controls of Chloride Loading and Impairment at the River Network Scale in New England, 47, 839–847, https://doi.org/10.2134/jeq2017.11.0418, 2018.

Zuidema, S., Grogan, D., Prusevich, A., Lammers, R., Gilmore, S., and Williams, P.: Interplay of changing irrigation technologies and water reuse: example from the upper Snake River basin, Idaho, USA, 24, 5231–5249, https://doi.org/10.5194/hess-24-5231-2020, 2020.

**Reviewer #3**

General comments

The paper describes the WBM global hydrological model in detail. WBM is one of the earliest global hydrological models which contributed to the establishing the field of global hydrology. This paper provides the full description of the model together with the development history which will be quite useful for the modeling community. In particular, the water source tracking function is novel and very interesting. The paper is well prepared and mostly very readable. I have only minor technical comments.

**We would like to thank Reviewer #3 for their time in providing thoughtful comments on this paper.**

Specific comments

Line 189 "Soil moisture balance calculations for natural landcovers are fully described in (Wisser et al., 2010a) and crop landcovers in (Grogan, 2016).": Better to show the essence here because soil moisture balance calculation is the most fundamental function of any hydrological models.

**We added a description of the soil moisture balance calculations.**

Line 270 "PyGEM's standard output format is not gridded; rather, post-processed PyGEM output is required as input for WBM (Prusevich, et al., 2021).": How frequently is the glacier fraction updated (e.g. daily, monthly, annually)?

**Glacier water (glacier runoff) is updated at monthly timesteps in the model in accordance with the source PyGEM glacier point data. Some PyGEM variables such as glacier volume and area are updated at annual timesteps so WBM also updates those layers at the annual timestep. We added the timestep information to the manuscript text.**

Line 358 "Rather, they collect rainwater and surface runoff, storing it on the land surface and preventing it from reaching the rivers system": How are these processes formulated? What are the key inputs and parameters?

**Greater detail regarding how these small reservoirs are handled is available in the technical documentation that we now include as a supplement to the manuscript.**

Line 368 "WBM's inter-basin transfer methods were first developed and described in (Zaveri et al., 2016) and described again in (Liu et al., 2017).": Can this inter-basin transfer scheme be applied to global simulations? If so, how the parameters were set (i.e. is such information available)?

**An IBT database was developed for specific publications (e.g. Zaveri et al., 2016), however a global version has not been released. This functionality is available to users if they**

**develop tables representing inter-basin transfers with further details available in the technical documentation.**

Line 400 "Stream water available for extraction is estimated as 80% of water retained in river and reservoir storage following routing during the previous time-step $W^{k-1}$, plus the volume, Vstream, represented by flow through the reach during the previous time-step:" A bit hard to read and associate with Equation 26. What is Vstream? Is this representing the available surface water?

**We re-wrote this section to clarify variable definitions.**

Line 621 "The global simulations described above used a grid cell resolution of 0.5 degrees.": This should be mentioned in the previous paragraph.

**We moved the resolution to the previous paragraph.**

Line 629 "These continental-scale simulations of India used the same 0.5 degree spatial resolution as the global simulations.": What were the input meteorological data used in these simulations? The performance of river discharge simulation is largely dependent on the quality of input meteorological data (e.g. Hanasaki et al. 2022, HESS).

**The Zaveri et al (2016) paper used the Asia-specific APHRODITE climate drivers. We added this info to the text.**

Line 721 "We also calculate the Index of Agreement, d, (Willmott, 1981)": Why was this indicator chosen? I recall that most of the earlier works used NSE.

**We added NSE and KGE to our evaluation to make these results comparable to previous studies.**

Line 740 "Despite the global average good agreement, there is significant spatial variability, with lower MBE values across much of South America and East Asia (Figs. 5c and 6c).": When one looks at the absolute MBE, the performance of river discharge simulations in arid or semi-arid regions always appears to be "good" because the runoff is very small. This needs to be pointed out in the text.

**Thank you for pointing this out. We added this point to the text.**

Line 750 Figure 6: What is the difference between Figure 5 (c) and 6 (c)? Only the unit is different?

**Yes, the figures are showing the difference between daily and monthly metrics as identified by both the units, and in the captions of the two figures. We include both as many researchers running global simulations use monthly outputs in their papers. We also include daily metrics as the model runs at this time step.**

**References:**

Gupta, H.V., Kling, H., Yilmaz, K.K., Martinez, G.F., 2009. Decomposition of the mean squared error and NSE performance criteria: Implications for improving hydrological modelling. Journal of Hydrology 377, 80–91. https://doi.org/10.1016/j.jhydrol.2009.08.003

Hanasaki, N., Matsuda, H., Fujiwara, M., Hirabayashi, Y., Seto, S., Kanae, S., and Oki, T.: Toward hyper-resolution global hydrological models including human activities: application to Kyushu Island, Japan, Hydrol. Earth Syst. Sci, https://doi.org/10.5194/hess-2021-484, accepted, 2022.

Knoben, W.J.M., Freer, J.E., Woods, R.A., 2019. Technical note: Inherent benchmark or not? Comparing Nash–Sutcliffe and Kling–Gupta efficiency scores. Hydrology and Earth System Sciences 23, 4323–4331. https://doi.org/10.5194/hess-23-4323-2019

Krause, P., Boyle, D.P., Bäse, F., 2005. Comparison of different efficiency criteria for hydrological model assessment. Adv. Geosci. 5, 89–97. https://doi.org/10.5194/adgeo-5-89-2005

Zaveri, E., Grogan, D. S., Fisher-Vanden, K., Frolking, S., Lammers, R. B., Wrenn, D. H., Prusevich, A., and Nicholas, R. E.: Invisible water, visible impact: groundwater use and Indian agriculture under climate change, Environ. Res. Lett., 11, 084005, https://doi.org/10.1088/1748-9326/11/8/084005, 2016.

Reviewer #4

General comments

The paper "WBA: A scalable gridded global hydrologic model with water tracking functionality", by Grogan et al. provides a description of the first open source version of the University of New Hampshire Water Balance Model. The authors chose an approach that combines parts of a "classical" model description – i.e. description of the functionalities and fundamental equations, validation and a selection of case studies – with a literature review of the history of the model, previous studies with WBA and validation of previous model versions. Overall, this structure works really well and the paper is well written, thus, I only have a few minor suggestions.

**We thank the reviewer for taking the time to carefully read and review this paper.**

Minor comments

1) With respect to the model description, the authors did a very good job at providing a general overview over the basic equations and dependencies in the model without overloading the manuscript with technical details (which is perfectly reasonable given that most WBA components have been used in previous studies and have been well documented). However, it would be extremely helpful if the structure of the section 2.2 could be related to what is shown in figure 1, i.e. that all elements that are shown in the figure are discussed in the model description, preferably even in a way that each element in the figure has a subheading in the text.

**We edited Figure 1 to use the same names for fluxes and stocks as section 2.2, and added more description to the caption.**

2) I find the use of the term "unsustainable ground water" somewhat problematic, since it is not the groundwater itself that is unsustainable but its use e.g. for irrigation. A term that clearly states either which real world pool is represented – e.g. fossil water – or what it constitutes in the model – namely an unlimited water supply to balance demand-supply mismatches – would be more appropriate. Maybe the authors could also add some discussion to section 2.2.2, detailing how the use of this pool affects simulations (especially projections) in regions where fossil ground water is being depleted.

**From the model perspective the water users are tapping into a pool of water that is not explicitly recharged and therefore it is unsustainable, even though this pool is not finite. The concept and the term "unsustainable" has been used in our published papers (e.g. Grogan et al., 2017; Liu et al., 2017).**

**Several terms for this concept are used in the literature depending on the application, and we make particular reference to the useful table of definitions in Bierkens and Wada (2019). These different terms have been acknowledged in the text (starting on line 289). We selected our terminology as an adaptation to the "physically non-sustainable groundwater use or groundwater depletion" defined by Bierkens and Wada (2019), and**

**our conceptualization is consistent with that use. Because we define a flux of water that satisfies demand unmet by local resources (i.e. non-sustainable groundwater use), this implies a pool of water from which this water is drawn, which we do not represent explicitly, that is best identified in our use as a tracking descriptor as "non-sustainable groundwater". We adopt "unsustainable" as equivalent to "non-sustainable" because the word "unsustainable" is defined in both the Merriam-Webster and Cambridge dictionaries and "non-sustainable" is not. While we do agree with the reviewer that it is the flux or the use of groundwater that is unsustainable, it is illogical to describe subsequent fates or flow paths abstracted unsustainably as "use". For instance, in describing the fractions of primary water components that drain to the ocean, it is awkward to describe a fraction of this discharge as "unsustainable groundwater use", when we are clearly treating it as representing a fraction of flow within the river system.**

**We are also unsatisfied with the alternative terminology posed (fossil groundwater) by the reviewer. Again, following definitions from Bierkens and Wada (2019), some terms imply knowledge of the age of recharge to the aquifer that we do not characterize in typical WBM simulations (e.g. recharging 12,000 years before present (Jasechko et al. 2017) in the case of fossil groundwater). Therefore, we prefer terminology that implies no assumptions of the era of recharge.**

**We acknowledge that the terminology that we have been using in this and our prior papers is not ideal, though it has been deliberated extensively. However, we view it as satisfactory for the time-being and for the purposes we are using it. We added text to clarify our definition of unsustainable groundwater in the manuscript.**

3) I think it is a good idea to discuss existing model validation in this paper, rather than repeating the respective simulations with the present model version. However, it would be helpful if the authors could detail if and how the present model version differs to the model versions used in the previous studies and how these differences affect the results. Furthermore, with respect to the FrAMES model (component) I find the model validation a bit out of place here, as it is not merely a different version of WBA but a completely different model. I would expect the performance of the implemented functionalities to depend on many other aspects of the model and the forcing data, hence different simulated nitrogen concentrations with WBA. I think it would be sufficient to state, in the model description sec. 2.2.6, that WBA now includes these functionalities based on the parametrizations of the FrAMES model and reference the studies in which FrAMES was validated. However, I think it would be even better if the authors could actually perform the analysis and validate N and temperature in WBA simulations.

**The reviewer requested that we provide details of how the current version of the model differs from prior versions of the model. Table 7 describes the evolution of the WBM family of models; we added a new line for WBM v.1.0.0 to this table to clarify the open source version presented here as separate from previous work. The reviewer also asked us to comment on how differences in the current model and prior implementations of the model could affect results; however, we view this as an impossible request. Each study referenced in this section had specific motivations, input data, and simulated processes.**

**The older versions of the model, WBMplus and FrAMES, are closely related to the model version described in this paper. Despite the different name, FrAMES is not a completely different model, but it was built from WBMplus with the ability to simulate water quality (including temperature) incorporated into the code. We added a statement describing this connection to the text.**

**We would also like to respond to the reviewer's comment or request to repeat simulations of temperature or dissolved inorganic nitrogen (DIN) with WBM v1.0.0. DIN functionality within WBM is limited to regions with ample observational data. It is beyond the scope of this manuscript to apply this functionality to the two simulation domains presented.**

Specific comments

Line 22: " ... as well as perform model experiments in new ways". Please, clarify which are these new ways.

**We revised the abstract to be more specific.**

Lines 55 ff: Evapotranspiration will eventually lead to precipitation and a large fraction of the respective water is even recycled locally. Thus, the statement that only 50% of water is returned to "the system" is somewhat misleading. In contrast, when talking about specific pools in the system a 50% return rate is also often questionable, e.g. in case of fossil water, at least on a centennial timescale.

**Without an atmospheric model WBM is not able to assess the degree of local recycling of evaporate. We added a statement to the text clarifying that WBM tracks land surface hydrologic fluxes only, not atmospheric water.**

Fig. 1: Would it be possible to make the elements of the figure consistent with subheadings in section 2.2.? For example infiltration is not specifically discussed in the text.

**Yes, we edited the figure to reflect the terms used in section 2.2.**

Line 166: Eow is not defined.

**Thank you for catching that oversight. $E_{ow}$ is open water evaporation expressed in mm/day, which we added to the text.**

Line 170: "Storm runoff" is this the same as "stormwater runoff"?

**The term "storm runoff" provides a useful semantic distinction from "stormwater runoff". We view the collective understanding of stormwater runoff to refer to immediate runoff from impervious surfaces connected directly to water bodies via built infrastructure. While we include such a flux in WBM, we add two other fluxes to this variable in our internal calculations: 1) precipitation incident directly to open water within each pixel, and 2) water that overfills the surface flow pool ($R_{exc}$, Equation 13).**

Line 179: What does WBA do in these grid cells e.g. in case of endorheic basins?

**Endorheic basins are treated the same as ocean grid cells. We added a description of this method to the text.**

Line 183 f: Is this the only limit on infiltration? Is the state of the soil not taken into account?

**Yes, we do not simulate explicit Hortonian runoff. In the case that the soil is already saturated, any throughfall will be split between the quickflow and recharge flow paths. This may present a localized low bias in runoff for extreme precipitation events, but we consider it a suitable simplification for a macro-scale model.**

Line 190: It may be helpful to mention that WBA does not have soil layers and does not explicitly represent the vertical flux through the soil or a soil moisture profile.

**WBM has a single soil layer. Section 2.2.1 Land surface fluxes discusses the soil layer (defined by rooting depth on line 187) and section 2.2.2 Groundwater discusses the below-soil storage pool (shallow groundwater storage pool). We changed the labelling in Figure 1 to reflect these terms more. We also added a description of soil water balance within the root zone, and added text stating that the soil is only one layer.**

Line 257: How do you justify this default value of 1000 mm, i.e. that the model, in the default mode, has no real limit to the surface storage?

**You are correct, the default behavior is to effectively turn the process off. This default keeps model behavior more in line with prior model usage (from early Vorosmarty papers through Grogan et al. 2017), until such time that we have experience with the parameter to provide better direction to users.**

Line 280: I am a bit confused by the unit l/d is that per m^2?

**The shallow groundwater pool stores space-averaged groundwater in units of mm. The time-constant drains water from the pool with units of 1/d, yielding a flux to the stream in units of mm/day. We added a statement to the text to clarify these units, and added Eq. 22, which makes clear how the two units are multiplied together to result in a rate of mm/day.**

Line 490 ff & 503 ff: Is there a lag connected to the return flows?

**There is no lag; we add this statement to the text.**

Line 564: I am not familiar with the term "relic water", so I am not sure whether some definition is necessary.

**We added the definition to Table 3, and give the definition earlier now, in Section 2.2.7.**

Fig. 2: What is the meaning of the colors? Also why is the down-stream cycle different (no subheadings in "sources" and "water")?

**We revised the figure and caption to be clearer.**

Section 3.1: Could you maybe add a table for a quick overview?

**While we acknowledge that there are numerous numeric values and citations presented in these paragraphs, we have reservations about presenting this information as a table as. Much of the nuance and critical aspects of the findings of these studies are presented in the text, and the values are properly contextualized with the text.**

Line 611: What about the UDEL climate? In Fig. 3 The R2 looked very promising?

**Thank you for pointing out that the UDEL climate was not mentioned. We corrected that in the text. Also, Figure 3 was removed as it can just be cited.**

Fig. 4: Maybe use the same axis for subfigure b and c?

**We revised the figure to use the same axis**

Line 671: While I think it's a good way to use existing validation, I am not sure about the FrAMES model, as the respective formulations lead to a very different outcome in the WBA framework?

**Despite the different name, FrAMES is not a completely different model, but it was built from WBMplus with the ability to simulate water quality (including temperature) incorporated into the code. We added a statement describing this connection to the text.**

Line 725: Could you also include R2 to make it easier to compare the present simulations to those in section 3.1 ?

**We find that R2 values are not a good metric for evaluating GHMs, despite their previous use. These values will always be pretty good because large basins will have high runoff, and small basins will have small runoff. Here we used the MBE and Index of Agreement, and further added the NSE and KGE, which are more commonly accepted GHM evaluation metrics.**

Line 725: I would be very curious if you could also include an evaluation of the simulated evapotranspiration … maybe against GLEAM data?

**Evapotranspiration data is sparse and applies to scales much finer than resolved by the large grid cells used in this paper (5 minute cells). The GLEAM data (which is model output) uses a Priestley-Taylor approach and therefore any evaluation would be a comparison between the two PET functions. A comprehensive PET comparison in the WBM context can be found in Vorosmarty et al. (1998).**

Fig. 5 & 6: Why is there no Index for the Nile/Indus/Ganges in subfigure c ? Also, would a relative measure make more sense than MBE.

**We used Global Runoff Data Centre (GRDC) river flow stations for these maps, and applied selection criteria to ensure a minimum number of observations from 2000 – 2009. While there is some historical runoff data for the Nile, Indus, and Ganges basins, it is all from years prior to 2000, and so was not used here.**

**We added 2 metrics (NSE and KGE) to our evaluation. Also, we add a statement in the text informing readers that the absolute values of MBE will always be small in in low-runoff regions, and direct them to look at the unitless metrics NSE and KGE.**

Fig. 8 & 9: Could you do such a figure also for evapotranspiration?

**Unfortunately, our tracking system for evapotranspiration is not sufficiently complete to provide equivalent panels to Fig. 8 and 9.  We do thank the reviewer for pointing this out, and we have added this to our model development to do list.**

**The model can provide the tracking data to make such a figure for irrigated crop ET, but it will largely follow the pattern of the existing figures.**

Fig. 10: I find the purple and blue colors are very similar, and I am not sure that its only an issue related to my printer.

**We separated the components into panels (a-c) to make it easier for all readers to see the color scales and differences.**

Line 906 ff: "… published in (Vörösmarty et al., 1989)". I would not use the brackets here.

**Corrected**

**References**

**Bierkens, M. F., and Wada, Y. (2019). Non-renewable groundwater use and groundwater depletion: a review. Environmental Research Letters, 14(6), 063002.**

**Jasechko, S., Perrone, D., Befus, K. et al. Global aquifers dominated by fossil groundwaters but wells vulnerable to modern contamination. Nature Geosci 10, 425–429 (2017). https://doi.org/10.1038/ngeo2943**

**Grogan, D. S., Wisser, D., Prusevich, A., Lammers, R. B., and Frolking, S.: The use and re-use of unsustainable groundwater for irrigation: a global budget, Environ. Res. Lett., 12, 034017, https://doi.org/10.1088/1748-9326/aa5fb2, 2017.**

**Liu, J., Hertel, T. W., Lammers, R. B., Prusevich, A., Baldos, U. L. C., Grogan, D. S., and Frolking, S.: Achieving sustainable irrigation water withdrawals: global impacts on food**

security and land use, Environ. Res. Lett., 12, 104009, https://doi.org/10.1088/1748-9326/aa88db, 2017.

Vörösmarty, C. J. and Iii, B. M.: Modeling basin-scale hydrology in support of physical climate and global biogeochemical studies: An example using the Zambezi River, Surv Geophys, 12, 271–311, https://doi.org/10.1007/BF01903422, 1991.

Vörösmarty, C. J., Moore, B., Grace, A. L., Gildea, M. P., Melillo, J. M., Peterson, B. J., Rastetter, E. B., and Steudler, P. A.: Continental scale models of water balance and fluvial transport: An application to South America, Global Biogeochem. Cycles, 3, 241–265, https://doi.org/10.1029/GB003i003p00241, 1989.

Vörösmarty, C. J., Federer, C. A., and Schloss, A. L.: Potential evaporation functions compared on US watersheds: Possible implications for global-scale water balance and terrestrial ecosystem modeling, Journal of Hydrology, 207, 147–169, https://doi.org/10.1016/S0022-1694(98)00109-7, 1998.

Zaveri, E., Grogan, D. S., Fisher-Vanden, K., Frolking, S., Lammers, R. B., Wrenn, D. H., Prusevich, A., and Nicholas, R. E.: Invisible water, visible impact: groundwater use and Indian agriculture under climate change, Environ. Res. Lett., 11, 084005, https://doi.org/10.1088/1748-9326/11/8/084005, 2016.

Zuidema, S., Grogan, D., Prusevich, A., Lammers, R., Gilmore, S., and Williams, P.: Interplay of changing irrigation technologies and water reuse: Example from the Upper Snake River Basin, Idaho, USA, 24, 5231–5249, https://doi.org/10.5194/hess-24-5231-2020, 2020.

**Reviewer #5**

Overall comments:

The authors present and describe a new open-source version of the global hydrologic model WBM, emphasizing new capabilities for tracking water sources. The paper is well written, and the overview is fairly comprehensive, including theory, examples, plentiful references to earlier literature, and a discussion of how this open-source version relates to other versions of WBM that have been used over the 3 decades since the first version was created. Not only is the model now open source, but the authors have provided a Singularity container to simplify access/usage. Overall, this is a nice contribution, and I recommend publication after minor revisions.

**We appreciate the reviewer's endorsement for this manuscript. We thank the reviewer for making numerous straightforward editorial suggestions that we agree with.**

Below are listed specific comments keyed to particular line numbers, sections, or equations:

Introduction: I appreciate the overview of applications of GHMs, which seems like a useful entry point for those new to the topic.

**Thanks**

Sec 1.1: the need for water tracking is well motivated here.

69-70 typo?

**Yes, it is a typo and we have corrected it.**

86-88 at some point around this section it would be useful to describe how WBM handles gridding. Here an example is given of a fixed-width (120 m) grid, and later examples are noted of lat-lon based grids. Does the user have a choice between these? Does the model account for the variable size of fixed-longitude grid boxes? What happens at the poles? Or does the global configuration exclude very high latitude regions like Antarctica?

**We added a description of how WBM handles gridding to Section 2.1 General Overview, as well as briefly to the Figure 1 caption.**

124 and ff: thank you for listing units of each variable. **You're welcome!**

Eq 3 & 4, should this be P^e? (also, in general, instead of introducing equations with 'defined according to', it can be helpful to say something more descriptive like 'so-and-so depends on temperature T and precipitation rate P according to')

**The equations are correct as written. We do not have elevation dependency of precipitation represented in the model and apply a uniform precipitation rate over each**

**pixel. Therefore, we only identify elevation differences in the forms (frozen or rain) of precipitation.**

Eq 4 and other math: if you want text-like typesetting, e.g., the word "if" in eq 4 or a sub/super-script like "max", use \text{if}, W_i^\text{max}, etc. (requires \usepackage{amsmath})

**The journal's typesetting process will handle this**

142 grammar around lapse rates

**We corrected the grammar error.**

158 is Pt a user-defined param? A fixed fraction of P? Calculated in some other way?

**$P_t$ is the throughfall flux. This is described in the text as "Throughfall is then calculated as the amount of water over-filling the canopy interception pool, while accounting for evaporation over the course of the time-step."**

Eq 7 and others: consider the more traditional use of a dot (\cdot) or no symbol at all to represent multiplication, as opposed to an asterisk (which I think traditionally means convolution, even though most modern programming languages use it as a multiplication operator)

**We changed it to a \cdot.**

212 no cap

**fixed**

eq 13 ff: this is somewhat confusing because the phrase 'immediately moved' suggests a discontinuity but the differential equation suggests differentiability / continuity. Please clarify (maybe via a delta operator in front of R_EXC, which is 1 if volume of retention pool exceeds the threshold and 0 otherwise?)

**We edited this equation (which is now Eq. 16) to better represent the immediate pulse of water, and added text to clarify this process.**

250ff isn't there a unit mis-match between W and R in eqs 14-18? And between R and T in 18?

**There is indeed a unit mismatch. We fixed this by correcting the equations to include the *dt* time step multiplication that leads to the correct units.**

286 unclear what 'stock' means here (it seems to be a standard term with WBM, so please define it before using)

**We use the term stock interchangeably with the term pool to denote a control volume. The term stock is widely used in ecological modeling.**

Sec 2.2.3: I appreciate the references to papers that describe the routing methods, but it would still be helpful to have a bit more information on linear reservoir routing. For example, does it mean that each grid cell's river discharge output to its neighbor is calculated as a linear reservoir, that is, as a function of river water within the cell? What, briefly, is the basis for assigning a reservoir coefficient? Are these constant or do they depend, for instance, on channel geometry?

**Yes, the linear reservoir scheme calculates reach outflow as a function of water within each pixel, and the release coefficient is a function of estimated celerity and reach length. These details are described in the technical documentation accompanying the model code on our GitHub repository, which has now been added as a supplement to the manuscript.**

418 does 'scaler' mean 'scaling factor' or 'scalar' or something else?

**We revised the term to scaling factor.**

429-437 Can you elaborate a bit on this treatment and why it is needed?

**Because WBM calculates a uniform soil water balance over potentially large pixels, water can be withdrawn in larger pulses when soil moisture falls below the crop depletion factor than is typical in practice. This formulation provides an alternative to more closely approximate the practice of irrigating a rotating subset of crops continuously. We added text to explane of the utility of this option.**

473 capitalization consistency

**fixed**

Sec 2.2.7 source tracking: this section is a bit confusing, partly (I think) because of the wide variety of sources that could be tracked. If I understand right, a user would not normally track ALL of these sources in any given model run, but rather would pick a type of source to track - is that right? Actually, reading forward in section 4, we learn that there are 3 options. It would be helpful to list these options up here in section 2. In addition, a couple of examples would potentially help a lot. They need not be very elaborate, but could be as simple as something like: 'a user interested in X might choose to track sources Y and Z'.

545 again, are these 3 mutually exclusive (i.e., one chooses from among them), or are all 3 tracked simultaneously? (later text suggests the former, but at this point in the text it is not clear)

**We added detail to Section 2.2.7 describing the inter-relationship between the tracking components, and also added some examples to help clarify how to use the tracking components simultaneously.**

557 daily time step: helpful to mention this much earlier.

**We agree, and added this to the Figure 1 caption and to Section 2.1 General Overview.**

Sec 3: helpful to define what you mean by validation (I'm not personally a stickler for semantics, but some would consider the term problematic, and better described by confirmation, testing, or evaluation).

**We agree, and find that the term "evaluation" is best for this context. We've changed "validation" to "evaluation" where appropriate. The term "validation" is only retained to describe the metrics used ('validation metric' is a commonly used term), or where prior studies used the term validation and we are referencing those studies.**

Sec 3.1: are the summarized validation studies performed in conjunction with some kind of calibration / parameter optimization? Or is calibration only used in regional applications? You kind of answer this question around 635-640 but it would be helpful to clarify near the start of this section.

**We added the following text: "**we refer readers to the publications themselves for descriptions of parameter value choices, as the level of calibration and the setting of default parameters varies depending on the study." **Because each study described included its own level of calibration, and it would be redundant and lengthy to describe them all here. However, we note that WBM parameters in most previous studies were not subject to rigorous calibration, with the exceptions being Zuidema et al. (2018, 2020) and Samal et al. (2017).**

**To help readers better understand typical ranges of WBM parameters, we improved Table 1 (which was previously Table 2) to include more parameters, a better description of them, and include reasonable ranges of values. A full list of WBM parameters is provided in the document WBM_Usage_and_Input_Reference.xlsx, which is on the WBM GitHub page.**

Validation generally: it would be interesting to summarize some of the lessons from testing and validation, it terms of what might be behind systematic under- or over-prediction of discharge. For example, have past validation exercises revealed certain gaps in knowledge, and/or mathematical approximations that would need to be refined in order to improve model performance? This might fit well under Results or Discussion.

**This is an important point; however, we feel this would be better treated in a separate paper with a dedicated focus. We currently are working on a project centered around uncertainty quantification and with respect to on-going work focused on the conterminous United States; however, broader evaluations that include WBM should be considered for future work.**

**We added text that contextualizes results from model intercomparison projects that identify common constraints of macro-scale modeling and the importance of including human components into large-scale hydrologic models (Nazemi and Wheater, 2015, Veldkamp et al. 2018, Zaherpour et al. 2018).**

Sec 3.2: thanks for differentiating between validation of different versions, and including this section devoted to the open-source version. It's a nice reminder (and demonstration) that testing

of models should ideally include the specific code implementation alongside the theory and numerical algorithms.

Eq 31: the first time I read this, my mind immediately went to cancellation of errors -- but then realized that this is actually desirable for a bias metric. You might consider reversing the order of 32 and 31, and introducing the MBE with a phrase like 'in order to measure systematic bias' or something to that effect, so readers don't get hung up on it.

**Thank you, we revised the order to follow your suggestion.**

711 observations per year, or total?

**Total, which is now clarified in the text.**

753 tense

**corrected**

Sec 5: I appreciate the code history and summary of different versions

943 uniformly spaced... in geographic coords? (again, helpful to explain grid set up early in the paper)

**Agreed, we use uniformly spaced gridded data, typically in geographic coordinates, but the model does work with any projection recognized by GDAL.**

1039 typo

**corrected**

**References:**

**GDAL/OGR contributors: GDAL/OGR Geospatial Data Abstraction software Library., https://doi.org/10.5281/zenodo.5884351, 2022.**

**Nazemi, A. and Wheater, H. S.: On inclusion of water resource management in Earth system models - Part 1: Problem definition and representation of water demand, 19, 33–61, https://doi.org/10.5194/hess-19-33-2015, 2015.**

**Oreskes, N., Shrader-Frechette, K., & Belitz, K. (1994). Verification, validation, and confirmation of numerical models in the earth sciences. Science, 263(5147), 641-646.**

**Samal, N. R., Wollheim, W., Zuidema, S., Stewart, R., Zhou, Z., Mineau, M., Borsuk, M., Gardner, K., Glidden, S., Huang, T., Lutz, D., Mavrommati, G., Thorn, A., Wake, C., and Huber, M.: A coupled terrestrial and aquatic biogeophysical model of the Upper Merrimack River watershed, New Hampshire, to inform ecosystem services evaluation and**

management under climate and land-cover change, 22, 18, https://doi.org/10.5751/ES-09662-220418, 2017.

Veldkamp, T. I. E., Zhao, F., Ward, P. J., de Moel, H., Aerts, J. C. J. H., Müller Schmied, H., Portmann, F. T., Masaki, Y., Pokhrel, Y., Liu, X., Satoh, Y., Gerten, D., Gosling, S. N., Zaherpour, J., and Wada, Y.: Human impact parameterizations in global hydrological models improve estimates of monthly discharges and hydrological extremes: a multi-model validation study, Environ. Res. Lett., 13, 055008, https://doi.org/10.1088/1748- 9326/aab96f, 2018.

**Zaherpour, J., Gosling, S. N., Mount, N., Müller Schmied, H., Veldkamp, T. I. E., Dankers, R., Eisner, S., Gerten, D., Gudmundsson, L., Haddeland, I., Hanasaki, N., Kim, H., Leng, G., Liu, J., Masaki, Y., Oki, T., Pokhrel, Y., Satoh, Y., Schewe, J., and Wada, Y.: Worldwide evaluation of mean and extreme runoff from six global-scale hydrological models that account for human impacts, Environ. Res. Lett., 13, 065015, https://doi.org/10.1088/1748-9326/aac547, 2018.**

**Zuidema, S., Wollheim, W., Mineau, M. M., Green, M. B., and Stewart, R. J.: Controls of Chloride Loading and Impairment at the River Network Scale in New England, 47, 839–847, https://doi.org/10.2134/jeq2017.11.0418, 2018.**

**Zuidema, S., Grogan, D., Prusevich, A., Lammers, R., Gilmore, S., and Williams, P.: Interplay of changing irrigation technologies and water reuse: example from the upper Snake River basin, Idaho, USA, 24, 5231–5249, https://doi.org/10.5194/hess-24-5231-2020, 2020.**

**Citation**: https://doi.org/10.5194/gmd-2022-59-AC1

---

## Referee Report (RR1)

This paper introduces and describes the most recent version (1.0.0) of the University of New Hampshire Water Balance Model with source tracking functionality.  The authors evaluates the current model performance against GRDC river discharge observations and FAO's irrigation withdrawals for the period 2000-2009.  Considering the above-mentioned variables, the model performs well in regions such as North America and poorly in Asia.   The source tracking functionality as demonstrated with examples in the paper distinguishes this model from other GHM's.

Overall, this paper provide a quite thorough description of the model. The source tracking functionality is indeed interesting. However, a few questions should be addressed prior to publication in GMD.

**Abstract**

Line 11:  "WBM was first published in 1989; here we describe the first fully open source WBM version'"

Could the authors please add the model version number here for clarity?

Line 16: "Users can determine what proportion of any flux consists of each of the primary inputs of water to the surface of the terrestrial hydrologic cycle, previously extracted water for human uses, or runoff generated from any place on the Earth's surface."

I believe you want to highlight the fact that a user can know proportions of the primary source in a given flux but is it difficult to grasp such meaning with this kind of sentence formulation and length. Kindly rephrase for clarity.

**Introduction**

Here I would also like to see a closing paragraph on the progression of the WBM models. What was lacking in those models and the reason for the new model? Why should user's care about this new model?

In addition, I suggest explicitly stating your research question, which will improve comprehension.

Eg. The aim of this paper is to provide an overview of the newest model version WBM 1.0.0 by

> 1.  Describing the new model comprehensively
> 2. Showing and discussing standard model output for selected domain X or the entire globe
> 3. Providing insights into model evaluation, and giving guidance for the users of model output (***conditional if the focus is not on providing standard outputs for users***)

**Model description**

The authors really provided a thorough description of the model.

Line 104:  "WBM is modular and is able to accept climate, land use/land cover, water management, and water demand inputs from other models  and data sources…….'",

With Modularity, you mean WBM code (or software) is written in small modules right? If so, you could state the number of modules that makes up WBM or show the software architecture, which will highlight WBM's modularity.

Line 134. "Table 1 presents a cross-section of parameters that are typically..."

Add references to Table 1 if any

**Model Evaluation**

Line 935: **'**Model code': This is well suited for the appendix as it breaks the continuity of the results and dissection and decrease comprehension. You could also briefly summarize the important aspect of the model code (language, code structure, etc.). In addition, refer the readers to the appendix for full and detailed description of the model code in addition to how to use the code

**Discussion**

This study obviously has limitations. I expect to see more of them discussed here.

---

## Author Response (AR2)

Report #1 (Anonymous Referee #2):

I am pleased with the authors' response to my initial set of questions and suggestions. Just a couple exceptions listed below:

Near Line 140 the authors refer to several studies which 'evaluate' the model performance against observations of SWE (snow), but make no effort to quantify the performance skill.

**Response:**

**We have added a summary of the metrics reported in the previous publications' SWE validations to Section 3.1 Published WBM validation and evaluation *Additional validation and evaluation metrics.***

The term 'unsustainable groundwater' makes sense, as thoroughly explained by the authors. My main issue was that the phrase unlimited, unsustainable groundwater seems like an internal contradiction.

**Response:**

**While in reality, unsustainable groundwater must be limited by definition, we use the word "unlimited" here to clarify that the model does not have any cap on the volume of water that can be extracted from this groundwater category. This caveat is very important for understanding the limitation of the model for evaluation sustainability of groundwater resources. We've clarified this in the text in Section 2.2.2 Groundwater.**

Report #2 (Anonymous Referee #6)

This paper introduces and describes the most recent version (1.0.0) of the University of New

Hampshire Water Balance Model with source tracking functionality. The authors evaluates the

current model performance against GRDC river discharge observations and FAO's irrigation

withdrawals for the period 2000-2009. Considering the above-mentioned variables, the model

performs well in regions such as North America and poorly in Asia. The source tracking functionality

as demonstrated with examples in the paper distinguishes this model from other GHM's.

Overall, this paper provide a quite thorough description of the model. The source tracking

functionality is indeed interesting. However, a few questions should be addressed prior to

publication in GMD.

Abstract

Line 11: "WBM was first published in 1989; here we describe the first fully open source

WBM version'"

Could the authors please add the model version number here for clarity?

**Response: we have added the version number**

Line 16: "Users can determine what proportion of any flux consists of each of the primary inputs of
water to the surface of the terrestrial hydrologic cycle, previously extracted water for human uses,
or runoff generated from any place on the Earth's surface."
I believe you want to highlight the fact that a user can know proportions of the primary source in a
given flux but is it difficult to grasp such meaning with this kind of sentence formulation and length.
Kindly rephrase for clarity.

**Response: We have revised this sentence for clarity.**

Introduction

Here I would also like to see a closing paragraph on the progression of the WBM models. What was
lacking in those models and the reason for the new model? Why should user's care about this new
model?

**Response: The progression of WBM models is covered in detail in Section 5, including statements on
what is new about each model.  Given the length of this paper, we prefer not to add redundant text
on this topic to the introduction.**

In addition, I suggest explicitly stating your research question, which will improve comprehension.

Eg. The aim of this paper is to provide an overview of the newest model version WBM 1.0.0 by

1. Describing the new model comprehensively

2. Showing and discussing standard model output for selected domain X or the entire globe

3. Providing insights into model evaluation, and giving guidance for the users of model

output (conditional if the focus is not on providing standard outputs for users)

**Response: We have clarified the scope of the manuscript at the end of section 1.  However, as the
manuscript is a model description paper, we have not framed the manuscript around a research
question and suggest that we can address the referees comment without framing the introduction
around a question.**

Model description

The authors really provided a thorough description of the model.

Line 104: "WBM is modular and is able to accept climate, land use/land cover, water management, and water demand inputs from other models and data sources.......'",

With Modularity, you mean WBM code (or software) is written in small modules right? If so, you could state the number of modules that makes up WBM or show the software architecture, which will highlight WBM's modularity.

**Response: We added a list of the modules that can be turned off or on to this paragraph.**

Line 134. "Table 1 presents a cross-section of parameters that are typically..."

Add references to Table 1 if any

**Response: All parameters in Table 1 are described in the relevant text, which contains all required references for values and ranges.**

Model Evaluation

Line 935: 'Model code': This is well suited for the appendix as it breaks the continuity of the results and dissection and decrease comprehension. You could also briefly summarize the important aspect of the model code (language, code structure, etc.). In addition, refer the readers to the appendix for full and detailed description of the model code in addition to how to use the code

**Response: Other reviewers found this section useful, and we think it is helpful to readers for understanding the model as per the scope and instructions provided by GMD for model description papers. If the editor would like us to move this section to the appendix, we will do so. Otherwise, we suggest leaving it where it is.**

Discussion

This study obviously has limitations. I expect to see more of them discussed here

**Response: The evaluation metrics are intended to quantify the model's limitations. We use four different evaluation metrics, and describe the results of prior studies that report additional metrics for different model fluxes and domains. Since this is a model description paper, and not a research paper, it is somewhat challenging to discuss limitations beyond quantifying the model performance under a default parameter set. As discussed in the paper, regional studies must parameterize and evaluate the model for the given study region, and must then identify the model limitations for that specific region**

**and research question. We have added a sentence emphasizing this regional limit to the discussion section. Further, we do describe the limitations of evaluating the tracking functionality in section 4, and have moved a sentence from the conclusion to the discussion to remind the reader that corroboration of tracking functionality is not yet possible.**

Report #3 (Anonymous Referee #4):

The authors have adequately addressed all comments and I think the manuscript is in a really good shape.

The only minor detail is that (l. 28) there seems to be no reference for Gochis et a., 2020 (which should also be et "al.") and if you want to to distinguish between LSMs and ESMs here, shouldn't it be CESM rather than CLM?

**Response: We have added the Gochis et al. (2020) reference to the list, and fixed the typo. We categorize CLM as the land surface model, and add CESM as the earth system model.**